# Understanding Fairness Surrogate Functions in Algorithmic Fairness

**Wei Yao** $^\star$                                                                     *wei.yao@ruc.edu.cn*
*Gaoling School of Artificial Intelligence*
*Renmin University of China, Beijing*
*Beijing Key Laboratory of Big Data Management and Analysis Methods, Beijing*

**Zhanke Zhou** $^\star$                                                        *cszkzhou@comp.hkbu.edu.hk*
*TMLR Group, Department of Computer Science*
*Hong Kong Baptist University*

**Zhicong Li**                                                                       *zhicongli@ruc.edu.cn*
*Gaoling School of Artificial Intelligence*
*Renmin University of China, Beijing*
*Beijing Key Laboratory of Big Data Management and Analysis Methods, Beijing*

**Bo Han**                                                                          *bhanml@comp.hkbu.edu.hk*
*TMLR Group, Department of Computer Science*
*Hong Kong Baptist University*

**Yong Liu** $^\dagger$                                                              *liuyonggsai@ruc.edu.cn*
*Gaoling School of Artificial Intelligence*
*Renmin University of China, Beijing*
*Beijing Key Laboratory of Big Data Management and Analysis Methods, Beijing*

**Reviewed on OpenReview:** *https://openreview.net/forum?id=iBgmoMTlaz*

## Abstract

It has been observed that machine learning algorithms exhibit biased predictions against certain population groups. To mitigate such bias while achieving comparable accuracy, a promising approach is to introduce surrogate functions of the concerned fairness definition and solve a constrained optimization problem. However, it is intriguing in previous work that such fairness surrogate functions may yield unfair results and high instability. In this work, in order to deeply understand them, taking a widely used fairness definition—demographic parity as an example, we show that there is a *surrogate-fairness gap* between the fairness definition and the fairness surrogate function. Also, the theoretical analysis and experimental results about the "gap" motivate us that the fairness and stability will be affected by the points far from the decision boundary, which is the *large margin points issue* investigated in this paper. To address it, we propose the general sigmoid surrogate to simultaneously reduce both the surrogate-fairness gap and the variance, and offer a rigorous fairness and stability upper bound. Interestingly, the theory also provides insights into two important issues that deal with the *large margin points* as well as obtaining a more *balanced dataset* are beneficial to fairness and stability. Furthermore, we elaborate a novel and general algorithm called Balanced Surrogate, which iteratively reduces the "gap" to mitigate unfairness. Finally, we provide empirical evidence showing that our methods consistently improve fairness and stability while maintaining accuracy comparable to the baselines in three real-world datasets.

---

$^\star$ indicates equal contribution.
$^\dagger$ indicates the corresponding author.

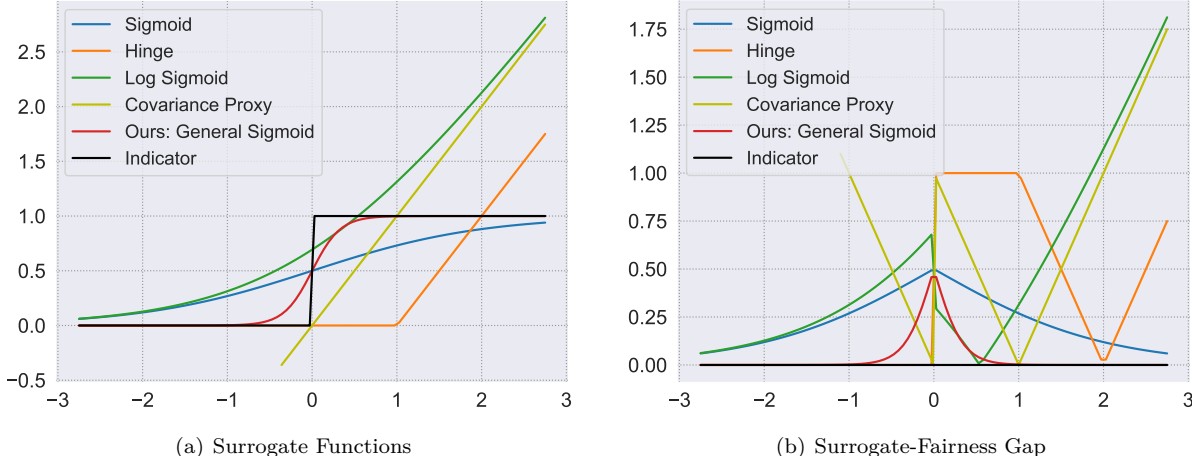

(a) Surrogate Functions          (b) Surrogate-Fairness Gap

Figure 1: (a) Some examples of fairness surrogate functions. The closer the surrogate functions are to the indicator function, the better they represent DP. More details are introduced in Section 3.2. (b) The surrogate-fairness gap of different surrogate functions. It measures the difference between surrogate functions and the indicator function. There is a much smaller gap for our general sigmoid surrogate.

## 1  Introduction

Recently, increasing attention has been paid to the fairness issue in supervised machine learning. That is, although the classifiers seek a higher accuracy, some groups with certain sensitive features (e.g., sex, race, age) may be unfairly treated, which raises ethical problems (Julia Angwin & Kirchner, 2016; Mehrabi et al., 2021; Caton & Haas, 2020). One can be litigated for committing adverse impacts if his/her decision-making process disproportionately treats groups with sensitive attributes (Barocas & Selbst, 2016).

To quantitatively measure the extent of fairness violation, a usual way is adopting the fairness definition, *demographic parity* (DP), which requires the decision makers to accept a roughly equal proportion of each group (Barocas et al., 2019). Existing methods follow the fairness-aware manner of solving a constrained optimization problem, where the learning objective is integrated with the standard loss and a fairness constraint. To incorporate DP into the constraint, fairness surrogate functions are used to replace the indicator function, which is intractable for gradient-based algorithms (Lohaus et al., 2020; Bendekgey & Sudderth, 2021) (refer to Figure 1(a) for some examples). To date, various surrogate functions have been proposed to incorporate fairness definitions into constraints (Wu et al., 2019; Goh et al., 2016; Padh et al., 2021; Zafar et al., 2017a;b;c; Bendekgey & Sudderth, 2021). They are widely applied in various machine learning domains, such as differential privacy (Ding et al., 2020), meta-learning (Zhao et al., 2020), and semi-supervised learning (Zhang et al., 2020). Unfortunately, these surrogate functions encounter two risks. One risk is that if these fairness constraints are used, even when the constraints are perfectly satisfied, there is *no guarantee* whether DP is satisfied (Lohaus et al., 2020). And the fairness surrogate functions may lead to even unfair solutions (Radovanović et al., 2022). Moreover, another risk arises from the high variance issue observed in existing fairness-aware algorithms with surrogate functions (Friedler et al., 2019), rendering them unstable for deployment under fairness requirements (Ganesh et al., 2023).

In this paper, we evaluate these multifarious surrogate functions in algorithmic fairness with both rigorous theorems and extensive experiments. Firstly, we stress the importance of the "**surrogate-fairness gap**", which is *the disparity between the fairness surrogate function and the fairness definition*. It is the decisive factor of whether the fairness surrogate function can lead to fair outcomes and should be minimized. Additionally, we delve into the **variance** of the substitute for DP, highlighting the adverse impact of unbounded surrogates on stability. Drawing upon the inherent property of the surrogate-fairness gap and instability, we conduct an in-depth examination of the **large margin points issue** within the context of unbounded surrogate functions. To reduce the "gap", we propose two solutions to improve the existing surrogate func-

tions: a theoretically motivated fairness surrogate function named *general sigmoid* with upper bounds of the violation of DP, and a novel algorithm called the *balanced surrogate* to iteratively reduce the gap during training.

Our analysis is general for fairness surrogate functions in the case of a common fairness definition, DP. Additionally, we employ a widely recognized covariance proxy (Zafar et al., 2017c) as an illustrative instance of fairness surrogate function. In particular, we first derive the violation of DP for the fairness surrogate functions in Section 4. The violation of DP depends on two factors: the surrogate function itself and the surrogate-fairness gap. The "gap" (shown in Figure 1(b)) directly determines whether a surrogate function is an appropriate substitute for DP. Secondly, we explore the variance of the surrogate function in Section 4.1, emphasizing the detrimental impact of unbounded surrogates on stability. Furthermore, driven by the "gap" and variance, we recognize that large margin points—those data points lying significantly distant from the decision boundary—pose challenges in constraining the fairness and stability for unbounded surrogate functions. This observation is validated through a case study on three real-world datasets in Section 4.2. With theoretical motivations, we introduce the general sigmoid surrogate in Section 5.1 to address large margin points and simultaneously bound the "gap" and variance. We theoretically demonstrate that there is a reliable fairness and stability guarantee for it. Interestingly, the theorems also shed light on the importance of a balanced dataset for both fairness and stability. Furthermore, in Section 5.2, we propose balanced surrogates, a novel and general algorithm that iteratively reduces the "gap" to improve fairness. It is a plug-and-play learning paradigm for the naive fairness-aware training framework using fairness surrogate functions. In the experiments in Section 6 using three real-world datasets, our methods generally enhance fair predictions and stability, while maintaining accuracy comparable to the baselines. Overall, our main contributions are three-fold:

- We demonstrate the importance of *Surrogate-fairness Gap* for fairness surrogate functions and provide an analysis of the *variance*. We emphasize the importance for researchers to consider the impact of the *large margin points issue* on the fairness and stability of unbounded surrogate functions.

- We propose *General Sigmoid Surrogate* and demonstrate that it achieves fairness and stability guarantees. The theoretical results further provide insights to the community that *large margin points issue* needs to be solved and a *balanced dataset* is beneficial to obtain a fairer and more stable classifier.

- We present *Balanced Surrogate*, a novel and general method that iteratively reduces the "gap" to improve the fairness of any fairness surrogate functions.

## 2 Related Work

**Fairness-aware Algorithms.** To mitigate bias, there are various kinds of classical fair algorithms, most of which fall into three categories: pre-processing, in-processing, and post-processing. The pre-processing method is to learn a fair representation that tries to remove information correlated to the sensitive feature while preserving other information for training, e.g., (Calders et al., 2009; Kamiran & Calders, 2011; Zemel et al., 2013; Feldman et al., 2015; Calmon et al., 2017). The downstream tasks then use the fair representation instead of the original biased dataset. The post-processing method is to modify the prediction results to satisfy the fairness definition, e.g., (Kamiran et al., 2012; Fish et al., 2016; Hardt et al., 2016). The in-processing method is to remove unfairness during training. Some intuitive and easy-to-use ideas involve applying fairness constraints (Goh et al., 2016; Zafar et al., 2017a;b;c; Bechavod & Ligett, 2017; Wu et al., 2019; Bendekgey & Sudderth, 2021; Padh et al., 2021) and adding a regularization term to penalize unfairness (Kamishima et al., 2012; Berk et al., 2017; Agarwal et al., 2018; Lohaus et al., 2020; Shui et al., 2022b). Refer to Appendix C.1 for other fairness-aware in-processing approaches. Our paper focuses on in-processing methods, with a particular emphasis on fairness surrogate functions, which are widely used in fairness constraints and fairness regularization methods mentioned above.

**Fairness Surrogate Functions.** Although many existing popular surrogates work well in practice, for example, linear (Donini et al., 2018; Agarwal et al., 2018; Bechavod & Ligett, 2017), ramp (Goh et al.,

2016; Zafar et al., 2017b), convex-concave (Zafar et al., 2017a), hinge (Wu et al., 2019), sigmoid and log-sigmoid (Bendekgey & Sudderth, 2021). They suffer from the same issue: there is not a fairness guarantee for them (Lohaus et al., 2020). And using fairness constraints or regularization can unexpectedly yield unfair solutions (Radovanović et al., 2022). Refer to Appendix C.2 for a meticulous overview of existing works, most of which present counterexamples for analysis. The high variance issue has been observed in existing fairness-aware algorithms (Ganesh et al., 2023), including the instability of the covariance proxy (Friedler et al., 2019). In this paper, in addition to empirically showing counter-examples, we both theoretically and empirically underscore the significance of the surrogate-fairness gap and variance, which are fundamental factors contributing to the two aforementioned problems, respectively. Our general sigmoid surrogate is shown to deal with the large margin points to simultaneously reduce both the surrogate-fairness gap and the variance. There is also fairness and stability upper bound for it, which is crucial in this field (Gallegos et al., 2023; Mehrabi et al., 2021; Caton & Haas, 2020). Additionally, in order to reduce the gap, we also devise a balanced surrogate approach to further improve the fairness and stability of surrogate functions, which may deserve a deeper exploration in this field for future work.

## 3  Preliminaries

### 3.1  Fairness-aware Classification

Note the general purpose of fairness-aware classification is to find a classifier with minimal accuracy loss while satisfying certain fairness constraints. For simplicity, we set up the problem as the binary classification task with only a binary-sensitive feature: with the training set $\mathcal{S} = \{(\mathbf{x}_i, y_i)\}_{i=1}^{N}$ consisting of feature vectors $\mathbf{x}_i \in R^d$ and the corresponding class labels $y_i \in \{0, 1\}$, one needs to predict the labels of a test set. Let $d_\theta(\mathbf{x})$ denotes the signed distance between the feature vector $\mathbf{x}$ and the decision boundary parameterized by $\theta$. Given a point $\mathbf{x}_i$ in the test set, a classifier will predict it as positive if $d_\theta(\mathbf{x}_i) > 0$ and zero if $d_\theta(\mathbf{x}_i) \leq 0$. Among the features of $\mathbf{x}$, there is one binary sensitive attribute $z \in \{-1, +1\}$ (e.g., sex, race, age).

As introduced in Section 1, a widely used fairness definition is called the *demographic parity* (DP) (Mehrabi et al., 2021; Caton & Haas, 2020). It states that each protected class should receive the positive outcome at equal rates, i.e.,

$$P(d_\theta(\mathbf{x}) > 0 | z = +1) = P(d_\theta(\mathbf{x}) > 0 | z = -1).$$

And further, the *difference of demographic parity* (DDP) metric (Lohaus et al., 2020) can be used to measure the degree to which demographic parity is violated:

$$DDP = P(d_\theta(\mathbf{x}) > 0 | z = +1) - P(d_\theta(\mathbf{x}) > 0 | z = -1).$$

Then, with this metric, whether a classifier satisfies demographic parity can be determined by the condition $|DDP| \leq \epsilon$, where $\epsilon \geq 0$ is a given threshold.

### 3.2  Surrogate Functions

We divide the training set into four classes according to the predicted labels and sensitive features:

$$\mathcal{N}_{1a} = \{(\mathbf{x}_i, y_i) \in \mathcal{S} \mid d_\theta(\mathbf{x}_i) > 0, z_i = +1\}, \qquad \mathcal{N}_{1b} = \{(\mathbf{x}_i, y_i) \in \mathcal{S} \mid d_\theta(\mathbf{x}_i) > 0, z_i = -1\},$$
$$\mathcal{N}_{0a} = \{(\mathbf{x}_i, y_i) \in \mathcal{S} \mid d_\theta(\mathbf{x}_i) \leq 0, z_i = +1\}, \qquad \mathcal{N}_{0b} = \{(\mathbf{x}_i, y_i) \in \mathcal{S} \mid d_\theta(\mathbf{x}_i) \leq 0, z_i = -1\},$$

where $N_{1a}, N_{1b}, N_{0a}, N_{0b}$ are sizes of $\mathcal{N}_{1a}, \mathcal{N}_{1b}, \mathcal{N}_{0a}, \mathcal{N}_{0b}$, respectively. To consider DDP as fairness constraints for optimization, the probability in it cannot be computed directly, so frequency is used to estimate them:

$$
\begin{aligned}
\widehat{DDP}_\mathcal{S} &= \frac{N_{1a}}{N_{1a} + N_{0a}} - \frac{N_{1b}}{N_{1b} + N_{0b}} \\
&= \frac{\sum_{(\mathbf{x},y) \in \mathcal{N}_{1a} \cup \mathcal{N}_{0a}} \mathbb{1}_{d_\theta(\mathbf{x}) > 0}}{N_{1a} + N_{0a}} - \frac{\sum_{(\mathbf{x},y) \in \mathcal{N}_{1b} \cup \mathcal{N}_{0b}} \mathbb{1}_{d_\theta(\mathbf{x}) > 0}}{N_{1b} + N_{0b}},
\end{aligned}
\tag{1}
$$

where $\mathbb{1}_{[\cdot]} : \mathbb{R} \to \{0, 1\}$ is the indicator function that returns 1 if the condition is true and 0 otherwise.

In application, $\widehat{DDP}_{\mathcal{S}}$ usually serves as a substitute for $DDP$ to judge the fairness of a classifier. However, due to $\mathbb{1}_{d_\theta(\mathbf{x})>0}$, it is intractable to directly incorporate $\widehat{DDP}_{\mathcal{S}}$ into constraints for gradient-based algorithms. So smooth **surrogate function** $\phi : \mathbb{R} \to \mathbb{R}$ is used to replace $\mathbb{1}_{d_\theta(\mathbf{x})>0}$ with $\phi(d_\theta(\mathbf{x}))$:

$$\widetilde{DDP}_{\mathcal{S}}(\phi) = \frac{\sum_{(\mathbf{x},y)\in\mathcal{N}_{1a}\cup\mathcal{N}_{0a}} \phi(d_\theta(\mathbf{x}))}{N_{1a} + N_{0a}} - \frac{\sum_{(\mathbf{x},y)\in\mathcal{N}_{1b}\cup\mathcal{N}_{0b}} \phi(d_\theta(\mathbf{x}))}{N_{1b} + N_{0b}}. \tag{2}$$

Then $\widetilde{DDP}_{\mathcal{S}}(\phi)$ can be incorporated into constraints as $\left|\widetilde{DDP}_{\mathcal{S}}(\phi)\right| \leq \epsilon$, where $\epsilon$ is the threshold. In this way, the fairness-aware classification problem becomes a feasible constrained optimization problem.

One popular fairness surrogate function is *covariance proxy* (CP), which is introduced by (Zafar et al., 2017c). Empirical study shows that CP can reflect the difference of demographic parity and can be incorporated as constraints for fairness-aware classification problem (Ding et al., 2020; Zhao et al., 2020; Zhang et al., 2020). We defined a general version of it as

$$\widehat{Cov}_{\mathcal{S}}(\phi) = \frac{1}{N} \sum_{i=1}^{N} (z_i - \overline{z})\phi(d_\theta(\mathbf{x}_i)), \tag{3}$$

where $\overline{z}$ is the mean of $z$ over the training set. If $\phi(x) = x$, the equation (3) recovers the original definition of CP in (Zafar et al., 2017c). Also, $\widetilde{DDP}_{\mathcal{S}}(\phi) \propto \widehat{Cov}_{\mathcal{S}}(\phi)$ and the proof can be found in Appendix A.1. It means that *the original CP is equivalent to the linear surrogate function* $\phi(x) = x$, which is also explained in previous work (Lohaus et al., 2020; Bendekgey & Sudderth, 2021). We provide theoretical results for $\widetilde{DDP}_{\mathcal{S}}(\phi)$ in the main paper, and extend them to $\widehat{Cov}_{\mathcal{S}}(\phi)$ in Appendix A.2 with the same conclusions.

## 4 The Surrogate-fairness Gap

We first emphasize the importance of surrogate-fairness gap, and then point out two issues: instability (Section 4.1) and large margin points (Section 4.2), which may influence the surrogate-fairness gap.

Firstly, we connect $\widehat{DDP}_{\mathcal{S}}$ and $\widetilde{DDP}_{\mathcal{S}}(\phi)$ together, and build the surrogate-fairness gap between them.

**Proposition 1.** *Define the magnitude of the signed distance by* $D_\theta(\mathbf{x})$, *i.e.,* $D_\theta(\mathbf{x}) = |d_\theta(\mathbf{x})|$. *It satisfies:*

$$\underbrace{\widehat{DDP}_{\mathcal{S}} - \widetilde{DDP}_{\mathcal{S}}(\phi)}_{\text{surrogate-fairness gap}} = \frac{\sum_{(\mathbf{x},y)\in\mathcal{N}_{1a}\cup\mathcal{N}_{0a}} \left[\mathbb{1}_{d_\theta(\mathbf{x})>0} - \phi(d_\theta(\mathbf{x}))\right]}{N_{1a} + N_{0a}} - \frac{\sum_{(\mathbf{x},y)\in\mathcal{N}_{1b}\cup\mathcal{N}_{0b}} \left[\mathbb{1}_{d_\theta(\mathbf{x})>0} - \phi(d_\theta(\mathbf{x}))\right]}{N_{1b} + N_{0b}}. \tag{4}$$

There is a surrogate-fairness gap between $\widehat{DDP}_{\mathcal{S}}$ and $\widetilde{DDP}_{\mathcal{S}}(\phi)$. For $\widetilde{DDP}_{\mathcal{S}}(\phi)$, it can serve as a fairness constraint or regularization in the algorithm. The algorithm will automatically find a solution that penalizes large $\left|\widetilde{DDP}_{\mathcal{S}}(\phi)\right|$. Unfortunately, it is different for the "gap", which comes from the inherent difference between the indicator function and the fairness surrogate function. For the ideal case $\phi(x) = \mathbb{1}_{x>0}$, which means that $\phi(d_\theta(\mathbf{x})) = \mathbb{1}_{d_\theta(\mathbf{x})>0}$, the gap is zero and reducing the constraint or regularization term $\left|\widetilde{DDP}_{\mathcal{S}}(\phi)\right|$ is equivalent to reducing $\left|\widehat{DDP}_{\mathcal{S}}\right|$. However, for any surrogate function $\phi$, the gap will be inevitably introduced unless $\phi(x) = \mathbb{1}_{x>0}$. When the surrogate-fairness gap is small enough, there is fairness guarantee for the classifier. In practice, Figure 1(b) shows the surrogate-fairness gap of different fairness surrogate functions, including CP (which is equivalent to linear surrogate function) (Zafar et al., 2017c), hinge (Wu et al., 2019), log-sigmoid as well as sigmoid (Bendekgey & Sudderth, 2021), and our general sigmoid surrogate. It suggests that unbounded surrogate functions tend to exhibit a larger surrogate-fairness gap. The bounded surrogate functions, such as sigmoid and general sigmoid, both exhibit a bounded "gap".

### 4.1 Instability

The variance of $\widetilde{DDP}_{\mathcal{S}}(\phi)$ is $Var\left(\widetilde{DDP}_{\mathcal{S}}(\phi)\right) = \mathbb{E}\left(\widetilde{DDP}_{\mathcal{S}}(\phi)\right)^2 - \left[\mathbb{E}\left(\widetilde{DDP}_{\mathcal{S}}(\phi)\right)\right]^2$. If we choose bounded surrogate $\phi(x) \in [0,1]$, then $\widetilde{DDP}_{\mathcal{S}}(\phi) \in [-1,1]$, which means that $Var\left(\widetilde{DDP}_{\mathcal{S}}(\phi)\right) \in [0,1]$. Therefore, there is an stability guarantee for $\widetilde{DDP}_{\mathcal{S}}(\phi)$ if $\phi(x) \in [0,1]$. However, if we choose unbounded surrogate function (such as $\phi(x) = x \in [-\infty, +\infty]$ for the original CP), the resulting values of $\phi(x)$ are not constrained within the range $[0,1]$. Therefore, we cannot conclude that $\widetilde{DDP}_{\mathcal{S}}(\phi) \in [-1,1]$. Consequently, we also cannot conclude that $Var\left(\widetilde{DDP}_{\mathcal{S}}(\phi)\right) \in [0,1]$. As a result, there is no longer a stability guarantee for $\widetilde{DDP}_{\mathcal{S}}(\phi)$.

To summarize the aforementioned problems, incorporating $\widetilde{DDP}_{\mathcal{S}}(\phi)$ into fairness regularization and constraints to indirectly minimize $\widehat{DDP}_{\mathcal{S}}$ may encounter difficulties for two reasons. Firstly, due to the existence of the surrogate-fairness gap, minimizing $\widetilde{DDP}_{\mathcal{S}}(\phi)$ is not equivalent to minimizing $\widehat{DDP}_{\mathcal{S}}$. Secondly, if unbounded surrogate functions are employed, the uncontrollable variance of $\widetilde{DDP}_{\mathcal{S}}(\phi)$ makes it even more challenging to be an appropriate estimator of $\widehat{DDP}_{\mathcal{S}}$.

### 4.2 The Large Margin Points

In this section, we emphasize the trouble of large margin points, which may influence the surrogate-fairness gap. In this paper, the points with too large $D_\theta(\mathbf{x})$ are called *large margin points* and others are normal points. Unfortunately, we regret to assert that these large margin points may simultaneously worsen the first two issues mentioned above. To illustrate, we take CP (linear surrogate $\phi(x) = x$) as an example. For three famous real-world data sets, Adult (Kohavi, 1996), COMPAS (Julia Angwin & Kirchner, 2016) and Bank Marketing (S. Moro & Rita, 2014), we provide the boxplot of $d_\theta(\mathbf{x})$ in the test set in Figure 2. The experimental details are in Appendix B.2. There are three main observations in Figure 2: $(i)$ Most of the points are near the decision boundary. $(ii)$ Over 5% points are large margin points for Adult and COMPAS. $(iii)$ Almost all the large margin points are predicted as positive class.

Firstly, for the surrogate-fairness gap problem, the gap in Equation (1) may be amplified in the presence of such large margin points. For example, Figure 2 shows that most of the large margin points are predicted positive, so there is not a tight bound for $\sum_{(\mathbf{x},y)\in\mathcal{N}_{1a}} d_\theta(\mathbf{x})$ and $\sum_{(\mathbf{x},y)\in\mathcal{N}_{1b}} d_\theta(\mathbf{x})$. Also, Figure 2 suggests that the points with negative prediction exhibit a relatively smaller $|d_\theta(\mathbf{x})|$ comparing to those points with positive prediction. Thus, $\sum_{(\mathbf{x},y)\in\mathcal{N}_{0a}} d_\theta(\mathbf{x})$ and $\sum_{(\mathbf{x},y)\in\mathcal{N}_{0b}} d_\theta(\mathbf{x})$ are bounded (for instance, In Figure 2(b), we have $\left|\sum_{(\mathbf{x},y)\in\mathcal{N}_{0a}} d_\theta(\mathbf{x})\right| \leq 2N_{0a}$). Therefore, there is still not a tight bound for $\sum_{(\mathbf{x},y)\in\mathcal{N}_{1a}\cup\mathcal{N}_{0a}} d_\theta(\mathbf{x})$ and $\sum_{(\mathbf{x},y)\in\mathcal{N}_{1b}\cup\mathcal{N}_{0b}} d_\theta(\mathbf{x})$, which may lead to a large surrogate-fairness gap in Equation (1). Finally, if the surrogate-fairness gap becomes large, constraining the fairness surrogate function is inconsistent with the specific fairness definition, which may lead to unfair result.

Secondly, regarding the instability issue, while the majority of points are close to the decision boundary, a small number of large margin points contribute to the increased variance of $d_\theta(\mathbf{x})$, thereby influencing both $\widetilde{DDP}_{\mathcal{S}}(\phi)$ and $Var\left(\widetilde{DDP}_{\mathcal{S}}(\phi)\right)$. The presence of large margin points, along with the use of an unbounded surrogate function, surpasses the constraint on $Var\left(\widetilde{DDP}_{\mathcal{S}}(\phi)\right)$ and may result in unstable fairness guidance for the classifier. These analytical insights above will be further validated through our experiments.

## 5 Our Approach

We devise the general sigmoid surrogate function with fairness and stability guarantees in Section 5.1. The theory suggests that addressing the large margin points issue and obtaining a more balanced dataset contribute to a fairer classifier. Then we present our balanced surrogates in Section 5.2, which is a novel iterative approach to reduce the "gap" and thus improving fairness.

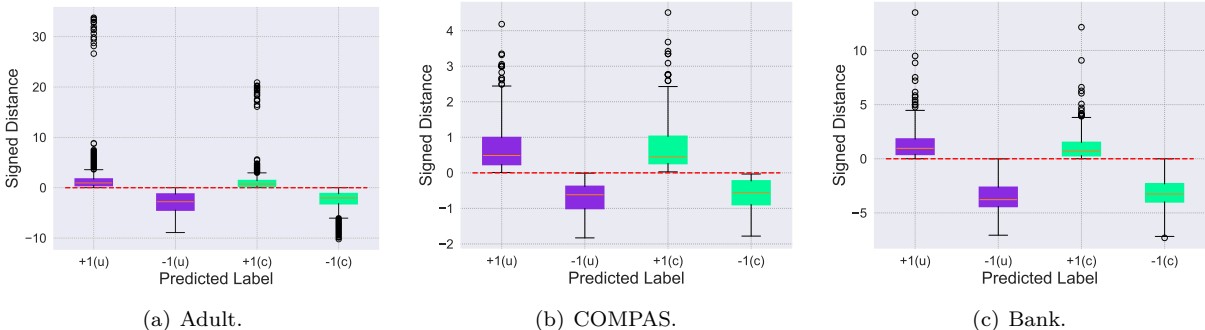

Figure 2: The boxplot for the unconstrained logistic classifier (u) and logistic classifier with fairness constraints using linear surrogate (c) in three datasets. $+1$ and $-1$ represent the predicted label. The red dashed line means $d_\theta(\mathbf{x}) = 0$. The orange line in the box is the median. The circles outside the box are large margin points. The rates of large margin points are 7.61%, 5.12% and 0.82% respectively.

## 5.1 General Sigmoid Surrogates

We generalize sigmoid function as

$$G(x) = \sigma(wx), \tag{5}$$

where $\sigma(x)$ is the sigmoid function and $w > 0$ is the parameter. The general sigmoid surrogate is flexible because of the adjustable $w$. Moreover, of paramount importance, it achieves a much lower surrogate-fairness gap, making it more consistent with DP, which is shown in Figure 1(b). Additionally, it enjoys stability guarantees as its values fall within the range $[0, 1]$, ensuring that $Var\left(\widetilde{DDP}_{\mathcal{S}}(G)\right) \in [0, 1]$.

### 5.1.1 Fairness Guarantees

The following Theorem 1 provides the upper bound of $\left|\widehat{DDP}_{\mathcal{S}}\right|$ when $G(D_\theta(\mathbf{x}))$ is close to 1 for all the points under the $\widetilde{DDP}_{\mathcal{S}}(\phi)$ fairness constraint.

**Theorem 1.** *We assume that $G(D_\theta(\mathbf{x})) \in [1 - \gamma, 1]$, where $\gamma > 0$. $\forall \epsilon > 0$, if $\left|\widetilde{DDP}_{\mathcal{S}}(G)\right| \le \epsilon$, then it holds:*

$$\left|\widehat{DDP}_{\mathcal{S}}\right| \le \frac{1}{2}\epsilon + \gamma. \tag{6}$$

The proof can be found in Appendix A.3. The first term is similar to that in (4). Now with the assumption $G(D_\theta(\mathbf{x})) \in [1 - \gamma, 1]$, the gap here can be limited to a small range of variation. If the general sigmoid surrogates limit $G(D_\theta(\mathbf{x}))$ to around 1 so that $\gamma$ is small enough, then there are fairness guarantees for the classifier. In contrast, the gap for CP is influenced by the magnitude of $D_\theta(\mathbf{x})$ for every large margin point and thus hard to be bounded.

**Remark.** *In Appendix D.2, considering CP, we provide extensions of Theorem 1 to other five fairness definitions: three kinds of disparate mistreatment (Theorem 5-10 in Appendix D.2) and balance for positive (negative) class (Theorem 11-12 in Appendix D.4). In particular, CP is generalized to disparate mistreatment (Zafar et al., 2017a), and we theoretically devise the form of CP to better meet with disparate mistreatment, which is empirically validated in (Zafar et al., 2019).*

However, in some cases, we do not need to guarantee that the assumption $G(D_\theta(\mathbf{x})) \in [1 - \gamma, 1]$ holds for all the points. So the theorem below relaxes the assumption by giving the upper bound of $\left|\widehat{DDP}_{\mathcal{S}}\right|$ when most of the points satisfy the assumption in Theorem 1.

**Theorem 2.** *We assume that $k$ points satisfy $G(D_\theta(\mathbf{x})) \in [0, 1 - \gamma]$ and others satisfy $G(D_\theta(\mathbf{x})) \in [1 - \gamma, 1]$, where $\gamma > 0$. $\forall \epsilon > 0$, if $\left|\widetilde{DDP}_{\mathcal{S}}(G)\right| \le \epsilon$, then it holds:*

$$\left|\widehat{DDP}_{\mathcal{S}}\right| \leq \frac{1}{2}\epsilon + \gamma + \underbrace{\frac{1}{2}\left(\frac{1}{N_{1a}+N_{0a}} + \frac{1}{N_{1b}+N_{0b}}\right)k}_{relaxation\ factor}. \tag{7}$$

The proof can be found in Appendix A.4. Comparing Theorem 1 with Theorem 2, the relaxation of the assumption produces an extra relaxation factor. According to Theorem 1, if $w$ is large, then $\gamma$ can be small enough, thus leading to a classifier with fairness guarantee. But an arbitrarily too large $w$ makes training more challenging because of the diminished gradient magnitude for general sigmoid surrogate. Theorem 2 tells us that we need not to assume that all points satisfy $G(D_\theta(\mathbf{x})) \in [1 - \gamma, 1]$. Few points violating the assumption (a small $k$) can also be tolerated. So it supports the idea that we do not have to choose a large $w$ because a relatively small $w$ can also guarantee fairness.

### 5.1.2 Insights from the Theorems

**Large Margin Points Issue.** An obvious takeaway of Theorem 2 is that addressing the large margin points issue makes $k$ lower and thus obtaining a tighter bound of $\left|\widehat{DDP}_{\mathcal{S}}\right|$. While Bendekgey & Sudderth (2021) also explores the influence of outliers on the model under fairness constraints, their theoretical analysis in primarily centers on loss degeneracy under fairness constraints, whereas our paper aims to establish upper bounds concerning fairness.

**Balanced Dataset.** Another interesting take-away of Theorem 1-2 is that we need a *balanced dataset* to obtain a fairer classifier. The approximately same number between two sensitive groups contributes to a tighter $\left|\widehat{DDP}_{\mathcal{S}}\right|$ upper bound. First of all, for Theorem 1, the coefficient of $\epsilon$ satisfies $\frac{N^2}{4(N_{1a}+N_{0a})(N_{1b}+N_{0b})} \geq 1$. The equality holds if and only if $N_{1a} + N_{0a} = N_{1b} + N_{0b}$, which means that the two sensitive groups share the equal size. Therefore, a more balanced dataset will obtain a tighter $\left|\widehat{DDP}_{\mathcal{S}}\right|$ upper bound. Similarly, comparing to Theorem 1, we discover that those $k$ points in Theorem 2 relaxed the original bound by a relaxation factor. The coefficient of the relaxation factor $\frac{1}{2}\left(\frac{N}{N_{1a}+N_{0a}} + \frac{N}{N_{1b}+N_{0b}}\right) \geq 2$, and the equality holds if and only if $N_{1a} + N_{0a} = N_{1b} + N_{0b}$. So, if we use a balanced dataset, then $N_{1a} + N_{0a}$ and $N_{1b} + N_{0b}$ are close to each other, thus making the fairness bounded tighter. The imbalance issue also applies to other surrogates and one can balance the dataset in advance to achieve better fairness performance.

Notably, certain theoretical investigations illuminate the positive impact of *a balanced dataset* on fostering fairness in machine learning. For example, the impossibility theorem (Bell et al., 2023) in fairness literature states that, in the context of binary classification, equalizing some specific set of multiple common performance metrics between protected classes is impossible, except in two special cases: a perfect predictor and equal *base rate* (Chouldechova, 2017; Kleinberg et al., 2017; Pleiss et al., 2017). Furthermore, the reduction of variation in group *base rates* has been demonstrated to yield a diminished lower bound for separation gap and independence gap (Liu et al., 2019). Moreover, minimizing the difference in *base rates* results in a decreased lower bound for joint error across both sensitive groups (Zhao & Gordon, 2022). In contrast, our Theorem 2 provides an elucidation from the perspective of *upper bound*. It implies that a more balanced dataset results in a tighter upper bound on the violation of DP.

**Remark.** *In Appendix A.6, Theorem 4 shows that $Var\left(\widehat{DDP}_{\mathcal{S}}\right) \leq \frac{1}{4}\left(\frac{1}{N_a} + \frac{1}{N_b}\right)$. Therefore, it also highlights the advantage of a balanced dataset in reducing variance.*

## 5.2 Balanced Surrogates

The naive fairness-aware training framework can be formulated as

$$\min \quad L(\theta, \mathbf{x}, y) + \rho \cdot \widetilde{DDP}_{\mathcal{S}}(\phi), \tag{8}$$

where $L(\theta, \mathbf{x}, y) = \frac{1}{N}\sum_{(\mathbf{x},y)\in\mathcal{S}} \ell(\theta, \mathbf{x}, y)$ is the empirical loss over the training set, $\ell$ is a convex loss function, $\widetilde{DDP}_{\mathcal{S}}(\phi)$ is the fairness regularization, and $\rho > 0$ is the coefficient. Recall in Proposition 1 that whether

$\widetilde{DDP}_{\mathcal{S}}(\phi)$ is an appropriate estimation of $\widetilde{DDP}_{\mathcal{S}}$ depends on $\phi$. And as stated in Proposition 1, the fairness surrogate function $\phi$ directly affects the surrogate-fairness gap. Thus, the existence of such "gap" motivates us that it is still necessary to reduce "gap" to improve fairness.

Our balanced surrogates approach mitigates unfairness by treating different sensitive groups differently using a parameter being updated during training. The key idea of the updating procedure is *making the magnitude of "gap" as small as possible*. It is a general plug-and-play learning paradigm for training framework using fairness surrogate functions like (8), which is validated in the experiments.

Specifically, we consider different surrogates for two sensitive groups, i.e.,

$$\phi(d_\theta(\mathbf{x}_i)) = \begin{cases} \phi_1(d_\theta(\mathbf{x}_i)), & z_i = +1. \\ \phi_2(d_\theta(\mathbf{x}_i)), & z_i = -1. \end{cases} \tag{9}$$

With (9), we rewrite $\widetilde{DDP}_{\mathcal{S}}(\phi)$ as:

$$\widetilde{DDP}_{\mathcal{S}}(\phi) = \frac{\sum_{(\mathbf{x},y)\in\mathcal{N}_{1a}\cup\mathcal{N}_{0a}} \phi_1(d_\theta(\mathbf{x}))}{N_{1a} + N_{0a}} - \frac{\sum_{(\mathbf{x},y)\in\mathcal{N}_{1b}\cup\mathcal{N}_{0b}} \phi_2(d_\theta(\mathbf{x}))}{N_{1b} + N_{0b}}, \tag{10}$$

For simplicity, we assume

$$\phi_2(x) = \lambda\phi_1(x), \tag{11}$$

where $\lambda \geq 0$ is the *balance factor* to be updated to reduce the gap. Our objective is

$$\widehat{DDP}_{\mathcal{S}} - \widetilde{DDP}_{\mathcal{S}}(\phi) = 0, \tag{12}$$

which is equivalent to a surrogate-fairness gap of zero. Plug equations (1), (10) and (11) into the equation (12), we then solve for $\lambda$:

$$\lambda = \frac{(N_{1b} + N_{0b})\sum_{(\mathbf{x},y)\in\mathcal{N}_{1a}\cup\mathcal{N}_{0a}} \phi_1(d_\theta(\mathbf{x})) - (N_{1a}N_{0b} - N_{0a}N_{1b})}{(N_{1a} + N_{0a})\sum_{(\mathbf{x},y)\in\mathcal{N}_{1b}\cup\mathcal{N}_{0b}} \phi_1(d_\theta(\mathbf{x}))} \tag{13}$$

In this way, we can first specify $\phi_1$ in (9), initiate $\lambda$ as $\lambda_0$, and train an unconstrained classifier with the produced $\theta_0$ as the start point of the iteration. Then we iteratively solve (8) and compute (13) until convergence. In order to avoid oscillation and accelerate the convergence, we use exponential smoothing for $\lambda$:

$$\lambda_t = \begin{cases} \lambda_0, & t = 0. \\ \alpha\lambda'_t + (1-\alpha)\lambda_{t-1}, & t = 1, 2, \cdots, N. \end{cases} \tag{14}$$

Where $\lambda'_t$ comes from (13) after $t$ iterations, $\lambda_t$ is the result of $\lambda'_t$ after exponential smoothing and $0 \leq \alpha \leq 1$ is the smoothing factor. Notice that $\lambda \leq 0$ is meaningless, so when this happens, we abandon this algorithm and set $\lambda_t$ to 1, which recovers (8). When the difference of $\lambda_t$ between two successive iterations is less than a termination threshold $\eta$, the algorithm is over. If we choose the smoothing factor $\alpha$ and the termination threshold $\eta$ properly, then the loop will terminate after a few runs. The algorithmic representation of the balanced surrogates can be found in Appendix B.1.

## 6 Experiments

### 6.1 Experimental Setup

**Dataset.** We use three real-world datasets: Adult (Kohavi, 1996), Bank Marketing (S. Moro & Rita, 2014) and COMPAS (Julia Angwin & Kirchner, 2016), which are commonly used in fair machine learning (Mehrabi et al., 2021).

- **Adult**. The Adult dataset contains 48842 instances and 14 attributes. The goal is predicting whether the income for a person is more than \$50,000 a year. We consider sex as the sensitive feature with values male and female.

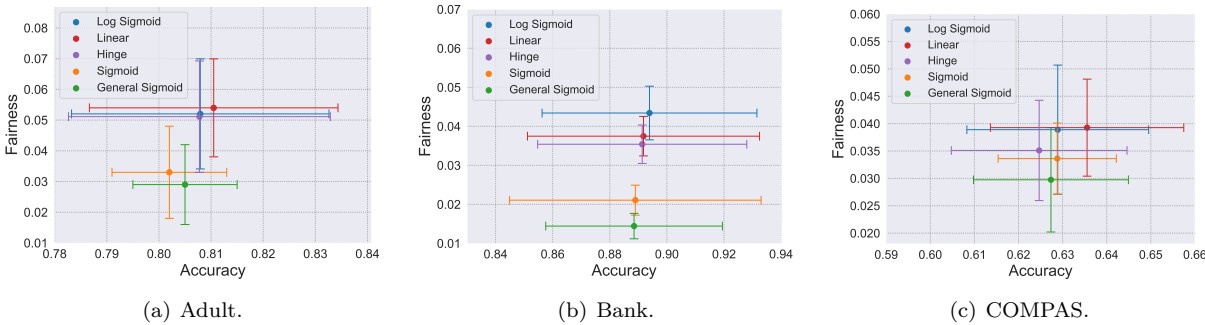

(a) Adult.     (b) Bank.     (c) COMPAS.

Figure 3: Results of different surrogate functions.

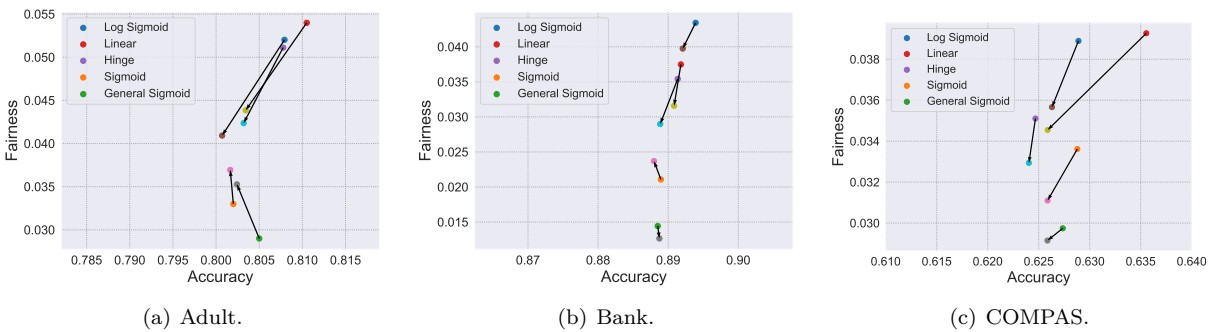

(a) Adult.     (b) Bank.     (c) COMPAS.

Figure 4: Results of applying balanced surrogate to different surrogate functions. The arrow starts with the result of a surrogate and ends with the result of the same surrogate function using balanced surrogate method.

- **Bank**. The Bank Marketing dataset contains 41188 instances and 20 input features. The goal is predicting whether the client will subscribe a term deposit. We follow (Zafar et al., 2017c) and consider age as the binary sensitive attribute, which is discretized into the case whether the client's age is between 25 and 60 years.

- **COMPAS**. The COMPAS dataset was compiled to investigate racial bias in recidivism prediction. The goal is predicting whether a criminal defendant will be a recidivist in two years. We use only the subset of the data with sensitive attribute Caucasian or African-American.

**Baseline.**  In addition to an unconstrained logistic regression classifier (denoted as 'Unconstrained'), we compare our general sigmoid surrogate (denoted as 'General Sigmoid') with other four surrogate functions below, which have also appeared in Figure 1.

- Linear surrogate (equivalent to CP) $\phi(x) = x$ (Zafar et al., 2017c) (denoted as 'Linear').

- Hinge-like surrogate $\phi(x) = \max(x + 1, 0)$ (Wu et al., 2019) (denoted as 'Hinge').

- Sigmoid surrogate $\phi(x) = \sigma(x)$ (Bendekgey & Sudderth, 2021) (denoted as 'Sigmoid').

- Log-sigmoid surrogate $\phi(x) = -\log\sigma(-x)$ (Bendekgey & Sudderth, 2021) (denoted as 'Log-Sigmoid').

## 6.2 Learning Fair Classifiers

We conduct two main experiments. One is the comparison among general sigmoid surrogate and other surrogate functions on classification tasks. The other is validating the effect of balanced surrogates method

by applying it to different surrogate functions. Following Bendekgey & Sudderth (2021), a linear classifier is used as the base classifier. The dataset is randomly divided into training set (70%), validation set (5%) and test set (25%). The parameter setting is discribed in Appendix B.3. We report two metrics on the test set: $\left|\widehat{DDP}_{\mathcal{S}}\right|$ (lower is better) and accuracy (higher is better), and standard deviation is shown for the metrics.

## 6.3   Main Results

The results are in Figures 3-4. Refer to Appendix B.5 for specific numerical results of all three datasets.

**General Sigmoid Surrogate.**   In Figure 3, on the one hand, we observe that the general sigmoid surrogate achieves better fairness than unbounded surrogate functions (Log sigmoid, Linear and Hinge), which indicates that the general sigmoid surrogate does effectively shorten the surrogate-fairness gap and therefore improve fairness. On the other hand, comparing to the sigmoid surrogate function, our proposed surrogate not only further reduces the gap, but it is also more flexible due to the parameter $w$. Moreover, it is intriguing to note that the variance of fairness and accuracy of general sigmoid surrogate is also comparatively smaller than other surrogate functions, demonstrating its superiority in terms of stability. It offers a simple solution to the long-standing high variance issue observed in existing fairness-aware algorithms (Friedler et al., 2019; Ganesh et al., 2023). We provide some theoretical analysis on variance to Appendix A.6. Exploring automated methods to search for a suitable parameter $w$ for improved fairness performance while reducing variance presents an intriguing avenue for future research.

**Balanced Surrogates Method.**   In Figure 4, we observe that balanced surrogates method succeeds in improving fairness of unbounded surrogate functions but sometimes slightly compromising fairness of bounded surrogate functions. For the reason of this phenomenon, fairness-aware algorithms aim to reduce $\left|\widehat{DDP}_{\mathcal{S}}\right|$. Firstly, fairness regularization aims at lowering $\left|\widetilde{DDP}_{\mathcal{S}}(\phi)\right|$. Secondly, the key idea of balanced surrogates is to reduce the magnitude of "gap" $\left|\widehat{DDP}_{\mathcal{S}} - \widetilde{DDP}_{\mathcal{S}}(\phi)\right|$ thus indirectly lower $\left|\widehat{DDP}_{\mathcal{S}}\right|$ (because $\left|\widehat{DDP}_{\mathcal{S}}\right| \le \left|\widetilde{DDP}_{\mathcal{S}}(\phi)\right| + \left|\widehat{DDP}_{\mathcal{S}} - \widetilde{DDP}_{\mathcal{S}}(\phi)\right|$). However, an infinitesimal $\left|\widehat{DDP}_{\mathcal{S}}\right|$ is not always better. In Appendix A.5, we show in Theorem 3 that there is still a discrepancy between $\widehat{DDP}_{\mathcal{S}}$ and $DDP$, indicating that a small enough $\left|\widehat{DDP}_{\mathcal{S}}\right|$ is not equivalent to a small $|DDP|$. When the "gap" is large (such as the un-bounded surrogate functions), balanced surrogates method can effectively reduce "gap" and achieves a fairer result. But when the "gap" is limited (such as sigmoid and general sigmoid), an infinitesimal magnitude of "gap" may sometimes undermine fairness instead. Interestingly, similar to the stability enhancement observed in general sigmoid surrogate, the numerical results in Appendix B.5 also show that our balanced surrogates method attains smaller variance compared to other surrogate functions. Incorporating mechanisms such as the balanced surrogate method into existing fairness-aware algorithms to enhance both fairness and stability is also a promising direction.

In Appendix B.6, we also compare our methods with other two in-processing methods: reduction (Agarwal et al., 2018) and adaptive sensitive reweighting (Krasanakis et al., 2018). The promising results also demonstrate the superiority of our methods.

## 6.4   Experimental Verification for the Theoretical Insights

**Large Margin Points Issue.**   The boxplot with our general sigmoid surrogate is shown in Figure 8 in Appendix B.7. Recall that with an unbounded surrogate function $\phi$, the large margin points will influence the "gap". Now they have a minor impact on the "gap" mentioned before because they are bounded ($G(D_\theta(\mathbf{x})) \le 1$). Furthermore, the two edges almost overlap, indicating that the variation of $D_\theta(\mathbf{x})$ is small, contributing to more stable results. Overall, the general sigmoid surrogate successfully deals with the large margin points, mitigating both the surrogate-fairness gap and instability simultaneously.

**Balanced Dataset.**   From the perspective of sensitive attribute, the Adult dataset is an *imbalanced* dataset: there are 32650 male instances and 16192 female instances. The ratio of the two groups is approximately

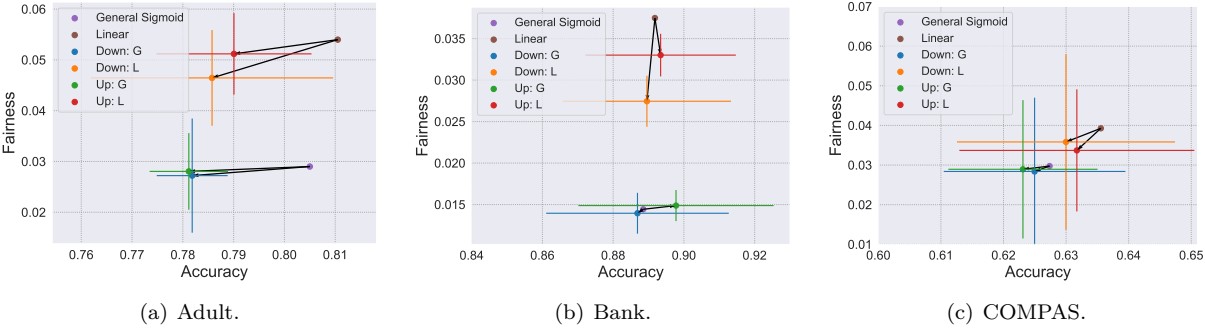

Figure 5: The result of balanced dataset. "G" and "L" indicates general sigmoid surrogate and linear surrogate, respectively. "Up" and "Down" correspond to upsampling and downsampling, respectively.

2:1. The Bank Marketing and COMPAS datasets also suffer from the imbalance issue. The ratio of the two groups are 39210:1978 (about 20:1) and 3175:2103 (about 3:2), respectively. Such imbalanced datasets lead to a loose bound in Theorem 2. We randomly split the dataset into training set (70%) and test set (30%). According to the number of minority group, we conduct two experiments: Downsampling and Upsampling, which means randomly downsampling (upsampling) the majority (minority) group in the original training set and form the new training set to make two demographic groups more balanced. Refer to Appendix B.4 for the details of our sampling schemes. The test set is partitioned in advance so that it is still imbalanced. We choose an unbounded surrogate function: Linear, and our bounded surrogate: general sigmoid.

The results in Figure 5 show that downsampling the majority group and upsampling the minority group contribute to a balanced dataset and a fairer result. However, in our experiments here, downsampling will lead to reduction of the training set, and upsampling will lead to replication of the training set, which may cause underfitting and overfitting problems, respectively. So the accuracy sometimes decreases. In conclusion, before fairness-aware training, we suggest using fair data augmentation strategies to obtain a balanced dataset, such as Fair Mixup (Chuang & Mroueh, 2021), and algorithms designed to address data imbalance, such as SMOTE (Chawla et al., 2002).

# 7 Conclusion

In this paper, we research on surrogate functions in algorithmic fairness. We derive the surrogate-fairness gap to indicate the difference between fairness surrogate function and . With boxplots, we find that unbounded surrogates are especially faced with the large margin points issue, which further amplify the "gap" and instability. To address these challenges, we propose general sigmoid surrogate with theoretically validated fairness and stability guarantees to deal with large margin points. The theoretical analysis further provides insights to the community that dealing with the large margin points issue as well as obtaining a more balanced dataset contribute to a fairer and more stable classifier. We further elaborate balanced surrogates method, which is an iterative algorithm to reduce the gap during training. It is also applicable to other fairness surrogate functions. Finally, our experiments using three real-world datasets not only validate the insights of our theorems, but also show that our methods get better fairness and stability performance.

# 8 Broader Impact and Ethics Statement

This study concentrates on better understanding the fairness surrogate functions in machine learning. Importantly, if someone claims the fairness guarantee of using unbounded fairness surrogate functions, it is worthy of suspicion and further investigation because of the surrogate-fairness gap issue discussed in this paper. Furthermore, the motivation of our general sigmoid surrogate and balanced surrogate methods are both centered on improving the fairness performance.

We acknowledge the sensitive nature of our study and guarantee adherence to all applicable legal and ethical standards. Our research is conducted within a safe and controlled setting to protect real-world systems' security. Only researchers who have received the appropriate clearance can access the most confidential parts of our experiments. Such measures are implemented to preserve the integrity of our research and to reduce any potential risks associated with the experiments.

## Acknowledgement

We sincerely appreciate the anonymous reviewers for their helpful suggestions and constructive comments. WY, ZCL and YL were supported by the National Natural Science Foundation of China (NO.62076234); the Beijing Natural Science Foundation (NO.4222029); the Intelligent Social Governance Interdisciplinary Platform, Major Innovation & Planning Interdisciplinary Platform for the "Double First Class" Initiative, Renmin University of China; the Beijing Outstanding Young Scientist Program (NO.BJJWZYJH012019100020098); the Public Computing Cloud, Renmin University of China; the Fundamental Research Funds for the Central Universities, and the Research Funds of Renmin University of China (NO.2021030199); the Huawei-Renmin University joint program on Information Retrieval; the Unicom Innovation Ecological Cooperation Plan; the CCF Huawei Populus Grove Fund; and the National Key Research and Development Project (NO.2022YFB2703102). ZKZ and BH were supported by the NSFC General Program No. 62376235, Guangdong Basic and Applied Basic Research Foundation Nos. 2022A1515011652 and 2024A1515012399.

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

# Contents

## Appendix

In the appendix, due to the length limit of every line, $\sum_{(\mathbf{x},y)\in\mathcal{N}_{1a}\cup\mathcal{N}_{0a}}\mathbb{1}_{d_\theta(\mathbf{x})>0}$ is denoted as $\sum_{\mathcal{N}_{1a},\mathcal{N}_{0a}}\mathbb{1}_{d_\theta(\mathbf{x})>0}$ for better presentation because it prevents endless line breaks.

We summarize the frequently used notations in Table 1:

Table 1: The most frequently used notations in this paper.

| Notations | Meanings |
|---|---|
| $\mathcal{S}$ | the training set |
| $N$ | the size of the training set |
| $\mathbf{x}$ | feature vector of a data point |
| $y$ | the corresponding label |
| $z$ | the corresponding sensitive attribute |
| $\theta$ | the parameters of the classifier |
| $d_\theta(\mathbf{x})$ | the signed distance between the feature vector and the decision boundary |
| $D_\theta(\mathbf{x})$ | the absolute value of $d_\theta(\mathbf{x})$ |
| $\phi(\cdot)$ | the surrogate function |
| $G(\cdot)$ | general sigmoid surrogate function |
| $\mathcal{N}_{1a}$ | the set of data points with positive prediction and positive sensitive attribute |
| $N_{1a}$ | the size of $\mathcal{N}_{1a}$ |
| $TP_+$ | the number of data points with positive prediction and positive sensitive attribute |
| $DDP$ | the difference of demographic parity |
| $\widehat{DDP}_\mathcal{S}$ | an estimation of DDP |
| $\widetilde{DDP}_\mathcal{S}(\phi)$ | the constraint/regularization term with fairness surrogate function $\phi$ |
| $\widehat{Cov}_\mathcal{S}(\phi)$ | the covariance proxy |
| $U_{1a}$ | The number of points satisfying $\phi(D_\theta(\mathbf{x})) \in [0, 1-\gamma]$ in $\mathcal{N}_{1a}$ |
| $\hat{P}(\cdot)$ | the predicted probability |

## A   Proof for the Main Paper

### A.1   Analysis about CP

In this subsection, we finish the proof of $\widetilde{DDP}_\mathcal{S}(\phi) \propto \widehat{Cov}_\mathcal{S}(\phi)$.

Now this is the proof for the claim that, when degenerating $\phi(x) = x$ (it is a linear function), we have $\widetilde{DDP}_\mathcal{S}(\phi) \propto \widehat{Cov}_\mathcal{S}(\phi)$. It is a step-by-step derivation of (3).

$$\underbrace{\frac{1}{N}\sum_{i=1}^{N}(z_i - \bar{z})d_\theta(\mathbf{x}_i)}_{\widehat{Cov}_\mathcal{S}(\phi),\text{ where }\phi(x)=x.}$$

$$=\frac{1}{N}\left[(1-\overline{z})\sum_{\mathcal{N}_{1a},\mathcal{N}_{0a}}d_\theta(\mathbf{x}_i)+(-1-\overline{z})\sum_{\mathcal{N}_{1b},\mathcal{N}_{0b}}d_\theta(\mathbf{x}_i)\right]$$

$$=\frac{1}{N}\left[\left(1-\frac{N_{1a}+N_{0a}-N_{1b}-N_{0b}}{N}\right)\sum_{\mathcal{N}_{1a},\mathcal{N}_{0a}}d_\theta(\mathbf{x}_i)+\left(-1-\frac{N_{1a}+N_{0a}-N_{1b}-N_{0b}}{N}\right)\sum_{\mathcal{N}_{1b},\mathcal{N}_{0b}}d_\theta(\mathbf{x}_i)\right]$$

$$=\frac{2}{N^2}\left[(N_{1b}+N_{0b})\sum_{\mathcal{N}_{1a},\mathcal{N}_{0a}}d_\theta(\mathbf{x}_i)-(N_{1a}+N_{0a})\sum_{\mathcal{N}_{1b},\mathcal{N}_{0b}}d_\theta(\mathbf{x}_i)\right]$$

$$=\underbrace{\frac{2(N_{1a}+N_{0a})(N_{1b}+N_{0b})}{N^2}}_{\text{constant}\leq\frac{1}{2}<1}\underbrace{\left(\frac{\sum_{\mathcal{N}_{1a},\mathcal{N}_{0a}}d_\theta(\mathbf{x}_i)}{N_{1a}+N_{0a}}-\frac{\sum_{\mathcal{N}_{1b},\mathcal{N}_{0b}}d_\theta(\mathbf{x}_i)}{N_{1b}+N_{0b}}\right)}_{\widehat{DDP}_\mathcal{S}(\phi),\text{ where }\phi(x)=x.}. \tag{15}$$

The proof is complete.

Furthermore, in fact, we have $\widetilde{DDP}_\mathcal{S}(\phi)\propto\widehat{Cov}_\mathcal{S}(\phi)$ holds for all $\phi$. But for the original CP (Zafar et al., 2017c), $\phi(x)=x$ is the default. So the original CP is equivalent to linear surrogate because $\phi(x)=x$. Similar proof is also explained in previous work (Lohaus et al., 2020; Bendekgey & Sudderth, 2021), where they call it "linear relaxations".

## A.2 Extension from Fairness Surrogate Functions to CP

This subsection aims to restate the theoretical analysis about $\widetilde{DDP}_\mathcal{S}(\phi)$ in Proposition 1 and Theorem 1-2 from the perspective of $\widehat{Cov}_\mathcal{S}(\phi)$.

From the above derivations, we have

$$\widehat{Cov}_\mathcal{S}(\phi)=\underbrace{\frac{2(N_{1a}+N_{0a})(N_{1b}+N_{0b})}{N^2}}_{\text{constant}}\widetilde{DDP}_\mathcal{S}(\phi).$$

Therefore, the theoretical analysis about $\widetilde{DDP}_\mathcal{S}(\phi)$ in Proposition 1 and Theorem 1-2 in the main paper can be naturally extended to $\widehat{Cov}_\mathcal{S}(\phi)$. Just by substituting the equation above into Proposition 1 and Theorem 1-2, we restate them below:

**Proposition 1** (Restatement). *Define the magnitude of the signed distance by $D_\theta(\mathbf{x})$, i.e., $D_\theta(\mathbf{x})=|d_\theta(\mathbf{x})|$. Let $\phi(x)=x$ and $S_{1a}=\sum_{\mathcal{N}_{1a}}\phi(D_\theta(\mathbf{x}_i)),T_{1a}=\sum_{\mathcal{N}_{1a}}(\phi(D_\theta(\mathbf{x}_i))-1)$ and it is similar for $\mathcal{N}_{1b},\mathcal{N}_{0a},\mathcal{N}_{0b}$. It satisfies:*

$$\widehat{DDP}_\mathcal{S}=\frac{N^2}{2(N_{1a}+N_{0a})(N_{1b}+N_{0b})}\widehat{Cov}_\mathcal{S}(\phi)-\underbrace{\left(\frac{T_{1a}-S_{0a}}{N_{1a}+N_{0a}}-\frac{T_{1b}-S_{0b}}{N_{1b}+N_{0b}}\right)}_{\textit{surrogate-fairness gap}}. \tag{16}$$

**Theorem 1** (Restatement). *We assume that $G(D_\theta(\mathbf{x}))\in[1-\gamma,1]$, where $\gamma>0$. $\forall\epsilon>0$, if $\left|\widehat{Cov}(G)\right|\leq\epsilon$, then it holds:*

$$\left|\widehat{DDP}_\mathcal{S}\right|\leq\frac{N^2}{4(N_{1a}+N_{0a})(N_{1b}+N_{0b})}\epsilon+\gamma.$$

**Theorem 2** (Restatement). *We assume that $k$ points satisfy $G(D_\theta(\mathbf{x}))\in[0,1-\gamma]$ and others satisfy $G(D_\theta(\mathbf{x}))\in[1-\gamma,1]$, where $\gamma>0$. $\forall\epsilon>0$, if $\left|\widehat{Cov}(G)\right|\leq\epsilon$, then it holds:*

$$\left|\widehat{DDP}_\mathcal{S}\right|\leq\frac{N^2}{4(N_{1a}+N_{0a})(N_{1b}+N_{0b})}\epsilon+\gamma+\underbrace{\frac{1}{2}\left(\frac{1}{N_{1a}+N_{0a}}+\frac{1}{N_{1b}+N_{0b}}\right)k}_{\textit{relaxation factor}}.$$

The restatement of them does not affect the analysis in the main paper since $\widehat{DDP}_\mathcal{S}(\phi) \propto \widehat{Cov}_\mathcal{S}(\phi)$.

## A.3   Proof of Theorem 1

**Proof sketch.**  Firstly, absolute value inequality is used to derive the upper bound of $\left|\widehat{DDP}_\mathcal{S}\right|$ given the upper bound of $\left|\widetilde{DDP}_\mathcal{S}(\phi)\right|$. Secondly, the assumption $G(D_\theta(\mathbf{x})) \in [1-\gamma, 1]$ is used to derive a tighter bound of $\left|\widehat{DDP}_\mathcal{S}\right|$ to complete the proof.

*Proof.* If $\phi$ passes through the origin and increases monotonically, then according to the sign of $d_\theta(\mathbf{x})$, we have

$$
\begin{aligned}
\widetilde{DDP}_\mathcal{S}(\phi) &= \frac{\sum_{\mathcal{N}_{1a},\mathcal{N}_{0a}} \phi(d_\theta(\mathbf{x}_i))}{N_{1a} + N_{0a}} - \frac{\sum_{\mathcal{N}_{1b},\mathcal{N}_{0b}} \phi(d_\theta(\mathbf{x}_i))}{N_{1b} + N_{0b}} \\
&= \frac{\sum_{\mathcal{N}_{1a}} \phi(D_\theta(\mathbf{x}_i)) - \sum_{\mathcal{N}_{0a}} \phi(D_\theta(\mathbf{x}_i))}{N_{1a} + N_{0a}} - \frac{\sum_{\mathcal{N}_{1b}} \phi(D_\theta(\mathbf{x}_i)) - \sum_{\mathcal{N}_{0b}} \phi(D_\theta(\mathbf{x}_i))}{N_{1b} + N_{0b}}.
\end{aligned}
\tag{17}
$$

We define $T_{1a} = \sum_{\mathcal{N}_{1a}} \phi(D_\theta(\mathbf{x}_i)) - N_{1a} = \sum_{\mathcal{N}_{1a}}(\phi(D_\theta(\mathbf{x}_i)) - 1)$. It is similar for $T_{1b}, T_{0a}, T_{0b}$. We now analyze the term in (17) by limiting it with a parameter $\delta > 0$:

$$
\left|\widetilde{DDP}_\mathcal{S}(\phi)\right| = \left| \frac{\sum_{\mathcal{N}_{1a}} \phi(D_\theta(\mathbf{x}_i)) - \sum_{\mathcal{N}_{0a}} \phi(D_\theta(\mathbf{x}_i))}{N_{1a} + N_{0a}} - \frac{\sum_{\mathcal{N}_{1b}} \phi(D_\theta(\mathbf{x}_i)) - \sum_{\mathcal{N}_{0b}} \phi(D_\theta(\mathbf{x}_i))}{N_{1b} + N_{0b}} \right| \le \delta
\tag{18}
$$

$$
\Rightarrow \left| \frac{N_{1a} - N_{0a}}{N_{1a} + N_{0a}} - \frac{N_{1b} - N_{0b}}{N_{1b} + N_{0b}} + \frac{N_{0a} - N_{1a} + \sum_{\mathcal{N}_{1a}} \phi(D_\theta(\mathbf{x}_i)) - \sum_{\mathcal{N}_{0a}} \phi(D_\theta(\mathbf{x}_i))}{N_{1a} + N_{0a}} + \right.
$$

$$
\left. \frac{N_{1b} - N_{0b} - \sum_{\mathcal{N}_{1b}} \phi(D_\theta(\mathbf{x}_i)) + \sum_{\mathcal{N}_{0b}} \phi(D_\theta(\mathbf{x}_i))}{N_{1b} + N_{0b}} \right| \le \delta
$$

$$
\Rightarrow \left| \frac{N_{1a} - N_{0a}}{N_{1a} + N_{0a}} - \frac{N_{1b} - N_{0b}}{N_{1b} + N_{0b}} \right|
$$

$$
\le \delta + \left| \frac{N_{0a} - N_{1a} + \sum_{\mathcal{N}_{1a}} \phi(D_\theta(\mathbf{x}_i)) - \sum_{\mathcal{N}_{0a}} \phi(D_\theta(\mathbf{x}_i))}{N_{1a} + N_{0a}} + \right.
$$

$$
\left. \frac{N_{1b} - N_{0b} - \sum_{\mathcal{N}_{1b}} \phi(D_\theta(\mathbf{x}_i)) + \sum_{\mathcal{N}_{0b}} \phi(D_\theta(\mathbf{x}_i))}{N_{1b} + N_{0b}} \right|
$$

$$
\Rightarrow \left| \frac{N_{1a} - N_{0a}}{N_{1a} + N_{0a}} - \frac{N_{1b} - N_{0b}}{N_{1b} + N_{0b}} \right|
$$

$$
\le \delta + \left| \frac{(\sum_{\mathcal{N}_{1a}} \phi(D_\theta(\mathbf{x}_i)) - N_{1a}) - (\sum_{\mathcal{N}_{0a}} \phi(D_\theta(\mathbf{x}_i)) - N_{0a})}{N_{1a} + N_{0a}} + \right.
$$

$$
\left. \frac{(\sum_{\mathcal{N}_{0b}} \phi(D_\theta(\mathbf{x}_i)) - N_{0b}) - (\sum_{\mathcal{N}_{1b}} \phi(D_\theta(\mathbf{x}_i)) - N_{1b})}{N_{1b} + N_{0b}} \right|
$$

$$
\Rightarrow \left| \frac{N_{1a} - N_{0a}}{N_{1a} + N_{0a}} - \frac{N_{1b} - N_{0b}}{N_{1b} + N_{0b}} \right| \le \delta + \left| \frac{T_{1a} - T_{0a}}{N_{1a} + N_{0a}} + \frac{T_{0b} - T_{1b}}{N_{1b} + N_{0b}} \right|
$$

$$
\Rightarrow \left| \frac{N_{1a}}{N_{1a} + N_{0a}} - \left(1 - \frac{N_{1a}}{N_{1a} + N_{0a}}\right) - \frac{N_{1b}}{N_{1b} + N_{0b}} + \left(1 - \frac{N_{1b}}{N_{1b} + N_{0b}}\right) \right|
$$

$$
\le \delta + \left| \frac{T_{1a} - T_{0a}}{N_{1a} + N_{0a}} + \frac{T_{0b} - T_{1b}}{N_{1b} + N_{0b}} \right|
$$

$$
\Rightarrow \left|\widehat{DDP}_\mathcal{S}\right| = \left| \frac{N_{1a}}{N_{1a} + N_{0a}} - \frac{N_{1b}}{N_{1b} + N_{0b}} \right| \le \frac{1}{2}\left(\delta + \left| \frac{T_{1a} - T_{0a}}{N_{1a} + N_{0a}} + \frac{T_{0b} - T_{1b}}{N_{1b} + N_{0b}} \right|\right).
\tag{19}
$$

The process of mathematical derivation for (19) mainly includes basic algebraic operations, equivalent substitution and absolute value inequalities. The inequality (19) will be used later to prove (20).

Now we begin to use the assumption $G(D_\theta(\mathbf{x})) \in [1 - \gamma, 1]$. A relaxed version of such assumption is $G(D_\theta(\mathbf{x})) \in [1 - \gamma, 1 + \gamma]$. If the theorem holds with the relaxed version, then it also holds for the original assumption. We replace the above $\phi$ with general sigmoid $G$.

With $G(D_\theta(\mathbf{x})) \in [1 - \gamma, 1 + \gamma]$, $T_{1a} \in [-N_{1a}\gamma, N_{1a}\gamma]$ and $T_{0a} \in [-N_{0a}\gamma, N_{0a}\gamma]$ hold. In other words, $\frac{T_{1a} - T_{0a}}{N_{1a} + N_{0a}} \in [-\gamma, \gamma]$ and $\frac{T_{0b} - T_{1b}}{N_{1b} + N_{0b}} \in [-\gamma, \gamma]$, which means that:

$$\left| \frac{T_{1a} - T_{0a}}{N_{1a} + N_{0a}} + \frac{T_{0b} - T_{1b}}{N_{1b} + N_{0b}} \right| \in [0, 2\gamma].$$

So with (19), we have

$$\left| \widehat{DDP}_\mathcal{S} \right| = \left| \frac{N_{1a}}{N_{1a} + N_{0a}} - \frac{N_{1b}}{N_{1b} + N_{0b}} \right| \leq \frac{1}{2} \left( \delta + \left| \frac{T_{1a} - T_{0a}}{N_{1a} + N_{0a}} + \frac{T_{0b} - T_{1b}}{N_{1b} + N_{0b}} \right| \right) \leq \frac{1}{2}\delta + \gamma. \tag{20}$$

Now we set $\delta = \epsilon$, combine (20) with (18), and obtain the conclusion: $\forall \epsilon > 0$, if $\left| \widetilde{DDP}_\mathcal{S}(G) \right| \leq \epsilon$, then we have

$$\left| \widehat{DDP}_\mathcal{S} \right| \leq \frac{1}{2}\epsilon + \gamma.$$

The proof is complete. $\qquad\square$

### A.4 Proof of Theorem 2

**Proof sketch.** The difference between Theorem 1 and Theorem 2 mainly lies in the assumption of $D_\theta(\mathbf{x})$. Therefore, Equation (19) can be also used here because it is independent with the assumption. Consequently, the task of the proof is providing a tighter bound than Equation (19). The process of derivation mainly includes basic algebraic operations.

*Proof.* The number of points satisfying $\phi(D_\theta(\mathbf{x})) \in [0, 1 - \gamma]$ in the four groups are denoted as $U_{1a}, U_{1b}, U_{0a}, U_{0b}$, respectively. So there holds $\phi(D_\theta(\mathbf{x})) - 1 \in [-1, -\gamma]$ for them and $\phi(D_\theta(\mathbf{x})) - 1 \in [-\gamma, 0]$ for others. Note that

$$U_{1a} + U_{1b} + U_{0a} + U_{0b} = k.$$

We have

$$T_{1a} = \sum_{\mathcal{N}_{1a}} (\phi(D_\theta(\mathbf{x}) - 1)) \in [-U_{1a} - (N_{1a} - U_{1a})\gamma, -U_{1a}\gamma],$$

and it is similar for $T_{1b}, T_{0a}, T_{0b}$. So

$$\frac{T_{1a} - T_{0a}}{N_{1a} + N_{0a}} \in \left[ \frac{-N_{1a}\gamma - U_{1a} + (U_{1a} + U_{0a})\gamma}{N_{1a} + N_{0a}}, \frac{N_{0a}\gamma + U_{0a} - (U_{1a} + U_{0a})\gamma}{N_{1a} + N_{0a}} \right].$$

For the lower bound:

$$\begin{aligned} \frac{-N_{1a}\gamma - U_{1a} + (U_{1a} + U_{0a})\gamma}{N_{1a} + N_{0a}} &= \frac{-N_{1a}\gamma}{N_{1a} + N_{0a}} + \frac{(U_{1a} + U_{0a})\gamma - U_{1a}}{N_{1a} + N_{0a}} \\ &\geq -\gamma + \frac{(U_{1a} + U_{0a})\gamma - U_{1a}}{N_{1a} + N_{0a}} \\ &\geq -\gamma - \frac{U_{1a}}{N_{1a} + N_{0a}} \\ &\geq -\gamma - \frac{k}{N_{1a} + N_{0a}}. \end{aligned} \tag{21}$$

For the upper bound:

$$
\begin{aligned}
\frac{N_{0a}\gamma + U_{0a} - (U_{1a} + U_{0a})\gamma}{N_{1a} + N_{0a}} &= \frac{N_{0a}\gamma}{N_{1a} + N_{0a}} + \frac{U_{0a} - (U_{1a} + U_{0a})\gamma}{N_{1a} + N_{0a}} \\
&\leq \gamma + \frac{U_{0a} - (U_{1a} + U_{0a})\gamma}{N_{1a} + N_{0a}} \\
&\leq \gamma + \frac{U_{0a}}{N_{1a} + N_{0a}} \\
&\leq \gamma + \frac{k}{N_{1a} + N_{0a}}.
\end{aligned}
\tag{22}
$$

Therefore, we have

$$
\frac{T_{1a} - T_{0a}}{N_{1a} + N_{0a}} \in \left[ -\gamma - \frac{k}{N_{1a} + N_{0a}}, \gamma + \frac{k}{N_{1a} + N_{0a}} \right],
$$

and

$$
\frac{T_{0b} - T_{1b}}{N_{1b} + N_{0b}} \in \left[ -\gamma - \frac{k}{N_{1b} + N_{0b}}, \gamma + \frac{k}{N_{1b} + N_{0b}} \right].
$$

So

$$
\left| \frac{T_{1a} - T_{0a}}{N_{1a} + N_{0a}} + \frac{T_{0b} - T_{1b}}{N_{1b} + N_{0b}} \right| \leq 2\gamma + \left( \frac{1}{N_{1a} + N_{0a}} + \frac{1}{N_{1b} + N_{0b}} \right) k.
\tag{23}
$$

The last step is similar to Appendix A.3, but a relaxation term is added. Combining (19) and (23) together, we have

$$
\left| \widehat{DDP}_{\mathcal{S}} \right| \leq \frac{1}{2}\epsilon + \gamma + \frac{1}{2} \left( \frac{1}{N_{1a} + N_{0a}} + \frac{1}{N_{1b} + N_{0b}} \right) k.
$$

The proof is complete.

$\square$

## A.5  Risk Bound

We show the risk bound of $DDP$, and then provide some discussion about the surrogate function in optimization.

**Theorem 3.** *For a given constant $\delta > 0$, there holds:*

$$
P\left( \left| \widehat{DDP}_{\mathcal{S}} - DDP \right| \leq t \right) \geq 1 - \delta,
$$

*where $t = \sqrt{\frac{1}{2} \ln \frac{2}{\delta}} \cdot \sqrt{\frac{N}{(N-1)^2}}$.*

*Proof.* We know that

$$
\begin{aligned}
\mathbb{E}\left( \widehat{DDP}_{\mathcal{S}} \right) &= \mathbb{E}\left( \frac{\sum_{\mathcal{N}_{1a}, \mathcal{N}_{0a}} \mathbb{1}_{d_\theta(\mathbf{x}) > 0}}{N_{1a} + N_{0a}} - \frac{\sum_{\mathcal{N}_{1b}, \mathcal{N}_{0b}} \mathbb{1}_{d_\theta(\mathbf{x}) > 0}}{N_{1b} + N_{0b}} \right) \\
&= \frac{\sum_{\mathcal{N}_{1a}, \mathcal{N}_{0a}} \mathbb{E}(\mathbb{1}_{d_\theta(\mathbf{x}) > 0})}{N_{1a} + N_{0a}} - \frac{\sum_{\mathcal{N}_{1b}, \mathcal{N}_{0b}} \mathbb{E}(\mathbb{1}_{d_\theta(\mathbf{x}) > 0})}{N_{1b} + N_{0b}} \\
&= \frac{\sum_{\mathcal{N}_{1a}, \mathcal{N}_{0a}} \mathbb{E}(\mathbb{1}_{d_\theta(\mathbf{x}) > 0 | z = +1})}{N_{1a} + N_{0a}} - \frac{\sum_{\mathcal{N}_{1b}, \mathcal{N}_{0b}} \mathbb{E}(\mathbb{1}_{d_\theta(\mathbf{x}) > 0 | z = -1})}{N_{1b} + N_{0b}} \\
&= \mathbb{E}(\mathbb{1}_{d_\theta(\mathbf{x}) > 0 | z = +1}) - \mathbb{E}(\mathbb{1}_{d_\theta(\mathbf{x}) > 0 | z = -1}) \\
&= P(d_\theta(\mathbf{x}) > 0 | z = +1) - P(d_\theta(\mathbf{x}) > 0 | z = -1) \\
&= DDP.
\end{aligned}
$$

Since the indicator function satisfies $\mathbb{1}_{[\cdot]} \in [0,1]$, so according to the Hoeffding inequality,

$$P(\left|\widehat{DDP}_{\mathcal{S}} - DDP\right| \leq t) \geq 1 - 2\exp\left(-2t^2 N\right).$$

We set $\delta = 2\exp\left(-2t^2 N\right)$ so that $t = \sqrt{\frac{1}{2}\ln\frac{2}{\delta}} \cdot \sqrt{\frac{1}{N}} = O(\frac{1}{\sqrt{N}})$. Substitute them into Theorem 3 above, we can prove the result. $\qquad\square$

**Fairness guarantee.** From the above theorem, one can see that, with a probability more than $1 - \delta$, the discrepancy between the $\widehat{DDP}_{\mathcal{S}}$ and $DDP$ can be small enough with the order $\mathcal{O}(\frac{1}{\sqrt{N}})$. Also, Proposition 1 and Theorem 3 together tell us that with a big dataset (N is large), if $\widetilde{\widehat{DDP}}_{\mathcal{S}}(\phi)$ and the gap are both small enough, then we can state with a high probability that the classifier satisfies the fairness constraint.

Recall that $\widetilde{\widehat{DDP}}_{\mathcal{S}}(\phi)$ is used for optimization, $DDP$ is the real fairness criterion, and $\widehat{DDP}_{\mathcal{S}}$ is an estimator of $DDP$. Therefore, it is crucial to theoretically build a bound between the practice $(\widetilde{\widehat{DDP}}_{\mathcal{S}}(\phi))$ and the objective $(DDP)$. To address this issue, note that:

$$\left|\widetilde{\widehat{DDP}}_{\mathcal{S}}(\phi) - DDP\right| \leq \underbrace{\left|\widetilde{\widehat{DDP}}_{\mathcal{S}}(\phi) - \widehat{DDP}_{\mathcal{S}}\right|}_{\text{surrogate-fairness gap}} + \underbrace{\left|\widehat{DDP}_{\mathcal{S}} - DDP\right|}_{\text{risk bound of DDP}}.$$

Therefore, if we have an very large dataset (a small risk bound of $DDP$) and an appropriate surrogate function with fairness guarantee (a bounded surrogate-fairness gap), then there is theoretical guarantee for $\left|\widetilde{\widehat{DDP}}_{\mathcal{S}}(\phi) - DDP\right|$.

**Is infinitesimal $\widehat{DDP}_{\mathcal{S}}$ better?** However, although the discrepancy between the $\widehat{DDP}_{\mathcal{S}}$ and $DDP$ can be small enough, the discrepancy still exists, which means that a classifier with $\widehat{DDP}_{\mathcal{S}} \approx 0$ may not satisfy $DDP \approx 0$. The principle is similar for ERM (Empirical Risk Minimization): a small enough empirical loss may lead to overfitting so that an infinitesimal loss is not always better. Therefore, an infinitesimal $\widehat{DDP}_{\mathcal{S}}$ is also not always better.

**The surrogate loss function in machine learning.** In machine learning, a surrogate loss function is an auxiliary function used during optimization, which is easier to optimize than the original objective function. Some examples of surrogate loss functions are in Figure 6. Surrogate loss functions are particularly useful when dealing with complex or non-differentiable objective functions. The idea is to approximate the difficult parts of the objective with a surrogate that has more tractable properties (like smoothness or convexity) for optimization algorithms. Here are some examples:

- Surrogate loss functions are often smooth and convex, even if the original objective is not. This allows for the use of gradient-based optimization techniques, which can speed up the learning process.

- Use in various algorithms: Surrogate loss functions are used in various machine learning algorithms, including support vector machines (where the hinge loss function acts as a surrogate for the 0-1 loss function), boosting methods (which build a surrogate to focus on examples that are hard to predict).

- Loss approximation: In classification, surrogate loss functions are used to approximate the original loss function. For example, logistic loss and hinge loss are surrogates for the 0-1 loss function in logistic regression and support vector machines, respectively.

**Surrogate loss functions and fairness surrogate functions.** The foundation of fairness surrogate function is similar to the surrogate loss function: both of them aim to approximate the 0-1 function. When minimizing the loss function, there are two kinds of insights to choose surrogate functions:

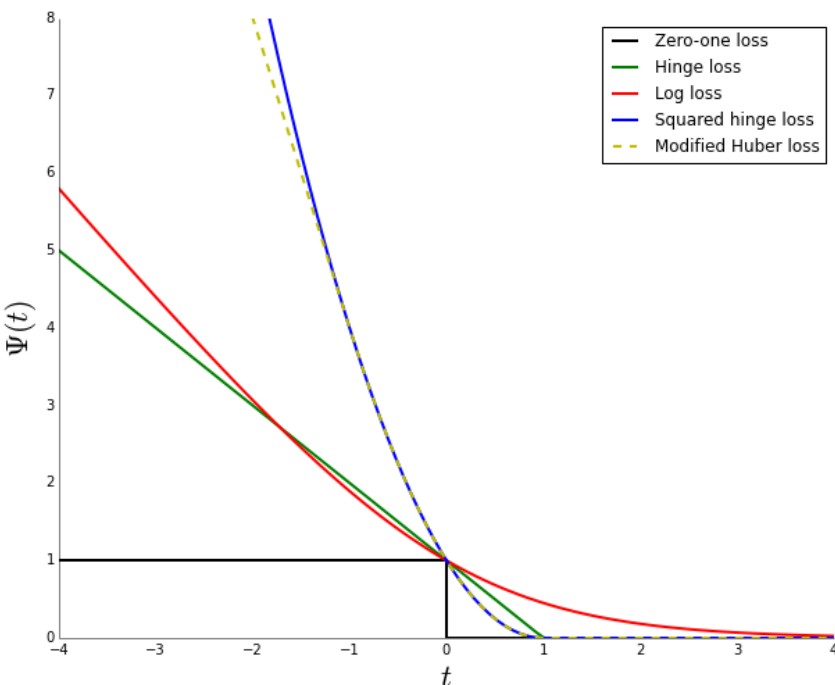

Figure 6: The surrogate loss functions $\Psi$ in machine learning. The figure is adapted from this link.

- **Upper bound of 0-1 function**: Taking surrogate loss functions as an example. As suggested in Figure 6, all of these surrogate loss functions are upper bounds of 0-1 function. Therefore, when the loss function is minimized to a small value, there is generalization guarantee for the classifier (Mohri et al., 2018). Unfortunately, as suggested in Figure 1, the fairness surrogate functions mentioned in existing work are not upper bounds of 0-1 function.

- **Estimator of 0-1 function**: As shown in Figure 1, we aim to better estimate the 0-1 function for fairness. Our empirical results show that our general sigmoid estimation performs fairer results.

Overall, we believe that the fairness community can be inspired by machine learning and optimization. How to design better fairness surrogate functions is intriguing. Insights from surrogate loss functions may be promising in the fairness community for future work.

### A.6 Variance Analysis

**Theorem 4.** *Let* $N_a = N_{1a} + N_{0a}, N_b = N_{1b} + N_{0b}, P_a = P(d_\theta(\mathbf{x}) > 0|z = +1), P_b = P(d_\theta(\mathbf{x}) > 0|z = -1).$ *We assume that each dataset consisting of $N$ points is independently drawn from a dataset distribution. The variance is computed over the dataset distribution. Then there holds:*

$$Var\left(\widehat{DDP}_{\mathcal{S}}\right) \leq \frac{1}{4}\left(\frac{1}{N_a} + \frac{1}{N_b}\right). \tag{24}$$

*Proof.* Using the equation $Var(x) = \mathbb{E}(x^2) - (\mathbb{E}x)^2$, we have

$$Var\left(\widehat{DDP}_{\mathcal{S}}\right) = Var\left(\frac{\sum_{\mathcal{N}_{1a},\mathcal{N}_{0a}} \mathbb{1}_{d_\theta(\mathbf{x})>0}}{N_{1a} + N_{0a}} - \frac{\sum_{\mathcal{N}_{1b},\mathcal{N}_{0b}} \mathbb{1}_{d_\theta(\mathbf{x})>0}}{N_{1b} + N_{0b}}\right)$$

$$= \mathbb{E} \underbrace{\left( \frac{\sum_{\mathcal{N}_{1a},\mathcal{N}_{0a}} \mathbb{1}_{d_\theta(\mathbf{x})>0}}{N_{1a} + N_{0a}} - \frac{\sum_{\mathcal{N}_{1b},\mathcal{N}_{0b}} \mathbb{1}_{d_\theta(\mathbf{x})>0}}{N_{1b} + N_{0b}} \right)^2}_{T} - \left[ \underbrace{\mathbb{E}(\widehat{DDP}_\mathcal{S})}_{=DDP} \right]^2 \tag{25}$$

where

$$T = \mathbb{E} \left\{ \underbrace{\left[ \frac{\sum_{\mathcal{N}_{1a},\mathcal{N}_{0a}} \mathbb{1}_{d_\theta(\mathbf{x})>0}}{N_{1a} + N_{0a}} \right]^2}_{T_1} + \underbrace{\left[ \frac{\sum_{\mathcal{N}_{1b},\mathcal{N}_{0b}} \mathbb{1}_{d_\theta(\mathbf{x})>0}}{N_{1b} + N_{0b}} \right]^2}_{T_2} - \underbrace{\frac{2(\sum_{\mathcal{N}_{1a},\mathcal{N}_{0a}} \mathbb{1}_{d_\theta(\mathbf{x})>0})(\sum_{\mathcal{N}_{1b},\mathcal{N}_{0b}} \mathbb{1}_{d_\theta(\mathbf{x})>0})}{(N_{1a} + N_{0a})(N_{1b} + N_{0b})}}_{T_3} \right\}$$

With the linearity of expectation, we deal with the three terms in $T$ respectively. First of all,

$$\mathbb{E}(T_1) = \frac{1}{(N_{1a} + N_{0a})^2} \mathbb{E} \left[ \sum_{\mathcal{N}_{1a},\mathcal{N}_{0a}} \mathbb{1}_{d_\theta(\mathbf{x})>0} \right]^2 \tag{26}$$

and

$$\mathbb{E} \left[ \sum_{\mathcal{N}_{1a},\mathcal{N}_{0a}} \mathbb{1}_{d_\theta(\mathbf{x})>0} \right]^2$$

$$= Var \left( \sum_{\mathcal{N}_{1a},\mathcal{N}_{0a}} \mathbb{1}_{d_\theta(\mathbf{x})>0} \right) + \left( \mathbb{E} \sum_{\mathcal{N}_{1a},\mathcal{N}_{0a}} \mathbb{1}_{d_\theta(\mathbf{x})>0} \right)^2$$

$$= \sum_{\mathcal{N}_{1a},\mathcal{N}_{0a}} Var \left( \mathbb{1}_{d_\theta(\mathbf{x})>0} \right) + \left( \sum_{\mathcal{N}_{1a},\mathcal{N}_{0a}} \mathbb{E} \left( \mathbb{1}_{d_\theta(\mathbf{x})>0} \right) \right)^2$$

$$= \sum_{\mathcal{N}_{1a},\mathcal{N}_{0a}} Var \left( \mathbb{1}_{d_\theta(\mathbf{x})>0|z=+1} \right) + \left( \sum_{\mathcal{N}_{1a},\mathcal{N}_{0a}} \mathbb{E} \left( \mathbb{1}_{d_\theta(\mathbf{x})>0|z=+1} \right) \right)^2$$

$$= (N_{1a} + N_{0a}) Var \left( \mathbb{1}_{d_\theta(\mathbf{x})>0|z=+1} \right) + (N_{1a} + N_{0a})^2 \left[ P(d_\theta(\mathbf{x}) > 0|z = +1) \right]^2 \tag{27}$$

Also,

$$Var \left( \mathbb{1}_{d_\theta(\mathbf{x})>0|z=+1} \right) = \mathbb{E}(\mathbb{1}_{d_\theta(\mathbf{x})>0|z=+1})^2 - (\mathbb{E}(\mathbb{1}_{d_\theta(\mathbf{x})>0|z=+1}))^2$$
$$= \mathbb{E}(\mathbb{1}_{d_\theta(\mathbf{x})>0|z=+1}) - (P(d_\theta(\mathbf{x}) > 0|z = +1))^2$$
$$= P(d_\theta(\mathbf{x}) > 0|z = +1) - (P(d_\theta(\mathbf{x}) > 0|z = +1))^2 \tag{28}$$

Combining (26), (27) and (28), we have

$$\mathbb{E}(T_1) = \left( 1 - \frac{1}{N_{1a} + N_{0a}} \right) (P(d_\theta(\mathbf{x}) > 0|z = +1))^2 + \frac{1}{N_{1a} + N_{0a}} P(d_\theta(\mathbf{x}) > 0|z = +1). \tag{29}$$

Similarly, we have

$$\mathbb{E}(T_2) = \left( 1 - \frac{1}{N_{1b} + N_{0b}} \right) (P(d_\theta(\mathbf{x}) > 0|z = -1))^2 + \frac{1}{N_{1b} + N_{0b}} P(d_\theta(\mathbf{x}) > 0|z = -1). \tag{30}$$

Finally, because of the fact that

$$\mathbb{E} \left( \sum_{\mathcal{N}_{1a},\mathcal{N}_{0a}} \mathbb{1}_{d_\theta(\mathbf{x})>0} \quad \cdot \sum_{\mathcal{N}_{1b},\mathcal{N}_{0b}} \mathbb{1}_{d_\theta(\mathbf{x})>0} \right)$$

$$=\mathbb{E}\left(\sum_{\mathcal{N}_{1a},\mathcal{N}_{0a}}\mathbb{1}_{d_\theta(\mathbf{x})>0}\right)\cdot\mathbb{E}\left(\sum_{\mathcal{N}_{1b},\mathcal{N}_{0b}}\mathbb{1}_{d_\theta(\mathbf{x})>0}\right)+Cov\left(\sum_{\mathcal{N}_{1a},\mathcal{N}_{0a}}\mathbb{1}_{d_\theta(\mathbf{x})>0}\quad,\quad\sum_{\mathcal{N}_{1b},\mathcal{N}_{0b}}\mathbb{1}_{d_\theta(\mathbf{x})>0}\right)$$

$$=(N_{1a}+N_{0a})P(d_\theta(\mathbf{x})>0|z=+1)\cdot(N_{1b}+N_{0b})P(d_\theta(\mathbf{x})>0|z=-1)+$$

$$\sum_{\mathcal{N}_{1a},\mathcal{N}_{0a}}\sum_{\mathcal{N}_{1b},\mathcal{N}_{0b}}\underbrace{Cov\left(\mathbb{1}_{d_\theta(\mathbf{x})>0|z=+1},\mathbb{1}_{d_\theta(\mathbf{x})>0|z=-1}\right)}_{0}$$

$$=(N_{1a}+N_{0a})P(d_\theta(\mathbf{x})>0|z=+1)\cdot(N_{1b}+N_{0b})P(d_\theta(\mathbf{x})>0|z=-1),$$

we have

$$\mathbb{E}(T_3)=\mathbb{E}\left[\frac{2(\sum_{\mathcal{N}_{1a},\mathcal{N}_{0a}}\mathbb{1}_{d_\theta(\mathbf{x})>0})(\sum_{\mathcal{N}_{1b},\mathcal{N}_{0b}}\mathbb{1}_{d_\theta(\mathbf{x})>0})}{(N_{1a}+N_{0a})(N_{1b}+N_{0b})}\right]$$

$$=\frac{2(N_{1a}+N_{0a})P(d_\theta(\mathbf{x})>0|z=+1)\cdot(N_{1b}+N_{0b})P(d_\theta(\mathbf{x})>0|z=-1)}{(N_{1a}+N_{0a})(N_{1b}+N_{0b})}$$

$$=2P_aP_b. \tag{31}$$

Overall, combining (29),(30),(31) with (25), we have:

$$Var\left(\widehat{DDP}_{\mathcal{S}}\right)=\mathbb{E}(T_1)+\mathbb{E}(T_2)-\mathbb{E}(T_3)-(DDP)^2$$

$$=\underbrace{\left(1-\frac{1}{N_a}\right)P_a^2+\frac{1}{N_a}P_a}_{\mathbb{E}(T_1)}+\underbrace{\left(1-\frac{1}{N_b}\right)P_b^2+\frac{1}{N_b}P_b}_{\mathbb{E}(T_2)}-\underbrace{2P_aP_b}_{\mathbb{E}(T_3)}-\underbrace{(P_a-P_b)^2}_{DDP^2}$$

$$=-\frac{1}{N_a}P_a^2-\frac{1}{N_b}P_b^2+\frac{1}{N_a}P_a+\frac{1}{N_b}P_b$$

$$=\frac{1}{4}\left(\frac{1}{N_a}+\frac{1}{N_b}\right)-\frac{1}{N_a}(P_a-\frac{1}{2})^2-\frac{1}{N_b}(P_b-\frac{1}{2})^2 \qquad (P_a+P_b=1)$$

$$=\frac{1}{4}\left(\frac{1}{N_a}+\frac{1}{N_b}\right)-\left(\frac{1}{N_a}+\frac{1}{N_b}\right)(P_a-\frac{1}{2})^2 \tag{32}$$

$$<1$$

For the maximum variance, $P_a$ takes the value $\frac{1}{2}$, and the maximum variance is

$$\max\left[Var\left(\widehat{DDP}_{\mathcal{S}}\right)\right]=\frac{1}{4}\left(\frac{1}{N_a}+\frac{1}{N_b}\right). \tag{33}$$

When $P_a=0$ or $P_a=1$, there holds $Var\left(\widehat{DDP}_{\mathcal{S}}\right)=0$. Furthermore, the result also provides insights into the advantages of a *balanced dataset* for stability. $\square$

We place the computation of $Var\left(\widehat{DDP}_{\mathcal{S}}\right)$ for the convenience of researchers interested in exploration. For example, designing alternative surrogate functions and algorithms based on these expressions to reduce variance could be an intriguing avenue for research.

## B  Further Experimental Details and Results

### B.1  The Balanced Surrogates Algorithm

The process of balanced surrogates is shown in Algorithm 1. Computations associated with $\lambda_t$ cost $\mathcal{O}(N)$. Consider solving (8) gives rise to $\mathcal{O}(M)$ computations. So Algorithm 1 costs $\mathcal{O}(t(M+N))$.

---

**Algorithm 1: Balanced Surrogates**

---

1: **Input:** Training set $\{(\mathbf{x}_i, y_i)\}_{i=1}^N$.
2: **Initialization:** Balance factor $\lambda$, surrogate function $\phi_1$, termination threshold $\eta$, maximum iterations $\tau$
    and other parameters.
3: Train an unconstrained classifier and obtain the model parameter $\theta_0$.
4: Time step $t = 0$.
5: **repeat**
6:      $t = t + 1$.
7:      Solve the optimization problem (8) (start at $\theta_0$).
8:      Compute $d_\theta(\mathbf{x}_i), i = 1, \cdots, N$ and $N_{1a}, N_{1b}, N_{0a}, N_{0b}$.
9:      Obtain $\lambda'_t$ via (13).
10:     Obtain $\lambda_t$ by exponential smoothing via (14).
11:     **if** $\lambda_t \leq 0$ **then**
12:         Set $\lambda_t = 1$.
13:     **end if**
14: **until** $|\lambda_t - \lambda_{t-1}| \leq \eta$ **or** $t = \tau$ **or** $\lambda_t \leq 0$
15: **Output:** $\lambda_t, d_\theta(\mathbf{x}_i), i = 1, \cdots, N$ and $N_{1a}, N_{1b}, N_{0a}, N_{0b}$.

---

**Discussion of motivation of this algorithm.** One may question the motivation of balanced surrogate that the learning algorithm will try to find a balance between the empirical loss and the fairness regularization, and it may suffice to use the upper bound of $\widehat{DDP}_\mathcal{S}$ as the fairness regularization. In order to clarify it, we provide more discussion of motivation of the balanced surrogate algorithm.

We believe that previous algorithms with fairness regularization aim to find a balance between the empirical loss and the fairness regularization. However, in this paper, we want to uncover the issue that when we use fairness surrogate functions as the constraint/regularization, there will be a surrogate-fairness gap between the constraint/regularization and the true fairness definition. Therefore, we emphasize that some widely used fairness surrogate function, such as the unbounded surrogates analyzed in this paper, may not serve as an appropriate regularization because of the surrogate-fairness gap.

Thus, focusing on fairness surrogate functions, we provide the balanced surrogate algorithm to reduce the gap, thus trying to help fairness surrogate functions to achieve a better fairness performance. In the experiments, the balanced surrogate method consistently helps the unbounded surrogates (log-sigmoid, linear, and hinge surrogates) to be fairer, while it benefits the bounded surrogates (sigmoid and the general sigmoid surrogate) in most cases. It suggests that reducing the gap is useful for unbounded surrogates. In this case, it may not suffice to use the unbounded upper bound of $\widehat{DDP}_\mathcal{S}$. However, when we use bounded surrogates, making the gap infinitesimal may not always be beneficial to fairness. In this case, such phenomenon may be in line with your concern. Overall, how to strike a balance between the gap and the surrogate function for better fairness is intriguing, but it may be outside the scope of this work.

### B.2 Details in Figure 2

The datasets are mentioned in Section 6. We randomly split each dataset into train (70%) and test (30%) set. Then for each dataset, we train two logistic regression classifiers, one is the unconstrained classifier and the other is the constrained optimization problem (Zafar et al., 2017c) with 0 covariance threshold.

### B.3 The Parameters

**General sigmoid surrogate.**

- The regularization coefficient $\lambda$ in (8) lies in the grid $[0.1, 0.2, \cdots, 5]$.

- The parameter $w$ in general sigmoid is chosen from $\{1, 2, 4, 8, 16\}$ according to the fairness performance on the validation set.

**Balanced surrogate.**

- The smoothing factor $\alpha = 0.9$.

- The termination threshold $\eta = 0.01$.

**Discussion of hyper-parameters in our algorithms.** Indeed, our approaches do require some additional hyper-parameter tuning:

- For $w$ in general sigmoid, as shown in our experimental settings in the appendix, it is selected from $\{1, 2, 4, 8, 16\}$ according to the fairness performance on the validation set. Thus, it requires selecting an approximate range for it.

- For the regularization coefficient $\gamma$, it is conventional in machine learning to adjust it. Our experiments also require selecting an approximate range for it.

- The smoothing factor $\alpha$, and termination threshold $\eta$ are hyper-parameters, but we fix them and never change them throughout our experiments.

- For $\gamma$ in the balanced surrogate approach, it does not require any hyper-parameter tuning because it is automatically specified in the algorithm.

Therefore, in our experiments, we only consider tuning two hyper-parameters: $w$ and $\gamma$. And hyper-parameter optimization tools (such as `hyperopt` [1] ) may be useful to aid the cost of hyper-parameter tuning.

### B.4 Upsampling and Downsampling

For Adult and COMPAS, the imbalance issue is not too severe (about 2:1 and 3:2, respectively), so we balance between different demographic groups by randomly selecting samples and form the new training dataset such that samples in each group are of *the same number*. However, for Bank Marketing, the ratio of two groups approaches 20:1, so the sampling scheme for Adult and COMPAS is not suitable because it will cause 19 copies of the minority groups (overfitting). As a result, we only incorporate 1 extra copy of the minority group into the training set when upsampling.

### B.5 Numerical Results of Figure 3-4

Full results are in Table 2. Standard deviation is shown for the metrics.

### B.6 Comparing with Other Fairness-aware Algorithms

In addition to CP (Zafar et al., 2017c) (denoted as 'Linear'), we compare with two other popular in-processing fairness-aware methods: reduction approach (Agarwal et al., 2018) (denoted as 'reduction') and adaptive sensitive reweighting (Krasanakis et al., 2018) (denoted as 'ASR'), which are not based on fairness surrogate functions. The results are shown in Figure 7.

- In the Bank dataset, our general sigmoid surrogate and balanced surrogate approaches consistently achieve much better fairness performance than 'reduction' and 'ASR' while maintaining comparable accuracy.

- In the COMPAS dataset, our approaches always achieve better fairness performance than others while maintaining comparable accuracy.

- In the Adult dataset, our methods achieve better fairness than 'ASR' while maintaining comparable accuracy.

---

[1]https://github.com/hyperopt/hyperopt

Table 2: Experimental results.

| Datasets | Methods | Accuracy | $\widehat{DDP}_{\mathcal{S}}$ |
|---|---|---|---|
| Adult | Unconstrained | $0.838 \pm 0.009$ | $0.182 \pm 0.018$ |
| | Linear | $0.810 \pm 0.031$ | $0.060 \pm 0.020$ |
| | B-Linear | $\mathbf{0.815 \pm 0.017}$ | $0.053 \pm 0.010$ |
| | Log-Sigmoid | $0.808 \pm 0.018$ | $0.052 \pm 0.025$ |
| | B-Log-Sigmoid | $0.801 \pm 0.018$ | $0.041 \pm 0.020$ |
| | Hinge | $0.808 \pm 0.018$ | $0.051 \pm 0.025$ |
| | B-Hinge | $0.803 \pm 0.016$ | $0.042 \pm 0.019$ |
| | Sigmoid | $0.802 \pm 0.015$ | $0.033 \pm 0.011$ |
| | B-Sigmoid | $0.802 \pm 0.012$ | $0.037 \pm 0.012$ |
| | General Sigmoid | $0.805 \pm 0.013$ | $\mathbf{0.029 \pm 0.010}$ |
| | B-General Sigmoid | $0.803 \pm 0.007$ | $0.037 \pm 0.016$ |
| Bank Marketing | Unconstrained | $0.910 \pm 0.004$ | $0.192 \pm 0.003$ |
| | Linear | $0.892 \pm 0.005$ | $0.037 \pm 0.041$ |
| | B-Linear | $0.891 \pm 0.005$ | $0.032 \pm 0.039$ |
| | Log-Sigmoid | $\mathbf{0.894 \pm 0.007}$ | $0.043 \pm 0.038$ |
| | B-Log-Sigmoid | $0.892 \pm 0.005$ | $0.040 \pm 0.037$ |
| | Hinge | $0.891 \pm 0.005$ | $0.035 \pm 0.037$ |
| | B-Hinge | $0.889 \pm 0.005$ | $0.029 \pm 0.046$ |
| | Sigmoid | $0.889 \pm 0.004$ | $0.021 \pm 0.044$ |
| | B-Sigmoid | $0.888 \pm 0.004$ | $0.024 \pm 0.042$ |
| | General Sigmoid | $0.889 \pm 0.003$ | $0.014 \pm 0.031$ |
| | B-General Sigmoid | $0.889 \pm 0.002$ | $\mathbf{0.013 \pm 0.027}$ |
| COMPAS | Unconstrained | $0.666 \pm 0.020$ | $0.281 \pm 0.035$ |
| | Linear | $\mathbf{0.636 \pm 0.009}$ | $0.039 \pm 0.022$ |
| | B-Linear | $0.626 \pm 0.009$ | $0.035 \pm 0.013$ |
| | Log-Sigmoid | $0.629 \pm 0.012$ | $0.039 \pm 0.021$ |
| | B-Log-Sigmoid | $0.626 \pm 0.010$ | $0.036 \pm 0.021$ |
| | Hinge | $0.625 \pm 0.009$ | $0.033 \pm 0.021$ |
| | B-Hinge | $0.624 \pm 0.008$ | $0.033 \pm 0.021$ |
| | Sigmoid | $0.629 \pm 0.006$ | $0.034 \pm 0.013$ |
| | B-Sigmoid | $0.626 \pm 0.009$ | $0.031 \pm 0.017$ |
| | General Sigmoid | $0.627 \pm 0.010$ | $0.030 \pm 0.018$ |
| | B-General Sigmoid | $0.626 \pm 0.009$ | $\mathbf{0.029 \pm 0.016}$ |

Overall, it suggests that our methods are very competitive among these in-processing methods. We leave the comparative study between our methods and other in-processing methods for future work. We hope that our supplementary experiment helps people better understand the advantages of our approach.

### B.7 Boxplots with General Sigmoid Surrogate

Boxplots for three datasets with general sigmoid surrogate are shown in Figure 8.

## C Other Related Work

### C.1 Fairness-aware Algorithms

Our paper focuses on fairness surrogate functions from the in-processing methods. In addition to the afore-mentioned fairness constraints (Zafar et al., 2017a;c) and fairness regularization (Bendekgey & Sudderth, 2021), there are also various kinds of fairness-aware in-processing methods in the community. For example,

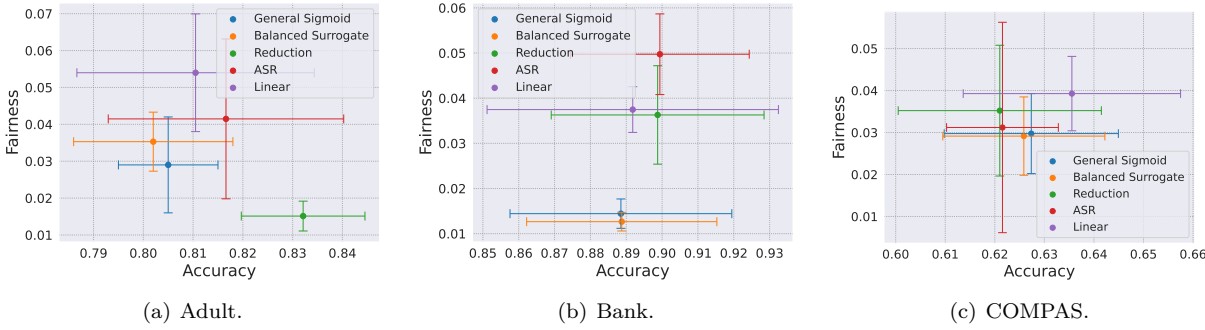

(a) Adult.

(b) Bank.

(c) COMPAS.

Figure 7: Results of other in-processing methods.

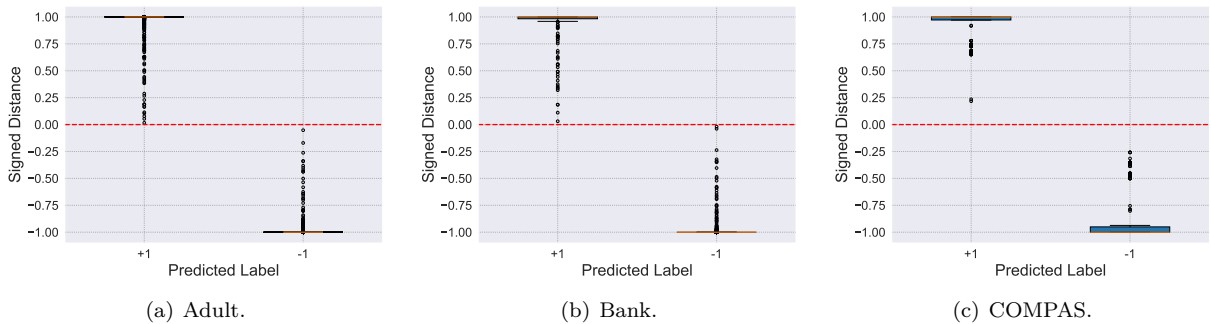

(a) Adult.

(b) Bank.

(c) COMPAS.

Figure 8: Boxplots for three datasets with general sigmoid surrogate. $+1$ and $-1$ represent the predicted label. The red dashed line means $d_\theta(\mathbf{x}) = 0$. The orange line in the box is the median. The median line and the edges of the boxplot almost overlap with the box itself.

fair adversarial learning (Zhang et al., 2018; Madras et al., 2018), fair reweighing (Krasanakis et al., 2018; Lahoti et al., 2020; Chai & Wang, 2022; Maheshwari & Perrot, 2023), fair sample selection (Roh et al., 2021a;b), and hyper-parameter optimization (Cruz et al., 2021; Tizpaz-Niari et al., 2022). In addition to DP, there are also fairness-aware algorithms for other fairness definitions, such as sufficiency (Shui et al., 2022a). And there are many other application scenarios, such as medical image analysis (Mehta et al., 2024; Shui et al., 2023).

To gain a comprehensive understanding of fairness-aware algorithms in machine learning, thorough surveys are conducted in Mehrabi et al. (2021); Caton & Haas (2020); Hort et al. (2022); Wan et al. (2023); Pessach & Shmueli (2023), providing a broader view of these algorithms.

## C.2 Fairness Surrogate Functions

For CP, even when the empirical covariance equals 0, there is no guarantee whether demographic parity is already satisfied (Zafar et al., 2019; Bendekgey & Sudderth, 2021; Wu et al., 2019; Lohaus et al., 2020). These papers give different counterexamples to explain it. Zafar et al. (2019) provides a simple example explaining that covariance constraints may perform unfavorably in the presence of large margin points. Bendekgey & Sudderth (2021) generates a synthetic dataset and observes that the decision boundary shifts dramatically when an extreme large margin point is added. Wu et al. (2019) provides two examples, respectively, to show that satisfying $\widehat{DDP}_\mathcal{S}$ and $\widehat{Cov}_\mathcal{S}(\phi)$ constraints are not necessarily related. Lohaus et al. (2020) draws heatmaps to illustrate that some popular surrogates do not manage to capture the violation of fairness reasonably. Radovanović et al. (2022) also conducts extensive results to show that optimizing the loss function given the fairness constraint or regularization for unfairness can surprisingly yield unfair solutions for Adult (Kohavi, 1996) and COMPAS (Julia Angwin & Kirchner, 2016) datasets. Instead of presenting the

counterexamples in previous work, we systematically point out the large margin points issue for covariance proxy as well as other unbounded surrogate functions with theoretical consideration.

# D   Extension to Other Fairness Definitions

To begin with, in Section D.1, we review the considered fairness definitions: disparate mistreatment, and balance for positive (negative) class. After that, in Section D.2, we provide a deeper understanding of CP for disparate mistreatment. The proof of the statements is in Section D.3. Finally, in Section D.4, we provide a deeper understanding of CP for balance for positive (negative) class. The corresponding proof is in Section D.5.

Since the only difference between the linear surrogate and the CP is a dateset-specific constant, the results in this paper about CP is also applicable for linear surrogate. As a result, by just replacing the linear surrogate function with other surrogate functions, we can obtain the same results for other surrogate functions.

## D.1   Fairness Definitions

### D.1.1   Disparate Mistreatment

It is a set of definitions based on both predicted and actual outcomes. Disparate mistreatment can be avoided if misclassification rate for different sensitive groups of people are the same. It can be specified w.r.t. a variety of misclassification metrics and each one corresponds to a kind of fairness definition. Note that it comes out in (Zafar et al., 2017a), and it includes some famous fairness notions, such as equality of opportunity. We list some examples used in this work below:

*Overall accuracy equality*:

To satisfy the definition, Overall Misclassification Rate (OMR) should be the same for protected and unprotected group, i.e.,

$$P(\hat{y} \neq y | z = -1) = P(\hat{y} \neq y | z = 1).$$

*False positive error rate balance*:

To satisfy the definition, False Positive Rate (FPR) should be the same for protected and unprotected group, i.e.,

$$P(\hat{y} \neq y | z = -1, y = -1) = P(\hat{y} \neq y | z = +1, y = -1).$$

*False negative error rate balance*:

To satisfy the definition, False Negative Rate (FNR) should be the same for protected and unprotected group, i.e.,

$$P(\hat{y} \neq y | z = -1, y = +1) = P(\hat{y} \neq y | z = +1, y = +1).$$

### D.1.2   Balance for Positive (Negative) Class

These two definitions consider the actual outcome as well as the predicted probability $\hat{P}(d_\theta(\mathbf{x}) > 0 | z = +1)$. Balance for positive class states that average predicted probability should be the same for those from positive class in protected and unprotected groups, i.e.,

$$\hat{P}(d_\theta(\mathbf{x}) > 0 | z = +1) = \hat{P}(d_\theta(\mathbf{x}) > 0 | z = -1).$$

Replace $d_\theta(\mathbf{x}) > 0$ with $d_\theta(\mathbf{x}) < 0$ above and we can get the definition of balance for negative class.

## D.2   Deeper Understanding of the Covariance Proxy for Disparate Mistreatment

In this part, we deal with disparate mistreatment. We denote $S$ as a certain set depending on each definition of disparate mistreatment. Notice that we have to consider misclassified points. In other words, the true label

Table 3: The corresponding $g_\theta(y, \mathbf{x})$ for each fairness definition and the set $S$ used for summation. For the first three lines, (Zafar et al., 2017a) only gives the form of $g_\theta(y, \mathbf{x})$ and $S$ defaults to the whole training set. We supplement it by providing $S$ for each definition. For the remaining two lines, we design the specific forms of $g_\theta(y, \mathbf{x})$ and $S$ for these two definitions. It is an extension work for the covariance proxy in algorithmic fairness.

| **Fairness Definitions** | $g_\theta(y, \mathbf{x})$ | $S$ |
|---|---|---|
| Overall accuracy equality | $\min(0, yd_\theta(x))$ | $\{(x_i, y_i)\|i = 1, 2...N\}$ |
| False positive error rate balance | $\min(0, \frac{1-y}{2}yd_\theta(x))$ | $\{(x_i, y_i)\|y_i = -1, i = 1, 2..., N\}$ |
| False negative error rate balance | $\min(0, \frac{1+y}{2}yd_\theta(x))$ | $\{(x_i, y_i)\|y_i = +1, i = 1, 2..., N\}$ |
| Balance for positive class | $\frac{1+y}{2}d_\theta(\mathbf{x})$ | $\{(x_i, y_i)\|y_i = +1, i = 1, 2..., N\}$ |
| Balance for negative class | $\frac{1-y}{2}d_\theta(\mathbf{x})$ | $\{(x_i, y_i)\|y_i = -1, i = 1, 2..., N\}$ |

$y$ should be considered. So $g_\theta(y, \mathbf{x})$ is used to replace $d_\theta(\mathbf{x})$. First of all, we prove that, with the assumption $D_\theta(\mathbf{x}) = 1$, if $\frac{1}{N}\sum_S(z_i - \bar{z})g_\theta(y, \mathbf{x}_i) = 0$, then it perfectly satisfies the corresponding fairness definition. In the first three rows in Table 3, the specific forms of $g_\theta(y, \mathbf{x})$ and $S$ are given for each kind of definition of disparate mistreatment. It is somewhat similar to (Lohaus et al., 2020) because they also discover that the covariance proxy proposed by (Zafar et al., 2017a) is equivalent to convex-concave surrogates. Secondly, we extend Theorem 1 to these three fairness definitions. Our general sigmoid surrogate can also be used for disparate mistreatment and we leave it for future work.

Note that no matter what the definition of disparate mistreatment is, $S$ defaults to the whole training set in (Zafar et al., 2017a). But we contend that it does not match their expected fairness definitions. The authors correct it in (Zafar et al., 2019), so our proof can also serve as an explanation of the correction. Our proof is in Appendix D.3.

### D.3   Proof for Appendix D.2

The proof process below is similar to that between the proxy and disparate impact. We provide the proof for the three fairness definitions of disparate mistreatment one by one. Before the proof, for the convenience of the notation, we divide the training set into 8 groups as follows:

Table 4: $z = +1$

| Predicted / Actual | +1 | -1 |
|---|---|---|
| +1 | $TP_+$ | $TN_+$ |
| -1 | $FP_+$ | $FN_+$ |

Table 5: $z = -1$

| Predicted / Actual | +1 | -1 |
|---|---|---|
| +1 | $TP_-$ | $TN_-$ |
| -1 | $FP_-$ | $FN_-$ |

For looking up different misclassification metrics conveniently, we provide the confusion matrix below:

Table 6: Confusion matrix

| Predicted / Actual | +1 | -1 |
|---|---|---|
| +1 | True Positive(TP) $PPV = \frac{TP}{TP+FP}$ $TPR = \frac{TP}{TP+FN}$ | False Negative(FN) $FOR = \frac{FN}{TN+FN}$ $FNR = \frac{FN}{TP+FN}$ |
| -1 | False Positive(FP) $FDR = \frac{FP}{TP+FP}$ $FPR = \frac{FP}{FP+TN}$ | True Negative(TN) $NPV = \frac{TN}{TN+FN}$ $TNR = \frac{TN}{TN+FP}$ |

### D.3.1 Overall Accuracy Equality

**Theorem 5.** *If a classifier satisfies $\frac{1}{N}\sum_{i=1}^{N}(z_i - \overline{z})g_\theta(y, \mathbf{x}_i) = 0$, where $g_\theta(y, \mathbf{x}) = min(0, yd_\theta(x))$, then there holds*

$$\overline{D_\theta(\mathbf{x})}_{y\neq\hat{y}|z=+1} = \overline{D_\theta(\mathbf{x})}_{y\neq\hat{y}|z=-1}$$

*Equal OMR in protected and unprotected group is equivalent to:*

$$\overline{sign(D_\theta(\mathbf{x}))}_{y\neq\hat{y}|z=+1} = \overline{sign(D_\theta(\mathbf{x}))}_{y\neq\hat{y}|z=-1}$$

*Proof.*

$$g_\theta(y, \mathbf{x}) = min(0, yd_\theta(x))$$

$$= \begin{cases} 0 & y = \hat{y} \; (correctly\; classified) \\ yd_\theta(x) & y \neq \hat{y} \; (misclassified) \end{cases}$$

$$= \begin{cases} 0 & y = \hat{y} \; (correctly\; classified) \\ -D_\theta(\mathbf{x}) & y \neq \hat{y} \; (misclassified) \end{cases}$$

$$\frac{1}{N}\sum_{i=1}^{N}(z_i - \overline{z})g_\theta(y, \mathbf{x}_i)$$

$$= -\frac{1}{N}\sum_{TN_+,FP_+,TN_-,FP_-}(z_i - \overline{z})D_\theta(x_i)$$

$$= -\frac{1}{N}\left(\sum_{TN_+,FP_+,TN_-,FP_-}z_iD_\theta(x_i) - \overline{z}\sum_{TN_+,FP_+,TN_-,FP_-}D_\theta(x_i)\right)$$

$$= -\frac{1}{N}\left[\sum_{TN_+,FP_+}D_\theta(x_i) - \sum_{TN_-,FP_-}D_\theta(x_i)\right.$$

$$-\frac{TP_+ + TN_+ + FP_+ + FN_+ - TP_- - TN_- - FP_- - FN_-}{N} \sum_{TN_+, FP_+, TN_-, FP_-} D_\theta(x_i)\Bigg]$$

$$= \frac{2}{N^2}\Bigg[(TP_+ + TN_+ + FP_+ + FN_+)\sum_{TN_-, FP_-} D_\theta(x_i) - (TP_- + TN_- + FP_- + FN_-)\sum_{TN_+, FP_+} D_\theta(x_i)\Bigg]$$

$$= \frac{2(TP_+ + TN_+ + FP_+ + FN_+)(TP_- + TN_- + FP_- + FN_-)}{N^2}$$

$$\cdot \left(\frac{\sum_{TN_-, FP_-} D_\theta(x_i)}{TP_- + TN_- + FP_- + FN_-} - \frac{\sum_{TN_+, FP_+} D_\theta(x_i)}{TP_+ + TN_+ + FP_+ + FN_+}\right)$$

$$= \frac{2(TP_+ + TN_+ + FP_+ + FN_+)(TP_- + TN_- + FP_- + FN_-)}{N^2}(\overline{D_\theta(\mathbf{x})}_{y \neq \hat{y}|z=-1} - \overline{D_\theta(\mathbf{x})}_{y \neq \hat{y}|z=+1}) \qquad (34)$$

We set the formula above equal to zero so there holds

$$\overline{D_\theta(\mathbf{x})}_{y \neq \hat{y}|z=+1} = \overline{D_\theta(\mathbf{x})}_{y \neq \hat{y}|z=-1}$$

Equal OMR in protected and unprotected group means $P(\hat{y} \neq y|z = +1) = P(\hat{y} \neq y|z = -1)$, i.e., $\frac{TN_- + FP_-}{TP_- + TN_- + FP_- + FN_-} = \frac{TN_+ + FP_+}{TP_+ + TN_+ + FP_+ + FN_+}$. It is equivalent to

$$\overline{sign(D_\theta(\mathbf{x}))}_{y \neq \hat{y}|z=+1} = \overline{sign(D_\theta(\mathbf{x}))}_{y \neq \hat{y}|z=-1}$$

So we complete the proof. $\qquad\square$

**Corollary 1.** *We assume that $D_\theta(\mathbf{x}) = 1$. If a classifier satisfies $\frac{1}{N}\sum_{i=1}^N (z_i - \bar{z})g_\theta(y, \mathbf{x}_i) = 0$, where $g_\theta(y, \mathbf{x}) = min(0, yd_\theta(x))$, then it perfectly meets with overall accuracy equality.*

*Proof.* The method is similar to that for disparate impact. We just substitute $D_\theta(\mathbf{x})$ with $sign(D_\theta(\mathbf{x}))$ and understand that this corollary is true. $\qquad\square$

**Theorem 6.** *We assume that $D_\theta(\mathbf{x}) \in [1 - \gamma, 1 + \gamma]$. $\forall \epsilon > 0$, if the proxy satisfies:*

$$\left|\frac{1}{N}\sum_{i=1}^N (z_i - \bar{z})g_\theta(y, \mathbf{x}_i)\right| < \epsilon$$

*where $g_\theta(y, \mathbf{x}) = min(0, yd_\theta(x))$, then the violation of overall accuracy equality satisfies:*

$$|P(\hat{y} \neq y|z = +1) - P(\hat{y} \neq y|z = -1)| < \frac{N^2}{2(TP_+ + TN_+ + FP_+ + FN_+)(TP_- + TN_- + FP_- + FN_-)}\epsilon +$$
$$\left(\frac{TN_- + FP_-}{TP_- + TN_- + FP_- + FN_-} + \frac{TN_+ + FP_+}{TP_+ + TN_+ + FP_+ + FN_+}\right)\gamma$$

*Proof.* It is connected to the theorem above.

$$\left|\frac{1}{N}\sum_{i=1}^N (z_i - \bar{z})g_\theta(y, \mathbf{x}_i)\right| < \epsilon$$

$$\Rightarrow \left|\frac{2(TP_+ + TN_+ + FP_+ + FN_+)(TP_- + TN_- + FP_- + FN_-)}{N^2}\right.$$

$$\left.\cdot\left(\frac{\sum_{TN_-, FP_-} D_\theta(x_i)}{TP_- + TN_- + FP_- + FN_-} - \frac{\sum_{TN_+, FP_+} D_\theta(x_i)}{TP_+ + TN_+ + FP_+ + FN_+}\right)\right| < \epsilon$$

$$\Rightarrow \left| \frac{\sum_{TN_-,FP_-} D_\theta(x_i)}{TP_- + TN_- + FP_- + FN_-} - \frac{\sum_{TN_+,FP_+} D_\theta(x_i)}{TP_+ + TN_+ + FP_+ + FN_+} \right|$$

$$< \frac{N^2}{2(TP_+ + TN_+ + FP_+ + FN_+)(TP_- + TN_- + FP_- + FN_-)}\epsilon$$

$$\Rightarrow \left| \frac{TN_- + FP_-}{TP_- + TN_- + FP_- + FN_-} - \frac{TN_+ + FP_+}{TP_+ + TN_+ + FP_+ + FN_+} \right|$$

$$< \frac{N^2}{2(TP_+ + TN_+ + FP_+ + FN_+)(TP_- + TN_- + FP_- + FN_-)}\epsilon$$

$$+ \left| \frac{\sum_{TN_-,FP_-}(D_\theta(x_i) - 1)}{TP_- + TN_- + FP_- + FN_-} - \frac{\sum_{TN_+,FP_+}(D_\theta(x_i) - 1)}{TP_+ + TN_+ + FP_+ + FN_+} \right| \quad (35)$$

If $D_\theta(\mathbf{x}) \in [1 - \gamma, 1 + \gamma]$, then $D_\theta(\mathbf{x}) - 1 \in [-\gamma, \gamma]$, which means that:

$$\left| \frac{\sum_{TN_-,FP_-}(D_\theta(x_i) - 1)}{TP_- + TN_- + FP_- + FN_-} - \frac{\sum_{TN_+,FP_+}(D_\theta(x_i) - 1)}{TP_+ + TN_+ + FP_+ + FN_+} \right|$$

$$< \left( \frac{TN_- + FP_-}{TP_- + TN_- + FP_- + FN_-} + \frac{TN_+ + FP_+}{TP_+ + TN_+ + FP_+ + FN_+} \right)\gamma$$

Notice that:

$$|P(\hat{y} \neq y | z = +1) - P(\hat{y} \neq y | z = -1)| = \left| \frac{TN_- + FP_-}{TP_- + TN_- + FP_- + FN_-} - \frac{TN_+ + FP_+}{TP_+ + TN_+ + FP_+ + FN_+} \right|$$

So by combining these formulas together and we can complete the proof. $\square$

### D.3.2 False Positive Error Rate Balance

**Theorem 7.** *S denotes the set of points which satisfy $y = -1$ in the whole training set and $g_\theta(y, \mathbf{x}) = min(0, \frac{1-y}{2}yd_\theta(x))$. If a classifier satisfies $\frac{1}{N}\sum_S (z_i - \overline{z})g_\theta(y, \mathbf{x}_i) = 0$, then there holds:*

$$\overline{D_\theta(\mathbf{x})}_{y \neq \hat{y} | z = +1, y = -1} = \overline{D_\theta(\mathbf{x})}_{y \neq \hat{y} | z = -1, y = -1}$$

*Equal FPR in protected and unprotected group is equivalent to:*

$$\overline{sign(D_\theta(\mathbf{x}))}_{y \neq \hat{y} | z = +1, y = -1} = \overline{sign(D_\theta(\mathbf{x}))}_{y \neq \hat{y} | z = -1, y = -1}$$

*Proof.*

$$g_\theta(y, \mathbf{x}) = min(0, \frac{1-y}{2}yd_\theta(x))$$

$$= \begin{cases} yd_\theta(x) & y = -1, \hat{y} = +1 \ (false \ positive) \\ 0 & others \end{cases}$$

$$= \begin{cases} -D_\theta(\mathbf{x}) & y = -1, \hat{y} = +1 \ (false \ positive) \\ 0 & others \end{cases}$$

$$\frac{1}{N}\sum_S (z_i - \overline{z})g_\theta(y, \mathbf{x}_i)$$

$$= -\frac{1}{N}\sum_{FP_+,FP_-} (z_i - \overline{z})D_\theta(x_i)$$

$$= -\frac{1}{N}\left[\sum_{FP_+,FP_-} z_i D_\theta(x_i) - \overline{z}\sum_{FP_+,FP_-} D_\theta(x_i)\right]$$

$$= -\frac{1}{N}\left[\sum_{FP_+} D_\theta(x_i) - \sum_{FP_-} D_\theta(x_i) - \frac{FP_+ + FN_+ - G - H}{FP_+ + FN_+ + FP_- + FN_-}\sum_{FP_+,FP_-} D_\theta(x_i)\right]$$

$$= -\frac{2(FP_+ + FN_+)(FP_- + FN_-)}{N(FP_+ + FN_+ + FP_- + FN_-)}\left(\frac{\sum_{FP_+} D_\theta(x_i)}{FP_+ + FN_+} - \frac{\sum_{FP_-} D_\theta(x_i)}{FP_- + FN_-}\right)$$

$$= -\frac{2(FP_+ + FN_+)(FP_- + FN_-)}{N(FP_+ + FN_+ + FP_- + FN_-)}\left(\overline{D_\theta(\mathbf{x})}_{y\neq\hat{y}|z=+1,y=-1} - \overline{D_\theta(\mathbf{x})}_{y\neq\hat{y}|z=-1,y=-1}\right) \tag{36}$$

Set this equal to zero and the formula above becomes:

$$\overline{D_\theta(\mathbf{x})}_{y\neq\hat{y}|z=+1,y=-1} = \overline{D_\theta(\mathbf{x})}_{y\neq\hat{y}|z=-1,y=-1}$$

Equal FPR in protected and unprotected group satisfies $\frac{FP_+}{FP_++FN_+} = \frac{FP_-}{FP_-+FN_-}$, which means

$$\overline{sign(D_\theta(\mathbf{x}))}_{y\neq\hat{y}|z=+1,y=-1} = \overline{sign(D_\theta(\mathbf{x}))}_{y\neq\hat{y}|z=-1,y=-1}$$

So we complete the proof. $\square$

**Corollary 2.** *$S$ denotes the set of points which satisfy $y = -1$ in the whole training set and $g_\theta(y,\mathbf{x}) = min(0,\frac{1-y}{2}yd_\theta(x))$. We assume that $D_\theta(\mathbf{x}) = 1$. If a classifier satisfies $\frac{1}{N}\sum_S(z_i - \overline{z})g_\theta(y,\mathbf{x}_i) = 0$, then it meets with false positive error rate balance.*

*Proof.* We just replace $D_\theta(\mathbf{x})$ with $sign(D_\theta(\mathbf{x}))$ to prove the result. $\square$

**Corollary 3.** *We assume that $D_\theta(\mathbf{x}) = 1$. If a classifier satisfies $\frac{1}{N}\sum_{i=1}^N(z_i - \overline{z})g_\theta(y,\mathbf{x}_i) = 0$, where $g_\theta(y,\mathbf{x}) = min(0,\frac{1-y}{2}yd_\theta(x))$, then it **does not** meet with false positive error rate balance.*

*Proof.*

$$\frac{1}{N}\sum_{i=1}^N(z_i - \overline{z})g_\theta(y,\mathbf{x}_i)$$

$$= -\frac{1}{N}\sum_{FP_+,FP_-}(z_i - \overline{z})D_\theta(x_i)$$

$$= -\frac{1}{N}\sum_{FP_+,FP_-}(z_i - \overline{z})$$

$$= -\frac{1}{N}\left(\sum_{FP_+,FP_-} z_i - \sum_{FP_+,FP_-}\overline{z}\right)$$

$$= -\frac{1}{N}\left[(FP+ -FP_-) - (FP+ +FP_-)\frac{TP_+ + TN_+ + FP_+ + FN_+ - E - F - G - H}{N}\right]$$

$$= -\frac{2}{N^2}\left[C(TP_- + TN_- + FP_- + FN_-) - G(TP_+ + TN_+ + FP_+ + FN_+)\right] \tag{37}$$

Set this equal to zero and the formula above becomes:

$$\frac{FP_+}{TP_+ + TN_+ + FP_+ + FN_+} = \frac{FP_-}{TP_- + TN_- + FP_- + FN_-}$$

However, equal FPR in protected and unprotected group satisfies the formula:

$$\frac{FP_+}{FP_+ + FN_+} = \frac{FP_-}{FP_- + FN_-}$$

So the two equations do not match. The proof is complete. $\quad\square$

**Theorem 8.** *We assume that $D_\theta(\mathbf{x}) \in [1 - \gamma, 1 + \gamma]$. $\forall \epsilon > 0$, if the proxy satisfies:*

$$\left| \frac{1}{N} \sum_S (z_i - \overline{z}) g_\theta(y, \mathbf{x}_i) \right| < \epsilon$$

*where $g_\theta(y, \mathbf{x}) = min(0, \frac{1-y}{2} y d_\theta(x))$, then the violation of false positive error rate balance satisfies:*

$$
|P(\hat{y} \neq y | z = +1, y = -1) - P(\hat{y} \neq y | z = -1, y = -1)| < \frac{N(FP_+ + FN_+ + FP_- + FN_-)}{2(FP_+ + FN_+)(FP_- + FN_-)} \epsilon
$$
$$
+ \left( \frac{FP_+}{FP_+ + FN_+} + \frac{FP_-}{FP_- + FN_-} \right) \gamma \qquad (38)
$$

*Proof.*

$$\left| \frac{1}{N} \sum_S (z_i - \overline{z}) g_\theta(y, \mathbf{x}_i) \right| < \epsilon$$

$$\Rightarrow \left| \frac{\sum_{FP_+} D_\theta(x_i)}{FP_+ + FN_+} - \frac{\sum_{FP_-} D_\theta(x_i)}{FP_- + FN_-} \right| < \frac{N(FP_+ + FN_+ + FP_- + FN_-)}{2(FP_+ + FN_+)(FP_- + FN_-)} \epsilon$$

$$\Rightarrow \left| \frac{FP_+}{FP_+ + FN_+} - \frac{FP_-}{FP_- + FN_-} \right| < \frac{N(FP_+ + FN_+ + FP_- + FN_-)}{2(FP_+ + FN_+)(FP_- + FN_-)} \epsilon$$

$$+ \left| \frac{\sum_{FP_+} (D_\theta(x_i) - 1)}{FP_+ + FN_+} - \frac{\sum_{FP_-} (D_\theta(x_i) - 1)}{FP_- + FN_-} \right| \qquad (39)$$

If $D_\theta(\mathbf{x}) \in [1 - \gamma, 1 + \gamma]$, then $D_\theta(\mathbf{x}) - 1 \in [-\gamma, \gamma]$, which means that:

$$\left| \frac{\sum_{FP_+} (D_\theta(x_i) - 1)}{FP_+ + FN_+} - \frac{\sum_{FP_-} (D_\theta(x_i) - 1)}{FP_- + FN_-} \right| < \left( \frac{FP_+}{FP_+ + FN_+} + \frac{FP_-}{FP_- + FN_-} \right) \gamma$$

Notice that

$$|P(\hat{y} \neq y | z = +1, y = -1) - P(\hat{y} \neq y | z = -1, y = -1)| = \left| \frac{FP_+}{FP_+ + FN_+} - \frac{FP_-}{FP_- + FN_-} \right|$$

So by combining these formulas together and we can complete the proof. $\quad\square$

### D.3.3 False Negative Error Rate Balance

**Theorem 9.** *$S$ denotes the set of points which satisfy $y = +1$ in the whole training set and $g_\theta(y, \mathbf{x}) = min(0, \frac{1+y}{2} y d_\theta(x))$. If a classifier satisfies $\frac{1}{N} \sum_S (z_i - \overline{z}) g_\theta(y, \mathbf{x}_i) = 0$, then there holds:*

$$\overline{D_\theta(\mathbf{x})}_{y \neq \hat{y} | z = +1, y = +1} = \overline{D_\theta(\mathbf{x})}_{y \neq \hat{y} | z = -1, y = +1}$$

*Equal FNR in protected and unprotected group is equivalent to:*

$$\overline{sign(D_\theta(\mathbf{x}))}_{y \neq \hat{y} | z = +1, y = +1} = \overline{sign(D_\theta(\mathbf{x}))}_{y \neq \hat{y} | z = -1, y = +1}$$

*Proof.*

$$g_\theta(y, \mathbf{x}) = min(0, \frac{1+y}{2} y d_\theta(x))$$

$$= \begin{cases} y d_\theta(x) & y = +1, \hat{y} = -1 \ (false \ negative) \\ 0 & others \end{cases}$$

$$= \begin{cases} -D_\theta(\mathbf{x}) & y = +1, \hat{y} = -1 \ (false \ negative) \\ 0 & others \end{cases}$$

$$\frac{1}{N} \sum_S (z_i - \bar{z}) g_\theta(y, \mathbf{x}_i)$$

$$= -\frac{1}{N} \sum_{TN_+, TN_-} (z_i - \bar{z}) D_\theta(x_i)$$

$$= -\frac{1}{N} \left[ \sum_{TN_+, TN_-} z_i D_\theta(x_i) - \bar{z} \sum_{TN_+, TN_-} D_\theta(x_i) \right]$$

$$= -\frac{1}{N} \left[ \sum_{TN_+} D_\theta(x_i) - \sum_{TN_-} D_\theta(x_i) - \frac{TP_+ + TN_+ - E - F}{TP_+ + TN_+ + TP_- + TN_-} \sum_{TN_+, TN_-} D_\theta(x_i) \right]$$

$$= -\frac{2(TP_+ + TN_+)(TP_- + TN_-)}{N(TP_+ + TN_+ + TP_- + TN_-)} \left( \frac{\sum_{TN_+} D_\theta(x_i)}{TP_+ + TN_+} - \frac{\sum_{TN_-} D_\theta(x_i)}{TP_- + TN_-} \right)$$

$$= -\frac{2(TP_+ + TN_+)(TP_- + TN_-)}{N(TP_+ + TN_+ + TP_- + TN_-)} \left( \overline{D_\theta(\mathbf{x})}_{y \neq \hat{y}|z=+1, y=+1} - \overline{D_\theta(\mathbf{x})}_{y \neq \hat{y}|z=-1, y=+1} \right) \tag{40}$$

Set this equal to zero and the formula above becomes:

$$\overline{D_\theta(\mathbf{x})}_{y \neq \hat{y}|z=+1, y=+1} = \overline{D_\theta(\mathbf{x})}_{y \neq \hat{y}|z=-1, y=+1}$$

Equal FNR in protected and unprotected group satisfies $\frac{TN_+}{TP_+ + TN_+} = \frac{TN_-}{TP_- + TN_-}$, which means

$$\overline{sign(D_\theta(\mathbf{x}))}_{y \neq \hat{y}|z=+1, y=+1} = \overline{sign(D_\theta(\mathbf{x}))}_{y \neq \hat{y}|z=-1, y=+1}$$

So we complete the proof. $\qquad\square$

**Corollary 4.** *S denotes the set of points which satisfy $y = +1$ in the whole training set and $g_\theta(y, \mathbf{x}) = min(0, \frac{1+y}{2} y d_\theta(x))$. We assume that $D_\theta(\mathbf{x}) = 1$. If a classifier satisfies $\frac{1}{N} \sum_S (z_i - \bar{z}) g_\theta(y, \mathbf{x}_i) = 0$, then it meets with false negative error rate balance.*

*Proof.* We just replace $D_\theta(\mathbf{x})$ with $sign(D_\theta(\mathbf{x}))$ to prove the result. $\qquad\square$

**Corollary 5.** *We assume that $D_\theta(\mathbf{x}) = 1$. If a classifier satisfies $\frac{1}{N} \sum_{i=1}^N (z_i - \bar{z}) g_\theta(y, \mathbf{x}_i) = 0$, where $g_\theta(y, \mathbf{x}) = min(0, \frac{1+y}{2} y d_\theta(x))$, then it **does not** meet with false negative error rate balance.*

*Proof.*

$$\frac{1}{N} \sum_{i=1}^N (z_i - \bar{z}) g_\theta(y, \mathbf{x}_i)$$

$$= -\frac{1}{N} \sum_{TN_+, TN_-} (z_i - \bar{z}) D_\theta(x_i)$$

$$= -\frac{1}{N} \sum_{TN_+, TN_-} (z_i - \bar{z})$$

$$= -\frac{1}{N} \left( \sum_{TN_+, TN_-} z_i - \sum_{TN_+, TN_-} \bar{z} \right)$$

$$= -\frac{1}{N} \left[ (TN_+ - TN_-) - (TN_+ + TN_-) \frac{TP_+ + TN_+ + FP_+ + FN_+ - TP_- - TN_- - FP_- - FN_-}{N} \right]$$

$$= -\frac{2}{N^2} \left[ TN_+(TP_- + TN_- + FP_- + FN_-) - TN_-(TP_+ + TN_+ + FP_+ + FN_+) \right] \tag{41}$$

Set this equal to zero and the formula above becomes:

$$\frac{TN_+}{TP_+ + TN_+ + FP_+ + FN_+} = \frac{TN_-}{TP_- + TN_- + FP_- + FN_-}$$

However, equal FNR in protected and unprotected group satisfies the formula:

$$\frac{TN_+}{TP_+ + TN_+} = \frac{TN_-}{TP_- + TN_-}$$

So the two equations do not match. The proof is complete. $\qquad \square$

**Theorem 10.** *We assume that $D_\theta(\mathbf{x}) \in [1 - \gamma, 1 + \gamma]$. $\forall \epsilon > 0$, if the proxy satisfies:*

$$\left| \frac{1}{N} \sum_S (z_i - \bar{z}) g_\theta(y, \mathbf{x}_i) \right| < \epsilon$$

*where $g_\theta(y, \mathbf{x}) = min(0, \frac{1+y}{2} y d_\theta(x))$, then the violation of false negative error rate balance satisfies:*

$$|P(\hat{y} \neq y | z = +1, y = +1) - P(\hat{y} \neq y | z = -1, y = +1)| < \frac{N(TP_+ + TN_+ + TP_- + TN_-)}{2(TP_+ + TN_+)(TP_- + TN_-)} \epsilon$$

$$+ \left( \frac{TN_+}{TP_+ + TN_+} + \frac{TN_-}{TP_- + TN_-} \right) \gamma \tag{42}$$

*Proof.*

$$\left| \frac{1}{N} \sum_S (z_i - \bar{z}) g_\theta(y, \mathbf{x}_i) \right| < \epsilon$$

$$\Rightarrow \left| \frac{\sum_{TN_+} D_\theta(x_i)}{TP_+ + TN_+} - \frac{\sum_{TN_-} D_\theta(x_i)}{TP_- + TN_-} \right| < \frac{N(TP_+ + TN_+ + TP_- + TN_-)}{2(TP_+ + TN_+)(TP_- + TN_-)} \epsilon$$

$$\Rightarrow \left| \frac{TN_+}{TP_+ + TN_+} - \frac{TN_-}{TP_- + TN_-} \right| < \frac{N(TP_+ + TN_+ + TP_- + TN_-)}{2(TP_+ + TN_+)(TP_- + TN_-)} \epsilon$$

$$+ \left| \frac{\sum_{TN_+} (D_\theta(x_i) - 1)}{TP_+ + TN_+} - \frac{\sum_{TN_-} (D_\theta(x_i) - 1)}{TP_- + TN_-} \right| \tag{43}$$

If $D_\theta(\mathbf{x}) \in [1 - \gamma, 1 + \gamma]$, then $D_\theta(\mathbf{x}) - 1 \in [-\gamma, \gamma]$, which means that:

$$\left| \frac{\sum_{TN_+} (D_\theta(x_i) - 1)}{TP_+ + TN_+} - \frac{\sum_{TN_-} (D_\theta(x_i) - 1)}{TP_- + TN_-} \right| < \left( \frac{TN_+}{TP_+ + TN_+} + \frac{TN_-}{TP_- + TN_-} \right) \gamma$$

Notice that

$$|P(\hat{y} \neq y | z = +1, y = +1) - P(\hat{y} \neq y | z = -1, y = +1)| = \left| \frac{TN_+}{TP_+ + TN_+} - \frac{TN_-}{TP_- + TN_-} \right|$$

So by combining these formulas together and we can complete the proof. $\qquad \square$

### D.4 Deeper Understanding of the Covariance Proxy for Balance for Positive (Negative) Class

In this section, we deal with balance for positive/negative class. These two fairness definitions are prepared for probabilistic classifiers, which are different from previous issues. Many probabilistic classifiers do not model the decision boundary directly, so it is hard to relate the proxy to these two definitions. To solve this problem, we creatively link $d_\theta(\mathbf{x})$ and the prediction probability $\hat{P}(d_\theta(\mathbf{x}) > 0|z = +1)$ together using an assumption.

To begin with, we first consider the relationship between $d_\theta(\mathbf{x})$ and the prediction probability $\hat{P}(d_\theta(\mathbf{x}) > 0|z = +1)$. Imagine that there exists a decision boundary for probabilistic classifiers. For a probabilistic classifier, the positive prediction result is the same as $\hat{P}(d_\theta(\mathbf{x}) > 0|z = +1) > \frac{1}{2}$, i.e., $\hat{P}(d_\theta(\mathbf{x}) > 0|z = +1) - \frac{1}{2} > 0$. Similarly, for margin-based classifiers, it is equivalent to $d_\theta(\mathbf{x}) > 0$. Thus, $d_\theta(\mathbf{x})$ and $\hat{P}(d_\theta(\mathbf{x}) > 0|z = +1) - \frac{1}{2}$ share a similar relative trend, but the values of them are not in the same range. $d_\theta(\mathbf{x}) \in \mathbb{R}$ , while $\hat{P}(d_\theta(\mathbf{x}) > 0|z = +1) - \frac{1}{2} \in [-\frac{1}{2}, \frac{1}{2}]$.

This suggests that we can view probabilistic classifiers as special margin-based classifiers. They find the location of the decision boundary, and at the same time, learn a mapping $f : \mathbb{R} \mapsto [-\frac{1}{2}, \frac{1}{2}]$ from $d_\theta(\mathbf{x})$ to $\hat{P}(d_\theta(\mathbf{x}) > 0|z = +1) - \frac{1}{2}$. In other words,

$$f(d_\theta(\mathbf{x})) = \hat{P}(d_\theta(\mathbf{x}) > 0|z = +1) - \frac{1}{2}.$$

It is challenging to find the mapping $f$ directly. However, intuitively, we know some properties of it. It is a bounded and monotonically increasing function which passes through the origin. Thus, we can design a function $\phi$ to approximately reflect the trend of $f$. For example, $\phi(x) = \sigma(x) - \frac{1}{2}$. We assume

$$\phi(d_\theta(\mathbf{x})) = \hat{P}(d_\theta(\mathbf{x}) > 0|z = +1) - \frac{1}{2},$$

and prove that if $\frac{1}{N}\sum_S (z_i - \overline{z})g_\theta(y, \mathbf{x}_i) = 0$, then it perfectly satisfies the corresponding fairness definition. The specific forms of $S$ and $g_\theta(y, \mathbf{x})$ for these two fairness notions are shown in the last two lines in Table 3. To the best of our knowledge, we are the first to connect the proxy to these two definitions. Please refer to Appendix D.5 for proof details.

Although the assumption is strong, we aim to provide the basis for future work about how to choose a suitable $\phi$ to approximate $f$, thereby inspiring the research community. We believe that it can be an interesting direction of future work for these fairness definitions based on predicted probability.

### D.5 Proof for Appendix D.4

#### D.5.1 Balance for Positive Class

**Theorem 11.** *S denotes the set of points which satisfy $y = +1$ in the whole training set and $g_\theta(y, \mathbf{x}) = \frac{1+y}{2}d_\theta(\mathbf{x})$. We assume that $\phi(d_\theta(\mathbf{x})) = \hat{P}(d_\theta(\mathbf{x}) > 0|z = +1) - \frac{1}{2}$, where $\phi : \mathbb{R} \mapsto [-\frac{1}{2}, \frac{1}{2}]$ and it is an odd function. If a classifier satisfies $\frac{1}{N}\sum_S (z_i - \overline{z})\phi(g_\theta(y, \mathbf{x}_i)) = 0$, then it perfectly meets with balance for positive class.*

*Proof.*

$$
\begin{aligned}
g_\theta(y, \mathbf{x}) =& \frac{1+y}{2}d_\theta(\mathbf{x}) \\
=& \begin{cases} d_\theta(\mathbf{x}) & y = +1 \\ 0 & others \end{cases} \\
=& \begin{cases} D_\theta(\mathbf{x}) & y = +1, \hat{y} = +1 \ (true \ positive) \\ -D_\theta(\mathbf{x}) & y = +1, \hat{y} = -1 \ (false \ negative) \\ 0 & others \end{cases}
\end{aligned}
$$

$$\frac{1}{N}\sum_S (z_i - \overline{z})\phi(g_\theta(y, \mathbf{x}_i))$$

$$=\frac{1}{N}\sum_{TP_+, TN_+, TP_-, TN_-} (z_i - \overline{z})\phi(d_\theta(x_i))$$

$$=\frac{1}{N}\left[\sum_{TP_+, TP_-}(z_i - \overline{z})\phi(D_\theta(x_i)) - \sum_{TN_+, TN_-}(z_i - \overline{z})\phi(D_\theta(x_i))\right]$$

$$=\frac{1}{N}\left[\sum_{TP_+}(1 - \overline{z})\phi(D_\theta(x_i)) - \sum_{TN_+}(1 - \overline{z})\phi(D_\theta(x_i)) + \sum_{TP_-}(-1 - \overline{z})\phi(D_\theta(x_i)) - \sum_{TN_-}(-1 - \overline{z})\phi(D_\theta(x_i))\right]$$

$$=\frac{1}{N}\left[(1 - \overline{z})\left(\sum_{TP_+}\phi(D_\theta(x_i)) - \sum_{TN_+}\phi(D_\theta(x_i))\right) + (-1 - \overline{z})\left(\sum_{TP_-}\phi(D_\theta(x_i)) - \sum_{TN_-}\phi(D_\theta(x_i))\right)\right]$$

$$=\frac{2(TP_+ + TN_+ + TP_- + TN_-)}{N}$$

$$\cdot\left[(TP_- + TN_-)\left(\sum_{TP_+}\phi(D_\theta(x_i)) - \sum_{TN_+}\phi(D_\theta(x_i))\right) - (TP_+ + TN_+)\left(\sum_{TP_-}\phi(D_\theta(x_i)) - \sum_{TN_-}\phi(D_\theta(x_i))\right)\right]$$

$$\tag{44}$$

Note that $TP_+ + TN_+ + TP_- + TN_-$ represents the number of points in the training set that belong to the positive class, so it is a constant. Set this formula equal to zero and it becomes:

$$\frac{\sum_{TP_+}\phi(D_\theta(x_i)) - \sum_{TN_+}\phi(D_\theta(x_i))}{TP_+ + TN_+} = \frac{\sum_{TP_-}\phi(D_\theta(x_i)) - \sum_{TN_-}\phi(D_\theta(x_i))}{TP_- + TN_-} \tag{45}$$

We notice that:

$$\overline{\phi(d_\theta(x_i))}_{z=+1, y=+1} = \frac{\sum_{TP_+}\phi(D_\theta(x_i)) - \sum_{TN_+}\phi(D_\theta(x_i))}{TP_+ + TN_+}$$

$$\overline{\phi(d_\theta(x_i))}_{z=-1, y=+1} = \frac{\sum_{TP_-}\phi(D_\theta(x_i)) - \sum_{TN_-}\phi(D_\theta(x_i))}{TP_- + TN_-}$$

so if $\phi(d_\theta(\mathbf{x})) = \hat{P}(d_\theta(\mathbf{x}) > 0|z = +1) - \frac{1}{2}$, then:

$$\hat{P}(d_\theta(\mathbf{x}) > 0|z = +1) = \frac{\sum_{TP_+}\phi(D_\theta(x_i)) - \sum_{TN_+}\phi(D_\theta(x_i))}{TP_+ + TN_+} + \frac{1}{2}$$

$$\hat{P}(d_\theta(\mathbf{x}) > 0|z = -1) = \frac{\sum_{TP_-}\phi(D_\theta(x_i)) - \sum_{TN_-}\phi(D_\theta(x_i))}{TP_- + TN_-} + \frac{1}{2} \tag{46}$$

With the equations (45) and (46), we get:

$$\hat{P}(d_\theta(\mathbf{x}) > 0|z = +1) = \hat{P}(d_\theta(\mathbf{x}) > 0|z = -1)$$

which perfectly meets with balance for positive class. The proof is complete. $\qquad\square$

### D.5.2 Balance for Negative Class

**Theorem 12.** *S denotes the set of points which satisfy $y = -1$ in the whole training set and $g_\theta(y, \mathbf{x}) = \frac{1-y}{2}\phi(d_\theta(\mathbf{x}))$. We assume that $\phi(d_\theta(\mathbf{x})) = \hat{P}(d_\theta(\mathbf{x}) < 0|z = +1) - \frac{1}{2}$, where $\phi : \mathbb{R} \mapsto [-\frac{1}{2}, \frac{1}{2}]$ and it is an odd function. If a classifier satisfies $\frac{1}{N}\sum_S(z_i - \overline{z})\phi(g_\theta(y, \mathbf{x}_i)) = 0$, then it perfectly meets with balance for negative class.*

*Proof.*

$$
\begin{aligned}
g_\theta(y, \mathbf{x}) &= \frac{1-y}{2}d_\theta(\mathbf{x}) \\
&= \begin{cases} d_\theta(\mathbf{x}) & y = -1 \\ 0 & others \end{cases} \\
&= \begin{cases} d_\theta(\mathbf{x}) & y = -1, \hat{y} = +1 \ (false\ positive) \\ -d_\theta(\mathbf{x}) & y = -1, \hat{y} = -1 \ (true\ negative) \\ 0 & others \end{cases}
\end{aligned}
$$

$$
\begin{aligned}
&\frac{1}{N}\sum_S (z_i - \overline{z})\,\phi\,(g_\theta\,(y, \mathbf{x}_i)) \\
=&\frac{1}{N}\sum_{FP_+, FN_+, FP_-, FN_-} (z_i - \overline{z})\,\phi\,(D_\theta\,(x_i)) \\
=&\frac{1}{N}\left[\sum_{FP_+, FP_-} (z_i - \overline{z})\,\phi\,(D_\theta\,(x_i)) - \sum_{FN_+, FN_-} (z_i - \overline{z})\,\phi\,(D_\theta\,(x_i))\right] \\
=&\frac{1}{N}\left[\sum_{FP_+} (1 - \overline{z})\,\phi\,(D_\theta\,(x_i)) - \sum_{FN_+} (1 - \overline{z})\,\phi\,(D_\theta\,(x_i)) + \sum_{FP_-} (-1 - \overline{z})\,\phi\,(D_\theta\,(x_i)) - \sum_{FN_-} (-1 - \overline{z})\,\phi\,(D_\theta\,(x_i))\right] \\
=&\frac{1}{N}\left[(1 - \overline{z})\left(\sum_{FP_+} \phi\,(D_\theta\,(x_i)) - \sum_{FN_+} \phi\,(D_\theta\,(x_i))\right) + (-1 - \overline{z})\left(\sum_{FP_-} \phi\,(D_\theta\,(x_i)) - \sum_{FN_-} \phi\,(D_\theta\,(x_i))\right)\right] \\
=&\frac{2\,(FP_+ + FN_+ + FP_- + FN_-)}{N} \\
&\cdot\left[(FP_- + FN_-)\left(\sum_{FP_+} \phi\,(D_\theta\,(x_i)) - \sum_{FN_+} \phi\,(D_\theta\,(x_i))\right) - (FP_+ + FN_+)\left(\sum_{FP_-} \phi\,(D_\theta\,(x_i)) - \sum_{FN_-} \phi\,(D_\theta\,(x_i))\right)\right]
\end{aligned}
$$

$$\tag{47}$$

Note that $FP_+ + FN_+ + FP_- + FN_-$ represents the number of points in the training set that belong to the negative class, so it is a constant. Set this formula equal to zero and it becomes:

$$
\frac{\sum_{FP_+}\phi(D_\theta(x_i)) - \sum_{FN_+}\phi(D_\theta(x_i))}{FP_+ + FN_+} = \frac{\sum_{FP_-}\phi(D_\theta(x_i)) - \sum_{FN_-}\phi(D_\theta(x_i))}{FP_- + FN_-} \tag{48}
$$

We notice that:

$$
\overline{\phi(d_\theta(\mathbf{x}))}_{z=+1, y=-1} = \frac{\sum_{FP_+}\phi(D_\theta(x_i)) - \sum_{FN_+}\phi(D_\theta(x_i))}{FP_+ + FN_+}
$$

$$\overline{\phi(d_\theta(\mathbf{x}))}_{z=-1,y=-1} = \frac{\sum_{FP_-} \phi(D_\theta(x_i)) - \sum_{FN_-} \phi(D_\theta(x_i))}{FP_- + FN_-}$$

so if $\phi(d_\theta(\mathbf{x})) = \hat{P}(d_\theta(\mathbf{x}) < 0 | z = +1) - \frac{1}{2}$, then:

$$\hat{P}(d_\theta(\mathbf{x}) < 0 | z = +1) = \frac{\sum_{FP_+} \phi(D_\theta(x_i)) - \sum_{FN_+} \phi(D_\theta(x_i))}{FP_+ + FN_+} + \frac{1}{2}$$

$$\hat{P}(d_\theta(\mathbf{x}) < 0 | z = -1) = \frac{\sum_{FP_-} \phi(D_\theta(x_i)) - \sum_{FN_-} \phi(D_\theta(x_i))}{FP_- + FN_-} + \frac{1}{2}$$

With the equations (48) and (49), we get:

$$\hat{P}(d_\theta(\mathbf{x}) < 0 | z = +1) = \hat{P}(d_\theta(\mathbf{x}) < 0 | z = -1)$$

which perfectly meets with balance for negative class. The proof is complete. $\square$

# E    Further Analysis

We provide some interesting results and analysis about this research. While the analysis may not have a direct impact on the main body of the paper, they could serve as potential directions for further exploration.

## E.1    Fairness Surrogate Functions in the Future

The goal of fairness surrogate functions in this paper is to approximate the indicator function. As can be seen in Figure 1, we propose the general sigmoid surrogate to better approximate it comparing to existing surrogate functions. However, in practice, the effect of surrogate function is also related to the dataset. We believe that more complex and even dynamic no-linear functions can be used. Such no-linear mapping can be learned via a neural network to automatically fit the flexible and complex mapping for each dataset. We hope that our work will help researchers better understand the fairness surrogate functions. We also hope that we have inspired interested researchers to invent fairness surrogate functions that can automatically fit the data distribution. We leave such interesting exploration for future work.

## E.2    The Insights for Large Language Models

Recently, as the capabilities of Large Language Models (LLMs) have increased, they are remarkably powerful, transcending traditional boundaries in technology and innovation (Achiam et al., 2023; Touvron et al., 2023). These advanced models harness vast amounts of data to understand and generate human-like text, solve complex problems with nuanced insights, and even create content that feels genuinely human-crafted (Liu et al., 2023b). Their applications span from enhancing natural language processing to driving advancements in fields like healthcare (Thirunavukarasu et al., 2023) and education (Kasneci et al., 2023), showcasing a transformative potential for both industry and society at large. While LLMs' fairness has become a focal point of widespread attention (Wang et al., 2023), one may ask how the insights in this paper can be extended to LLMs.

Generally, adapting the conclusions from this paper to large language models (LLMs) involves recognizing the complexity of biases these models can inherit from datasets (Brunet et al., 2019). For LLMs, identifying biases requires sophisticated analysis tools that can understand language use and cultural contexts. To our knowledge, this issue remains an unsolved challenge and has not yet been thoroughly explored (Li et al., 2023; Liu et al., 2023a). Addressing these biases involves not just algorithmic adjustments, but also curating training data, refining model architectures, and implementing continuous feedback loops to identify and mitigate bias. This approach may even emphasize the need for multidisciplinary efforts, combining technical, ethical, and social perspectives to enhance fairness in LLMs. Also, the task considered in this paper is discriminative, while LLM is widely used in generative task. Machine learning focuses on algorithms

learning from data to make predictions or decisions, whereas natural language processing (NLP) involves understanding, interpreting, and generating human language.

Although addressing unfairness in LLMs is crucial but goes a long way, we believe that some insights in this machine learning paper is also beneficial and enlightening for the NLP community. For example, our theoretical analysis highlights the risks of unbounded functions causing instability. It reminds us that when we pre-train LLMs or conduct AI alignment, we may consider some theoretically bounded loss function to mitigate drastic fluctuation during training. Techniques like gradient clipping are also recommended for stability (Mai & Johansson, 2021). Furthermore, our theoretical analysis as well as the experiments also shed light on the advantage of a balanced dataset. Additionally, our findings emphasize the benefits of balanced datasets, suggesting that pre-training language models with data balanced across sensitive attributes may enhance fairness. Also, we may need careful consideration of data proportions in reinforcement learning from human feedback when we align LLM with human values, particularly fairness.

### E.3 Biased Estimation of CP

It serves as a further analysis of CP. But because of the limited space, it does not appear in the main paper.

The original CP is $Cov = \mathbb{E}[(z - \overline{z})d_\theta(\mathbf{x})]$, where $\overline{z}$ is the mean of $z$ over the training set. The expectation can not be computed directly, so its empirical form $\widehat{Cov} = \frac{1}{N}\sum_{i=1}^{N}(z_i - \overline{z})d_\theta(\mathbf{x}_i)$ is proposed to estimate the expectation. We found that $\widehat{Cov}$ mentioned earlier is a biased estimator of $Cov$. The unbiased one is $\frac{1}{N-1}\sum_{i=1}^{N}(z_i - \overline{z})d_\theta(\mathbf{x}_i)$. The proof of it can be found in Appendix E.4. But it does not significantly impacts the results since $N$ is large in the reality.

First, we will give some prerequisites for the proof. The expectation of $z_i d_\theta(\mathbf{x}_j)$ is different for $i$ and $j$:

$$\mathbb{E}(z_i d_\theta(\mathbf{x}_j)) = \begin{cases} \mathbb{E}(z d_\theta(\mathbf{x})) & i = j \\ \mathbb{E}z \cdot \mathbb{E}(d_\theta(\mathbf{x})) & i \neq j \end{cases} \tag{49}$$

Because if $i = j$, each $z_i d_\theta(\mathbf{x}_i)$ is independent of each other, so in this case, $\mathbb{E}(z_i d_\theta(\mathbf{x}_j)) = \mathbb{E}(z d_\theta(\mathbf{x}))$. If $i \neq j$, then $z_i$ and $d_\theta(\mathbf{x}_j)$ are independent, so we derive that $\mathbb{E}(z_i d_\theta(\mathbf{x}_j)) = \mathbb{E}z \cdot \mathbb{E}(d_\theta(\mathbf{x}))$.

It is frequently used in the proof later. With this, now we can start the proof.

### E.4 Proof of Section E.3

*Proof.*

$$\mathbb{E}\left[\frac{1}{N}\sum_{i=1}^{N}(z_i - \overline{z})d_\theta(\mathbf{x}_i)\right]$$

$$= \frac{1}{N}\sum_{i=1}^{N}\mathbb{E}(z_i d_\theta(\mathbf{x}_i)) - \frac{1}{N}\sum_{i=1}^{N}\mathbb{E}[\overline{z}d_\theta(\mathbf{x}_i)]$$

$$= \mathbb{E}(z d_\theta(\mathbf{x})) - \frac{1}{N}\sum_{i=1}^{N}\mathbb{E}[(\frac{1}{N}\sum_{j=1}^{N}z_j)d_\theta(\mathbf{x}_i)]$$

$$= \mathbb{E}(z d_\theta(\mathbf{x})) - \frac{1}{N^2}\sum_{i=1}^{N}\sum_{j=1}^{N}\mathbb{E}[z_j d_\theta(\mathbf{x}_i)]$$

$$= \mathbb{E}(z d_\theta(\mathbf{x})) - \frac{1}{N^2}\sum_{i=1}^{N}\sum_{\substack{j=1 \\ j \neq i}}^{N}\mathbb{E}z_j \cdot \mathbb{E}d_\theta(\mathbf{x}_i) - \frac{1}{N^2}\cdot N \cdot \mathbb{E}[z d_\theta(\mathbf{x})]$$

$$= \mathbb{E}(z d_\theta(\mathbf{x})) - \frac{N-1}{N}\mathbb{E}z \cdot \mathbb{E}d_\theta(\mathbf{x}) - \frac{1}{N}\mathbb{E}[z d_\theta(\mathbf{x})]$$

$$= \frac{N-1}{N} \left[ \mathbb{E}(z d_\theta(\mathbf{x})) - \mathbb{E}z \cdot \mathbb{E}d_\theta(\mathbf{x}) \right]$$

$$= \frac{N-1}{N} Cov. \tag{50}$$

□

Also, we show the risk bound of $Cov$ is $O(\frac{1}{\sqrt{N}})$ below.

**Theorem 13.** *We assume that $D_\theta(\mathbf{x})$ is bounded by $\beta$, i.e., $D_\theta(\mathbf{x}) \leq \beta$. Then for a given constant $\delta > 0$, the inequality below holds:*

$$P\left( \left| Cov - \widehat{Cov} \right| \leq t \right) \geq 1 - \delta,$$

*where $t = \sqrt{2\beta^2 \ln \frac{2}{\delta}} \cdot \sqrt{\frac{N}{(N-1)^2}}$.*

*Proof.*

$$\frac{N}{N-1}(z_i - \bar{z}) d_\theta(\mathbf{x}_i) \in [LB, UB]$$

$$\frac{N}{N-1}(z_i - \bar{z}) d_\theta(\mathbf{x}_i) \geq \frac{N}{N-1}(0 - \bar{z})\beta \geq -\frac{N}{N-1}\beta = LB(Lower\ Bound)$$

$$\frac{N}{N-1}(z_i - \bar{z}) d_\theta(\mathbf{x}_i) \leq \frac{N}{N-1}(1 - \bar{z})\beta \leq \frac{N}{N-1}\beta = UB(Upper\ Bound)$$

According to the Hoeffding inequality,

$$P(\left| \widehat{Cov} - Cov \right| \leq t) \geq 1 - 2\exp\frac{-2Nt^2}{(UB - LB)^2}$$

$$= 1 - 2\exp\frac{-(N-1)^2 t^2}{2N\beta^2}$$

Let $\delta = 2\exp\left( \frac{-(N-1)^2 t^2}{2N\beta^2} \right)$, we get that $t = \sqrt{2\beta^2 \ln \frac{2}{\delta}} \cdot \sqrt{\frac{N}{(N-1)^2}} = O(\frac{1}{\sqrt{N}})$. □

### E.5 When are CP and General Sigmoid Surrogate Close to Each Other?

We show that the number of large margin points determines how close the general sigmoid surrogate is to CP. Under mild conditions, if there are a limited number of large margin points, then the general sigmoid surrogate is close to CP. Our general sigmoid surrogate is designed to address large margin points. However, when these points are scarce or even nonexistent, general sigmoid surrogate becomes close to conventional method like CP.

In the domain of fair machine learning, numerous studies have consistently underscored the pivotal role of data (Mehrabi et al., 2021; Caton & Haas, 2020), with a particular focus on theoretical insights (Chen et al., 2018; Lipton et al., 2018; Dutta et al., 2020). From this perspective, the theorem underscores the importance of data, suggesting that if we can mitigate the impact of these large-margin points either before or during the model training process, it may also contribute to the better accuracy and fairness even we don not use general sigmoid surrogate. This theorem expresses the following vision for data-centric AI (Zha et al., 2023): If we can deal with the data very well, maybe we do not have to use complex or even sophisticated algorithms that requires careful fine-tuning.

**Theorem 14.** *We assume that $k$ points satisfy $G(D_\theta(\mathbf{x})) \in [\zeta, \mu]$ and others satisfy $G(D_\theta(\mathbf{x})) \in [0, \zeta]$, then there holds:*

$$\left| \frac{1}{2} w \cdot Cov(z, d_\theta(\mathbf{x})) - Cov(z, G(d_\theta(\mathbf{x}))) \right| \leq \frac{k}{N} Q(\mu) + (1 - \frac{k}{N}) Q(\zeta), \tag{51}$$

where $Q(x) = \frac{1}{2}wx - G(x)$. The proof is provided in Appendix E.6. The key motivation of Theorem 14 is that the general sigmoid function is close to a linear function when $x$ approaches 0. If $|Cov(z, d_\theta(\mathbf{x}))| \leq \epsilon$, which means that $\left|\frac{1}{2}wCov(z, d_\theta(\mathbf{x}))\right| \leq \frac{1}{2}w\epsilon$. Then we have $|Cov(z, G(d_\theta(\mathbf{x})))| \leq \frac{1}{2}w\epsilon + \frac{k}{N}Q(\mu) + (1 - \frac{k}{N})Q(\zeta)$.

**Remark.** *Taking Figure 2(b) as an example, the result show that $\frac{k}{N} = 5.12\%, \zeta = 2, \mu = 5$. If we choose $w = \frac{1}{2}$, then for the right hand side of (51), we have $\frac{k}{N}Q(\mu) + (1 - \frac{k}{N})Q(\zeta) = 0.038$, which means that the results of bounding the general sigmoid surrogate and bounding CP are close to each other.*

### E.6 Proof of Theorem 14

*Proof.* To start with, notice that

$$\text{sign}(x) = 2\mathbb{1}_{x>0} - 1,$$

where $\text{sign}(x) : \mathbb{R} \to \{-1, 1\}$ returns the sign of $x$. So we can replace $\mathbb{1}_{d_\theta(\mathbf{x})>0}$ with $\text{sign}(d_\theta(\mathbf{x}))$ in (2).

we have

$$
\begin{aligned}
\widehat{DDP}_S &= \frac{N_{1a}}{N_{1a} + N_{0a}} - \frac{N_{1b}}{N_{1b} + N_{0b}} \\
&= \frac{1}{2}\left(\frac{\mathcal{N}_{1a} - \mathcal{N}_{0a}}{N_{1a} + N_{0a}} - \frac{N_{1b} - N_{0b}}{N_{1b} + N_{0b}}\right) \\
&= \frac{1}{2}\left(\frac{\sum_{\mathcal{N}_{1a},\mathcal{N}_{0a}} \text{sign}(d_\theta(\mathbf{x}))}{N_{1a} + N_{0a}} - \frac{\sum_{\mathcal{N}_{1b},\mathcal{N}_{0b}} \text{sign}(d_\theta(\mathbf{x}))}{N_{1b} + N_{0b}}\right) \\
&\propto \frac{\sum_{\mathcal{N}_{1a},\mathcal{N}_{0a}} \text{sign}(d_\theta(\mathbf{x}))}{N_{1a} + N_{0a}} - \frac{\sum_{\mathcal{N}_{1b},\mathcal{N}_{0b}} \text{sign}(d_\theta(\mathbf{x}))}{N_{1b} + N_{0b}},
\end{aligned}
\tag{52}
$$

Furthermore, we can design a new family of surrogates $\phi$ to approximate the sign function instead of the indicator function. They are equivalent to each other, but an odd function is more convenient for the proof. So we use $G(x) = 2\sigma(wx) - 1$ instead of $G(x) = \sigma(wx)$ in our proof.

Before the proof, we first review some properties of general sigmoid surrogate. We notice that we have

$$\lim_{x \to 0} \frac{dG(x)}{dx} = 2wg(wx) = \frac{1}{2}w.$$

where $g(x) = \frac{e^x}{(1+e^x)^2}$. So the theorem is intuitively reasonable because when $x$ approaches 0, the gradient approaches a fixed number, and thus the function itself approaches a linear function. Now we start the proof.

We know that $g(x)$ monotonically increases when $x \leq 0$ and decreases when $x > 0$. So $g(x) \leq g(0) = \frac{1}{4}$. Now we consider a linear function $L(x) = \frac{1}{2}wx$ and let $Q(x) = L(x) - G(x)$ to measure the gap between the two functions. So we have

$$\frac{dQ(x)}{dx} = \frac{1}{2}w - 2wg(wx) \leq 0,$$

which means that $Q(x)$ monotonically decreases in $\mathbb{R}$. Notice that $Q(0) = 0$, so we have $Q(x) \leq 0$ when $x \geq 0$ and $Q(x) \geq 0$ when $x < 0$. And $Q(x)$ is an odd function because $Q(-x) = -Q(x)$. According to these properties of $Q(x)$, we know that $|Q(x)|$ monotonically increases in $[0, +\infty]$.

Then there holds:

$$
\begin{aligned}
&\left|\frac{1}{2}w\frac{1}{N-1}\sum_{i=1}^{N}(z_i - \bar{z})d_\theta(\mathbf{x}_i) - \frac{1}{N-1}\sum_{i=1}^{N}(z_i - \bar{z})G(d_\theta(\mathbf{x}_i))\right| \\
=&\frac{1}{N-1}\left|\sum_{i=1}^{N}(z_i - \bar{z})(\frac{1}{2}wd_\theta(\mathbf{x}_i) - G(d_\theta(\mathbf{x}_i)))\right| \\
\leq&\frac{1}{N-1}\sum_{i=1}^{N}|z_i - \bar{z}|\left|\frac{1}{2}wd_\theta(\mathbf{x}_i) - G(d_\theta(\mathbf{x}_i))\right|
\end{aligned}
$$

$$\leq \frac{1}{N-1} \sum_{i=1}^{N} \left| \frac{1}{2} w d_\theta(\mathbf{x}_i) - G(d_\theta(\mathbf{x}_i)) \right|$$

$$\leq \frac{1}{N-1} \sum_{i=1}^{N} |Q(d_\theta(\mathbf{x}_i))|$$

$$= \frac{1}{N-1} \sum_{i=1}^{N} Q(D_\theta(\mathbf{x}_i))$$

$$\leq \frac{1}{N-1} [kQ(\mu) + (N-k)Q(\zeta)]$$

$$\leq \frac{k}{N} Q(\mu) + (1 - \frac{k}{N})Q(\zeta). \tag{53}$$

The proof is complete.

$\square$

