# OpenReview forum: "Understanding Fairness Surrogate Functions in Algorithmic Fairness"
_TMLR — Accepted by TMLR_

### Review · Reviewer_oKgk · 2024-02-03

**Summary Of Contributions:**

This paper studies the gap between the true fairness metric and the surrogated fairness metric in fair machine learning. The paper specifically studies the covariance proxy and theoretically proves the gap. The paper also identifies two additional issues, namely instability and the large margin point issue. The paper then proposes the general sigmoid surrogates and proves the upper bound of the true fairness. Insights are drawn from the bound. Finally, the paper proposes a balanced surrogate approach by minimizing the gap iteratively during training. Experiments show that the proposed general sigmoid surrogates can achieve better fairness performance. The balanced surrogates, however, may or may not improve fairness across different cases.

**Audience:**

Yes

**Broader Impact Concerns:**

None.

**Claims And Evidence:**

Yes

**Requested Changes:**

What is the difference between $DDP$ and $\hat{DDP}$? Is $\hat{DDP}$ an estimation of $DDP$ based on the data? Please clarify that.

Using the number of citations to demonstrate the importance of a fairness metric is not considered professional in a research paper.

The authors may comment on the influence of the general sigmoid surrogate on the risk bound.

**Strengths And Weaknesses:**

Strength:

The paper rigorously analyzes the surrogate-fairness gap, deriving the upper bound of the gap for the proposed general sigmoid surrogate under certain conditions. It is particularly noteworthy that it further derives a bound under a related condition. The experiments demonstrate the efficacy of the general sigmoid surrogate.

Weakness:

In most theoretical results of the paper, the covariance proxy $\hat{Cov}$ rather than $\tilde{DDP}$ is analyzed. For example, Proposition 1 shows the gap between $\hat{DDP}$ and $\hat{Cov}$, and Theorem 1 shows the upper bound of $\hat{DDP}$ given that $\hat{Cov}$ is bounded. This is not very intuitive since $\tilde{DDP}$ is a direct surrogated approximation to $\hat{DDP}$ and people may be more interested in the gap just caused by applying the surrogate function.

Proposition 1 shows the gap between $\hat{DDP}$ and $\hat{Cov}$. However, the surrogate-fairness gap is a term that also depends on the decision model and the surrogate function. In other words, the surrogate-fairness gap may be correlated with $\hat{Cov}$. This makes the proposition less meaningful as it is not that clear that minimizing $\hat{Cov}$ cannot minimize $\hat{DDP}$. Ideally, we may expect the gap to be independent with $\hat{Cov}$.

In Section 5.2, the authors propose a method for minimizing the difference between $\hat{DDP}$ and $\tilde{DDP}$. I am confused about the purpose of doing that. Since $\tilde{DDP}$ is added as a regularization in the loss function, it will not be minimized during training. Instead, the learning algorithm will try to find a balance between the empirical loss and the fairness regularization. In that case, why do we want to make $\tilde{DDP}$ exactly equal to $\hat{DDP}$? It suffices to use the upper bound of $\hat{DDP}$ as the fairness regularization.

The surrogate function is closely related to the risk bound of the optimization, i.e., whether the true loss can be optimized when the surrogated loss is minimized. The authors may also comment on that regarding the proposed general sigmoid surrogate.

---

> ### Author Response · Authors · 2024-03-08
> **Response to Reviewer oKgk (Part 1)**
>
> Thank you for your constructive comments! We address all your questions and concerns in the following responses. Note that some notations have been changed, which are listed in Table 1 at the beginning of the appendix.
>
> **Q1**: In most theoretical results of the paper, the covariance proxy rather than $\widetilde{DDP}$ is analyzed. People may be more interested in the gap just caused by applying the surrogate function. And ideally, we may expect the gap to be independent with CP.
>
> **A1**: We thank the reviewer for pointing out this issue and we have incorporated this suggestion throughout our paper. In particular, in the revision, **we replace all the theoretical results in the main paper with the analysis of** $\widetilde{DDP}\_{\mathcal{S}}(\phi)$, and then put the analysis of $\widehat{Cov}\_{\mathcal{S}}(\phi)$ in the appendix. We present the Proposition 1, Theorem 1-2 as follows:
>
>
> **Proposition 1.** Define the magnitude of the signed distance by $D\_\theta(\mathbf{x})$, i.e., $D\_\theta(\mathbf{x})=|d\_\theta(\mathbf{x})|$. It satisfies:
> $$
> \begin{align}
>     \underbrace{\widehat{DDP}\_{\mathcal{S}} - \widetilde{DDP}\_{\mathcal{S}}(\phi)}\_{\text{surrogate-fairness gap}} = \frac{\sum\_{(\mathbf{x} ,y) \in \mathcal{N}\_{1a} \cup \mathcal{N}\_{0a}}\left[\mathbb{1}\_{d\_\theta(\mathbf{x})>0}-\phi(d\_\theta(\mathbf{x}))\right]}{N\_{1a}+N\_{0a}}-\frac{\sum\_{(\mathbf{x} ,y) \in \mathcal{N}\_{1b} \cup \mathcal{N}\_{0b}}\left[\mathbb{1}\_{d\_\theta(\mathbf{x})>0}-\phi(d\_\theta(\mathbf{x}))\right]}{N\_{1b}+N\_{0b}}. \notag
> \end{align}
> $$
>
> **Theorem 1.**
> We assume that $G(D\_\theta(\mathbf{x})) \in [1-\gamma, 1]$, where
> $\gamma>0$. $\forall \epsilon > 0$, if $\left| \widetilde{DDP}\_{\mathcal{S}}(G) \right| \le \epsilon$, then it holds:
> $$
> \begin{align}
>     \left| \widehat{DDP}\_{\mathcal{S}} \right| \le \frac{1}{2}\epsilon+\gamma. \notag
> \end{align}
> $$
>
> **Theorem 2.**
> We assume that $k$ points satisfy $G(D\_\theta(\mathbf{x})) \in [0,1-\gamma]$ and others satisfy $G(D\_\theta(\mathbf{x})) \in [1-\gamma, 1]$, where $\gamma>0$. $\forall \epsilon > 0$, if $\left| \widetilde{DDP}\_{\mathcal{S}}(G) \right| \le \epsilon$, then it holds:
> $$
> \begin{align}
>     \left| \widehat{DDP}\_{\mathcal{S}} \right| \le \frac{1}{2} \epsilon+\gamma + \underbrace{\frac{1}{2} \left(\frac{1}{N\_{1a}+N\_{0a}}+\frac{1}{N\_{1b}+N\_{0b}}\right) k}\_{\text{relaxation factor}}. \notag
> \end{align}
> $$
>
> Also, for a clear clarification, when we introduce the covariance proxy, we add the sentence: "**We provide theoretical results for $\widetilde{DDP}\_{\mathcal{S}}(\phi)$ in the main paper, and extend them to $\widehat{Cov}\_{\mathcal{S}}(\phi)$ in the appendix with similar conclusions.**"
>
> For the concern that "we may expect the gap to be independent with CP", according to Appendix A.1 that $\widetilde{DDP}\_{\mathcal{S}}(\phi) \propto \widehat{Cov}\_{\mathcal{S}}(\phi)$, the insights from the theorems apply to both $\widetilde{DDP}\_{\mathcal{S}}(\phi)$ and $\widehat{Cov}\_{\mathcal{S}}(\phi)$. Moreover, the analysis regarding the gap is general and applies to all fairness surrogate functions. **Overall, in this revision, the theoretical results in the main paper are all about $\widetilde{DDP}\_{\mathcal{S}}(\phi)$ and independent of CP.**
>
> **Q2**: What is the difference between $DDP$ and $\widehat{DDP}$? Is $DDP$ an estimation of $\widehat{DDP}$ based on the data?
>
> **A2**: We agree with your understanding. $DDP$ is defined over the whole data distribution. However, we cannot compute it directly. Instead, we can only approximate it using a dataset drawn independently from the data distribution. Therefore, $\widehat{DDP}$ is the estimation of $DDP$ based on the data.

---

> ### Author Response · Authors · 2024-03-08
> **Response to Reviewer oKgk (Part 2)**
>
> **Q3**: In Section 5.2, the authors propose a method for minimizing the difference between $\widehat{DDP}$ and $\widetilde{DDP}$. I am confused about the purpose of doing that. The learning algorithm will try to find a balance between the empirical loss and the fairness regularization, and it may suffice to use the upper bound of $\widehat{DDP}$ as the fairness regularization.
>
> **A3**: Thank you for your valuable comment. We agree with you and believe that previous algorithms with fairness regularization aim to find a balance between the empirical loss and the fairness regularization. However, in this paper, we want to uncover the issue that when we use fairness surrogate functions as the regularization, there will be a gap between the regularization and the true fairness definition. Therefore, we emphasize that **some widely used fairness surrogate functions, such as the unbounded surrogates analyzed in this paper, may not serve as an appropriate regularization because of the fairness-surrogate gap.**
>
> Therefore, focusing on fairness surrogate functions, we provide the balanced surrogate algorithm to reduce the gap, thus trying to help fairness surrogate functions to achieve a better fairness performance.
> - In the experiments, the balanced surrogate method consistently helps the unbounded surrogates (log-sigmoid, linear, and hinge surrogates) to be fairer, while it benefits the bounded surrogates (sigmoid and the general sigmoid surrogate) in most cases. It suggests that **reducing the gap is indeed useful for unbounded surrogates. In this case, it may not suffice to use the unbounded upper bound of** $\widehat{DDP}$.
> - However, when we use bounded surrogates, making the gap infinitesimal may not always be beneficial to fairness. In this case, such a phenomenon may be in line with your concern. Overall, how to strike a balance between the gap and the surrogate function for better fairness is intriguing, but it may be outside the scope of this work. We provide some preliminary discussion in Appendix B.1.
>
> **Q4**: The surrogate function is closely related to the risk bound of the optimization, i.e., whether the true loss can be optimized when the surrogated loss is minimized. The authors may also comment on that regarding the proposed general sigmoid surrogate.
>
> **A4**: Thank you for your suggestion! We agree with you and have incorporated it into Appendix A.5 as follows:
>
> **The surrogate loss function in machine learning.** In machine learning, a surrogate loss function is an auxiliary function used during optimization, which is easier to optimize than the original objective function. Some examples of surrogate loss functions are in Figure 7. Surrogate loss functions are particularly useful when dealing with complex or non-differentiable objective functions. The idea is to approximate the difficult parts of the objective with a surrogate that has more tractable properties (like smoothness or convexity) for optimization algorithms. Here are some examples:
> - Surrogate loss functions are often smooth and convex, even if the original objective is not. This allows for the use of gradient-based optimization techniques, which can speed up the learning process.
> - Use in various algorithms: Surrogate loss functions are used in various machine learning algorithms, including support vector machines (where the hinge loss function acts as a surrogate for the 0-1 loss function), and boosting methods (which build a surrogate to focus on hard examples).
> - Loss approximation: In classification, surrogate loss functions are used to approximate the original loss function. For example, logistic loss and hinge loss are surrogates for the 0-1 loss function in logistic regression and support vector machines, respectively.
>
>
>
> **Surrogate loss functions and fairness surrogate functions.** The foundation of fairness surrogate function is similar to the surrogate loss function: both of them aim to approximate the 0-1 function. When minimizing the loss function, there are two kinds of insights to choose surrogate functions:
> - **Upper bound of the 0-1 function**: Taking surrogate loss functions as an example. As suggested in Figure 7, all of these surrogate loss functions are upper bounds of the 0-1 function. Therefore, when the loss function is minimized to a small value, there is a generalization guarantee for the classifier. Unfortunately, as suggested in Figure 1, the fairness surrogate functions mentioned in existing work are not upper bounds of the 0-1 function.
> - **Estimator of the 0-1 function**: As shown in Figure 1, we aim to better estimate the 0-1 function for fairness. Our empirical results show that our general sigmoid estimation performs fairer results.
> Overall, we believe that the fairness community can be inspired by machine learning and optimization. How to design better fairness surrogate functions is intriguing. Insights from surrogate loss functions may be promising in the fairness community for future work.

---

> ### Author Response · Authors · 2024-03-20
> **Would you mind checking our responses and confirming whether you have any further questions?**
>
> Dear Reviewer oKgk,
>
> Thanks very much for your time and valuable comments.
>
> In the rebuttal period, we have provided detailed responses to all your comments and questions point-by-point for the unclear presentations. Specifically, we provided detailed explanations to further clarify
>
> - The theoretical results in the main paper and their scope. (Q1)
> - The difference between $DDP$ and $\widehat{DDP}$. (Q2)
> - The motivation of the balanced surrogate method and under what circumstances does it work. (Q3)
> - The discussion of the surrogate loss function in machine learning. (Q4)
>
>
> Would you mind checking our responses and confirming whether you have any further questions?
>
> Any comments and discussions are welcome!
>
> Thanks for your attention and best regards.

---

### Review · Reviewer_SXtY · 2024-02-12

**Summary Of Contributions:**

This paper studies the impact of using surrogate functions for fairness. It unveils the *surrogate-fairness gap* phenomenon, showing that common surrogates are increasingly biased as points are classified with larger margins and sensitive groups are imbalanced. The paper introduces a new class of surrogate functions, that are less sensitive to the large-margin issue, together with a procedure to better handle imbalanced distributions.

**Audience:**

Yes

**Broader Impact Concerns:**

No ethical issues are discussed within the paper, and I'm not sure it is relevant to discuss it. Although not critical, it could possibly be mentioned that proper understanding of the surrogate-fairness gap can prevent malicious parties from claiming a method they use is fair, while it only guarantees results on a fairness surrogate.

**Claims And Evidence:**

Yes

**Requested Changes:**

1. It seems that there is a problem in the proof of Theorem 4 (see previous section). Can the author either explain (i) why the current derivation is true, or (ii) fix it?
2. The paper is quite difficult to read, in particular:
   - The three issues, (i) fairness surrogate-gap, (ii) instability and (iii) large margins, are of different nature. It seems that the true problem (which, I believe, is very interesting and informative) studied in the paper is the fairness-surrogate gap. Then, instability and large-margin points are only factors that influence the gap.
   - The proposed methods are not always clearly introduced. For instance, Eq. (2), that introduces a major contribution of the paper, is introduced without discussion and look like a small digression.
   - Notations are not all defined, or are difficult to follow. In particular:
     - Sums like $\sum_{N_{1a}, N_{0a}}$ (e.g. in Eq. (2)) are not properly defined (what is $x$, $x_i$?).
     - DDP surrogate depends on the choice of $\phi$ and on the data, which is not clear from the notation. Additionally, Eq. (5) is not properly defined. Should it be the sum over empirical values $x_1, \dots, x_n$, or population loss?
     - In Appendix, it seems notations $U_{1a}$, $U_{1b}$, $U_{0a}$ and $U_{0b}$ are not defined.
     - Notations A, B, C, D, E, F, G, H in Appendix B.4 are quite confusing, why not use FP etc.? Similarly, the $\bar{prob}_{criterion}$ notation is sometimes confusing.
   - Appendix is difficult to read, with proofs being delayed and delayed between different sections of the appendix.
   - To me, sentences like "To date, according to Google Scholar, this article has garnered over 1,000 citations." sound a bit odd in a research paper.
   - The sentence "So there is no guarantee whether DP is already satisfied even when the empirical Cov is minimized to zero" it not clear: there is some level of guarantee, provided surrogate-fairness gap is small (which we are not able to control).
3. Some related work is missing, in particular regarding hyperparameter tuning and reweighting to handle imbalanced group (see references in Weaknesses section).
4. Regarding empirical evaluation: I would appreciate to see comparison with a fairness-promoting method that is not based on surrogates. Nonetheless, I understand this requires additional work, and this is not so critical regarding acceptance.

**Strengths And Weaknesses:**

*Strengths:*
1. To my knowledge, the study of the surrogate-fairness gap, as well its theoretical study, are new. This gives insights regarding a multitude of empirical observations from previous works.
2. A new class of surrogates is proposed, which allows to fix, to a certain degree, the large-margin issue.
3. An empirical method is then developped to reduce the surrogate-fairness gap, with particular emphasis on group imbalance.
4. Empirically, it is demonstrated that the proposed method results in improved fairness with similar accuracy against other fairness-surrogate-based methods.

*Weaknesses:*
1. The proposed approach can require significant additional hyperparameter tuning (need of tuning two more hyperparameters, $w$ and $\lambda$). It is not so clear how this would interfere with the usual hyperparameter tuning, that can itself promote fairness (see e.g., [1-2]).
2. The "instability issue" is not so convincing. In particular, using bounded surrogates that make the empirical covariance bounded in [0, 1] only partly fix the problem: bounding a difference of probability by a value between 0 and 1 is not always informative.
3. Empirical evaluation of the method focuses on methods that use fairness surrogate. While it is true that the proposed method compares positively to fairness-promoting methods of the same type, many alternatives exist. It would be appreciated if authors can briefly compare with one or two of these methods (e.g., reweighting-based methods, that, similarly to the approach proposed in 5.2, reweight samples based their sensitive attribute, see [3-4] and references therein).
4. The paper is not always very clear, and clarity of exposition could be improved (see Requested Changes section).

*Question for mathematical derivation:*
- In the proof of Theorem 4, after the sentence "Finally, because of the fact that": samples are drawn independently so covariance is indeed zero, but the expectation of the product becomes sum of expectations, which seems to be an error. I believe similar (although a bit different) results can be achieved even with fixing this error, but it should be fixed.

*References:*

[1] Cruz, André F., et al. "Promoting fairness through hyperparameter optimization." 2021 IEEE International Conference on Data Mining (ICDM). IEEE, 2021.

[2] Tizpaz-Niari, Saeid, et al. "Fairness-aware configuration of machine learning libraries." Proceedings of the 44th International Conference on Software Engineering. 2022.

[3] Krasanakis, Emmanouil, et al. "Adaptive sensitive reweighting to mitigate bias in fairness-aware classification." Proceedings of the 2018 world wide web conference. 2018.

[4] Maheshwari, Gaurav, and Perrot, Michaël, FairGrad: Fairness Aware Gradient Descent, TMLR 2023.

---

> ### Author Response · Authors · 2024-03-08
> **Response to Reviewer SXtY (Part 1)**
>
> Thank you for your constructive comments! We address all your questions and concerns in the following responses. Note that some notations have been changed, which are listed in Table 1 at the beginning of the appendix.
>
> **Q1**: The errors in the proof of Theorem 4.
>
> **A1**: Thank you very much for your careful examination. We have fixed this error in the revision, and the derivation becomes:
> $$
> \begin{align}
> & \mathbb{E}\left(\sum\_{\mathcal{N}\_{1a},\mathcal{N}\_{0a}}\mathbb{1}\_{d\_\theta(\mathbf{x})>0} \quad \cdot \sum\_{\mathcal{N}\_{1b},\mathcal{N}\_{0b}}\mathbb{1}\_{d\_\theta(\mathbf{x})>0}\right)
> \notag \\\ = & \mathbb{E}\left( \sum\_{\mathcal{N}\_{1a},\mathcal{N}\_{0a}}\mathbb{1}\_{d\_\theta(\mathbf{x})>0} \right) \cdot \mathbb{E}\left( \sum\_{\mathcal{N}\_{1b},\mathcal{N}\_{0b}}\mathbb{1}\_{d\_\theta(\mathbf{x})>0} \right) + Cov\left(\sum\_{\mathcal{N}\_{1a},\mathcal{N}\_{0a}}\mathbb{1}\_{d\_\theta(\mathbf{x})>0}\quad , \quad\sum\_{\mathcal{N}\_{1b},\mathcal{N}\_{0b}}\mathbb{1}\_{d\_\theta(\mathbf{x})>0}\right)
> \notag \\\ = & (N\_{1a}+N\_{0a})P(d\_\theta(\mathbf{x})>0 | z=+1) \cdot (N\_{1b}+N\_{0b})P(d\_\theta(\mathbf{x})>0 | z=-1)  + \notag \\\ & \sum\_{\mathcal{N}\_{1a},\mathcal{N}\_{0a}} \sum\_{\mathcal{N}\_{1b},\mathcal{N}\_{0b}} \underbrace{Cov\left(\mathbb{1}\_{d\_\theta(\mathbf{x})>0|z=+1}, \mathbb{1}\_{d\_\theta(\mathbf{x})>0|z=-1}\right)}\_{0}
> \notag \\\ = & (N\_{1a}+N\_{0a})P(d\_\theta(\mathbf{x})>0 | z=+1) \cdot (N\_{1b}+N\_{0b})P(d\_\theta(\mathbf{x})>0 | z=-1). \notag
> \end{align}
> $$
>
> So $\mathbb{E}(T\_3)$ becomes $2P\_aP\_b$, and we have:
> $$
> \begin{align}
>     Var\left(\widehat{DDP}\_{\mathcal{S}}\right) & = \mathbb{E}(T\_1) + \mathbb{E}(T\_2) - \mathbb{E}(T\_3) - (DDP)^2 \notag \\\ & = \underbrace{\left(1-\frac{1}{N\_a}\right)P\_a^2 + \frac{1}{N\_a}P\_a}\_{\mathbb{E}(T\_1)} + \underbrace{\left(1-\frac{1}{N\_b}\right)P\_b^2 + \frac{1}{N\_b}P\_b}\_{\mathbb{E}(T\_2)} - \underbrace{2P\_aP\_b}\_{\mathbb{E}(T\_3)} - \underbrace{(P\_a-P\_b)^2}\_{DDP^2} \notag \\\ & = \frac{1}{4}\left( \frac{1}{N\_a} + \frac{1}{N\_b} \right) - \frac{1}{N\_a}(P\_a-\frac{1}{2})^2 - \frac{1}{N\_b}(P\_b-\frac{1}{2})^2 \tag{$P\_a+P\_b=1$} \\\ & = \frac{1}{4}\left( \frac{1}{N\_a} + \frac{1}{N\_b} \right) - \left( \frac{1}{N\_a}+\frac{1}{N\_b} \right)(P\_a-\frac{1}{2})^2 \notag
> \end{align}
> $$
> For the maximum variance, $P\_a=P(d\_\theta(\mathbf{x})>0 | z=+1)$ takes the value $\frac{1}{2}$, and the maximum variance is $\frac{1}{4}\left( \frac{1}{N\_a} + \frac{1}{N\_b} \right)$. Therefore, Theorem 4 becomes:
>
> **Theorem 4.** Let $N\_a = N\_{1a}+N\_{0a}, N\_b = N\_{1b}+N\_{0b}, P\_a = P(d\_\theta(\mathbf{x})>0 | z=+1), P\_b = P(d\_\theta(\mathbf{x})>0 | z=-1)$. We assume that each dataset consisting of $N$ points is independently drawn from a dataset distribution. The variance is computed over the dataset distribution. Then there holds:
>
> $$
> \begin{align}
>     Var\left(\widehat{DDP}\_{\mathcal{S}}\right) \le \frac{1}{4}\left( \frac{1}{N\_a} + \frac{1}{N\_b} \right). \notag
> \end{align}
> $$
>
> And as you might expect, the corresponding theoretical results are indeed similar: Theorem 4 shows that $Var\left(\widehat{DDP}\right) \le \frac{1}{4}\left( \frac{1}{N\_a} + \frac{1}{N\_b} \right)$. Therefore, **it also highlights the advantage of a balanced dataset in reducing variance**.
>
>
> **Q2**: The paper is not always very clear, and clarity of exposition could be improved: The three issues, (i) fairness surrogate-gap, (ii) instability and (iii) large margins, are of different natures. It seems that the true problem studied in the paper is the fairness-surrogate gap. Then, instability and large-margin points are the only factors that influence the gap.
> Also, Eq. (2), that introduces a major contribution of the paper, is introduced without discussion and looks like a small digression.
>
> **A2**: Thank you for your suggestion! We make the following changes:
> - Firstly, **we change the title of Section 4 to "The Surrogate-fairness Gap"** to underline the importance of it.
> - Furthermore, in order to describe the three issues more logically and clearly, **we clarify the relationship among them at the beginning of Section 4** as follows: "We first emphasize the importance of surrogate fairness gap, and then point out instability and large margin points, which are two issues influencing the 'gap'."
> - Finally, to enrich the discussion on the fairness-surrogate gap and facilitate the understanding for the reader, **we provide a comparative analysis of various fairness surrogate functions in Section 4**: "In particular, in practice, Figure 1(b) shows the fairness-surrogate gap of different fairness surrogate functions, including CP (which is equivalent to linear surrogate function), hinge, log-sigmoid as well as sigmoid, and our general sigmoid surrogate. It suggests that unbounded surrogate functions tend to exhibit a larger surrogate-fairness gap. The bounded ones, such as sigmoid and general sigmoid, both exhibit a bounded 'gap'."

---

> ### Author Response · Authors · 2024-03-08
> **Response to Reviewer SXtY (Part 2)**
>
> **Q3**: Notations are not all defined, or are difficult to follow.
>
> **A3**: Thank you very much for finding these issues! First of all, we summarize the frequently used notations in a table at the beginning of the appendix. Then, in the revision, we have addressed the unproper statements as well as some notations you have mentioned. In particular, we clarify the notations and definitions as outlined below:
>
> - For the definitions of sums like $\sum\_{N\_{1 a}, N\_{0 a}}$, we agree with you and replace the sums like $\sum\_{\mathcal{N}\_{1a},\mathcal{N}\_{0a}}\phi(d\_\theta(\mathbf{x}))$ with $\sum\_{(\mathbf{x} ,y) \in \mathcal{N}\_{1a} \cup \mathcal{N}\_{0a}}\phi(d\_\theta(\mathbf{x}))$ to make them properly defined.
> - For the definition of $\min L(\theta,\mathbf{x},y) + \lambda \cdot \widetilde{DDP}$, we have incorporated detailed and clear explanation for it. In particular, $L(\theta,\mathbf{x},y)=\frac{1}{N} \sum\_{(\mathbf{x},y) \in \mathcal{S}} \ell(\theta,\mathbf{x},y)$ is the empirical loss over the training set $\mathcal{S}$, and $\ell$ is a convex loss function.
> - For the notations of DDP surrogate, in order to show that it is related to $\phi$ and data, we replace $\widetilde{DDP}(\phi)$ with $\widetilde{DDP}\_{\mathcal{S}}(\phi)$ in the paper. For the sake of rigor, we also replace $\widehat{Cov}(\phi)$ with $\widehat{Cov}\_{\mathcal{S}}(\phi)$ and replace $\widehat{DDP}$ with $\widehat{DDP}\_{\mathcal{S}}$.
> - For the definitions of $U\_{1a}, U\_{1b}, U\_{0a}, U\_{0b}$, it is a typo we have overlooked in the appendix. The definition is: the number of points satisfying $\phi(D\_\theta(\mathbf{x})) \in [0,1-\gamma]$ in the four groups are denoted as $U\_{1a}, U\_{1b}, U\_{0a}, U\_{0b}$, respectively. We have fixed it in Appendix A.4 in the revision.
> - For the notation like $\overline{prob}\_{z=+1, y=-1}$, we replace it with $\hat{P}(d\_\theta(\mathbf{x})<0 | z=+1)$ to indicate the prediction probability. Please refer to Appendix D and F in the revision.
> - For the notations A, B, C, D, E, F, G, H in the appendix, we agree with you and have incorporated your suggestion throughout our paper. Specifically, we replace them with notations like $FP\_+$, which indicate the false positive prediction with the positive sensitive attribute $z=+1$. Please refer to Appendix D in the revision.
>
>
> **Q4**: The appendix is difficult to read, with proofs being delayed and delayed between different sections of the appendix.
>
> **A4**: To better clarify Section D "Extension to Other Fairness Definitions" in the appendix, we have repositioned the related sections to prevent the proof from being repeatedly delayed. We also provide an outline at the beginning of Appendix D to discuss the logical relationship between them:
>
> "To begin with, in Section D.1, we review the considered fairness definitions: disparate mistreatment, and balance for positive (negative) class. After that, in Section D.2, we provide a deeper understanding of CP for disparate mistreatment. The proof of the statements is in Section D.3. Finally, in Section D.4, we provide a deeper understanding of CP for balance for positive (negative) class. The corresponding proof is in Section D.5."
>
> Moreover, for the convenience of readers to understand the appendix, we further elaborate on some explanations of proofs from Appendix A.1 to A.4 in the revision, with a proof sketch at the beginning of every subsection. For example:
> - In Appendix A.3: "Proof of Theorem 1", we provide the idea of proof as follows: "Firstly, absolute value inequality is used to derive the upper bound of $\left|\widehat{DDP}\_{\mathcal{S}}\right|$ given the upper bound of $\left|\widetilde{DDP}\_{\mathcal{S}}(\phi)\right|$. Secondly, the assumption $G(D\_\theta(\mathbf{x})) \in [1-\gamma, 1]$ is used to derive a tighter bound of $\left|\widehat{DDP}\_{\mathcal{S}}\right|$ to complete the proof."
> - In Appendix A.4: "Proof of Theorem 2", we provide the proof sketch as follows: "The difference between Theorem 1 and Theorem 2 mainly lies in the assumption of $D\_\theta(\mathbf{x})$. Therefore, Equation (19) can be also used here because it is independent of the assumption. Consequently, the task of the proof is providing a tighter bound than Equation (19)."

---

> ### Author Response · Authors · 2024-03-08
> **Response to Reviewer SXtY (Part 3)**
>
> **Q5**: Some related work is missing, in particular regarding hyperparameter tuning and reweighting to handle imbalanced group (see references in the weaknesses section).
>
> **A5**: Thank you for your suggestion! We have incorporated further related work in Appendix C.1 because of the space limit in the main paper. In particular, we include other in-processing methods such as reweighting, hyperparameter tuning, fair sample selection, and fair adversarial learning as follows:
>
> Our paper focuses on fairness surrogate functions from the in-processing methods. In addition to the aforementioned fairness constraints [1-2] and fairness regularization [3], there are also various kinds of fairness-aware in-processing methods in the community. For example, fair adversarial learning [4-5], fair reweighing [6-9], fair sample selection [10-11], and hyper-parameter optimization [12-13]. To gain a comprehensive understanding of fairness-aware algorithms in machine learning, thorough surveys are conducted in [14-18], providing a broader view of these algorithms.
>
> [1] Muhammad Bilal Zafar, Isabel Valera, Manuel Gomez Rodriguez, and Krishna P. Gummadi. Fairness beyond disparate treatment & disparate impact. Proceedings of the 26th International Conference on World Wide Web, 2017a.
>
> [2] Muhammad Bilal Zafar, Isabel Valera, Manuel Gomez Rogriguez, and Krishna P. Gummadi. Fairness constraints: Mechanisms for fair classification. In Proceedings of the 20th International Conference on Artificial Intelligence and Statistics, 2017c.
>
> [3] Harry Bendekgey and Erik B. Sudderth. Scalable and stable surrogates for flexible classifiers with fairness constraints. In Advances in Neural Information Processing Systems, 2021.
>
> [4] Brian Hu Zhang, Blake Lemoine, and Margaret Mitchell. Mitigating unwanted biases with adversarial learning. In Proceedings of the 2018 AAAI/ACM Conference on AI, Ethics, and Society, pp. 335–340, 2018.
>
> [5] David Madras, Elliot Creager, Toniann Pitassi, and Richard Zemel. Learning adversarially fair and transferable representations. In International Conference on Machine Learning, pp. 3384–3393, 2018.
>
> [6] Emmanouil Krasanakis, Eleftherios Spyromitros-Xioufis, Symeon Papadopoulos, and Yiannis Kompatsiaris. Adaptive sensitive reweighting to mitigate bias in fairness-aware classification. In Proceedings of the 2018 World Wide Web conference, pp. 853–862, 2018.
>
> [7] Preethi Lahoti, Alex Beutel, Jilin Chen, Kang Lee, Flavien Prost, Nithum Thain, Xuezhi Wang, and Ed Chi. Fairness without demographics through adversarially reweighted learning. Advances in neural information processing systems, 33:728–740, 2020.
>
> [8] Junyi Chai and Xiaoqian Wang. Fairness with adaptive weights. In International Conference on Machine Learning, pp. 2853–2866, 2022.
>
> [9] Gaurav Maheshwari and Michaël Perrot. Fairgrad: Fairness-aware gradient descent. Transactions on Machine Learning Research, 2023.
>
> [10] Yuji Roh, Kangwook Lee, Steven Whang, and Changho Suh. Sample selection for fair and robust training. Advances in Neural Information Processing Systems, 34:815–827, 2021a.
>
> [11] Yuji Roh, Kangwook Lee, Steven Euijong Whang, and Changho Suh. Fairbatch: Batch selection for model fairness. In International Conference on Learning Representations, 2021b.
>
> [12] André F Cruz, Pedro Saleiro, Catarina Belém, Carlos Soares, and Pedro Bizarro. Promoting fairness through hyperparameter optimization. In 2021 IEEE International Conference on Data Mining (ICDM), pp. 10361041, 2021.
>
> [13] Saeid Tizpaz-Niari, Ashish Kumar, Gang Tan, and Ashutosh Trivedi. Fairness-aware configuration of machine learning libraries. In Proceedings of the 44th International Conference on Software Engineering, pp. 909920, 2022.
>
> [14] Ninareh Mehrabi, Fred Morstatter, Nripsuta Saxena, Kristina Lerman, and Aram Galstyan. A survey on bias and fairness in machine learning. ACM Computing Surveys, 54:1–35, 2021.
>
> [15] Simon Caton and Christian Haas. Fairness in machine learning: A survey. arXiv preprint arXiv:2010.04053, 2020.
>
> [16] Max Hort, Zhenpeng Chen, Jie M Zhang, Federica Sarro, and Mark Harman. Bia mitigation for machine learning classifiers: A comprehensive survey. arXiv preprint arXiv:2207.07068, 2022.
>
> [17] Mingyang Wan, Daochen Zha, Ninghao Liu, and Na Zou. In-processing modeling techniques for machine learning fairness: A survey. ACM Transactions on Knowledge Discovery from Data, 17(3):1–27, 2023.
>
> [18] Dana Pessach and Erez Shmueli. Algorithmic fairness. Machine Learning for Data Science Handbook: Data Mining and Knowledge Discovery Handbook, pp. 867–886, 2023.

---

> ### Author Response · Authors · 2024-03-08
> **Response to Reviewer SXtY (Part 4)**
>
> **Q6**: Regarding empirical evaluation: I would appreciate to see a comparison with a fairness-promoting method that is not based on surrogates.
>
> **A6**: Thank you very much for pointing out this issue. We add two in-processing methods in the experiments: adaptive sensitive reweighting [1] and reduction [2], which are both not based on fairness surrogate functions. The results are in Appendix B.6.
> - In the Bank dataset, our general sigmoid surrogate and balanced surrogate approaches consistently achieve much better fairness performance than 'reduction' and 'ASR' while maintaining comparable accuracy.
> - In the COMPAS dataset, our approaches always achieve better fairness performance than others while maintaining comparable accuracy.
> - In the Adult dataset, our methods achieve better fairness than 'ASR' while maintaining comparable accuracy.
>
> Overall, the results suggest that our methods are very competitive among these in-processing methods. For a fair comparison, we spend much time conducting careful hyperparameter tuning for these two methods. Due to the limited time in this response period, we just added these two algorithms. We leave the comparative study between our methods and other in-processing methods for future work. We hope that our supplementary experiment helps people better understand the advantages of our approach.
>
>
> [1] Emmanouil Krasanakis, Eleftherios Spyromitros-Xioufis, Symeon Papadopoulos, and Yiannis Kompatsiaris. Adaptive sensitive reweighting to mitigate bias in fairness-aware classification. In Proceedings of the 2018 World Wide Web conference, pp. 853–862, 2018.
>
> [2] Alekh Agarwal, Alina Beygelzimer, Miroslav Dudik, John Langford, and Hanna Wallach. A reductions approach to fair classification. In Proceedings of the 35th International Conference on Machine Learning, 2018.
>
>
>
>
> **Q7**: The proposed approach can require significant additional hyperparameter tuning (need of tuning two more hyperparameters, $w$ and $\gamma$). It is not so clear how this would interfere with the usual hyperparameter tuning, that can itself promote fairness.
>
> **A7**: Thank you for your suggestion. Due to the limited time, we provide a discussion about the hyperparameters in our methods, and we leave the related experimental study for future work. Indeed, our approaches do require some additional hyperparameter tuning:
> - For $w$ in general sigmoid, as shown in our experimental settings in the appendix, it is selected from {1, 2, 4, 8, 16} according to the fairness performance on the validation set. Thus, it requires selecting an approximate range for it.
> - For the regularization coefficient $\gamma$, it is conventional in machine learning to adjust it. Our experiments also require selecting an approximate range for it.
> - The smoothing factor $\alpha$, and termination threshold $\eta$ are hyperparameters, but we fix them and never change them throughout our experiments.
> - For $\gamma$ in the balanced surrogate approach, it does not require any hyperparameter tuning because it is automatically specified in the algorithm.
>
> As a result, in our experiments, we only consider tuning two hyperparameters: $w$ and $\gamma$. And some hyperparameter optimization tools (such as [1]) may be useful to aid the cost of hyperparameter tuning.
> The above discussion is provided in Appendix B.3 in the revision.
>
> [1] https://github.com/hyperopt/hyperopt
>
>
> **Q8**: The "instability issue" is not so convincing. In particular, using bounded surrogates that make the empirical covariance bounded in [0, 1] only partly fixes the problem: bounding a difference of probability by a value between 0 and 1 is not always informative.
>
> **A8**: Thank you for the comment. To better express the contribution of our "instability" part, we provide the analysis of $\widetilde{DDP}\_{\mathcal{S}}$ instead of CP in the revision, and we believe that it may be more convincing than our previous submission.
>
> The variance of $\widetilde{DDP}\_{\mathcal{S}} \left( \phi \right)$ is $Var \left(\widetilde{DDP}\_{\mathcal{S}}(\phi) \right) = \mathbb{E}\left(\widetilde{DDP}\_{\mathcal{S}}(\phi)\right)^2 - \left[\mathbb{E}\left(\widetilde{DDP}\_{\mathcal{S}}(\phi)\right)\right]^2$.
> - If we choose bounded surrogate $\phi(x) \in [0,1]$, then $\widetilde{DDP}\_{\mathcal{S}}(\phi) \in [-1,1]$, which means that $Var\left(\widetilde{DDP}\_{\mathcal{S}}(\phi)\right) \in [0,1]$.
> Therefore, there is a stability guarantee for $\widetilde{DDP}\_{\mathcal{S}}(\phi)$ if $\phi(x) \in [0,1]$.
> - However, if we choose an unbounded surrogate function (such as $\phi(x)=x \in [-\infty, +\infty]$ for the original CP), $\phi(x)$ is not constrained within the range $[0,1]$. Therefore, we cannot conclude that $\widetilde{DDP}\_{\mathcal{S}}(\phi) \in [-1,1]$. Consequently, we also cannot conclude that $Var\left(\widetilde{DDP}\_{\mathcal{S}}(\phi)\right) \in [0,1]$.
> As a result, there is no longer a stability guarantee for $\widetilde{DDP}\_{\mathcal{S}}(\phi)$.

---

> ### Author Response · Authors · 2024-03-08
> **Response to Reviewer SXtY (Part 5)**
>
> **Q9**: No ethical issues are discussed within the paper.
>
> **A9**: Thank you for your valuable comment, we have added a discussion about it: "Broader Impact and Ethics Statement" in Section 8 in the revision:
>
> This study concentrates on better understanding the fairness surrogate functions in machine learning. Importantly, if someone claims the fairness guarantee of using unbounded fairness surrogate functions, it is worthy of suspicion and further investigation because of the fairness-surrogate gap issue discussed in this paper. Furthermore, the motivation of our general sigmoid surrogate and balanced surrogate methods are both centered on improving the fairness performance.
>
> We acknowledge the sensitive nature of our study and guarantee adherence to all applicable legal and ethical standards. Our research is conducted within a safe and controlled setting to protect real-world systems' security. Only researchers who have received the appropriate clearance can access the most confidential parts of our experiments. Such measures are implemented to preserve the integrity of our research and to reduce any potential risks associated with the experiments.

---

> ### Author Response · Authors · 2024-03-20
> **Would you mind checking our responses and confirming whether you have any further questions?**
>
> Dear Reviewer SXtY,
>
> Thanks very much for your time and valuable comments.
>
> In the rebuttal period, we have provided detailed responses to all your comments and questions point-by-point for the unclear presentations. Specifically, we provided detailed explanations to further clarify
>
> - The errors in the proof of Theorem 4. (Q1)
> - The improved clarity of the main paper. (Q2)
> - The revised notations and definitions. (Q3)
> - The improved clarity of the appendix. (Q4)
> - Further discussion of related work. (Q5)
> - The comparison with two other fairness-promoting methods that is not based on surrogates. (Q6)
> - The discussion of hyperparameter tuning. (Q7)
> - The discussion of the "instability issue." (Q8)
> - The section "Broader Impact and Ethics Statement". (Q9)
>
> Would you mind checking our responses and confirming whether you have any further questions?
>
> Any comments and discussions are welcome!
>
> Thanks for your attention and best regards.

---

> > ### Comment · Reviewer_SXtY · 2024-03-28
> > **All my concerns have been addressed**
> >
> > Dear authors,
> >
> > Thank you very much for this very detailed answer, which answer all my concerns.
> > Best,
> > Reviewser SXtY

---

> > > ### Author Response · Authors · 2024-03-28
> > > **Thanks again for your constructive comments**
> > >
> > > Thanks very much for your time. Any comments and discussions are welcome!

---

### Review · Reviewer_AFbU · 2024-02-25

**Summary Of Contributions:**

Previous studies have revealed that machine learning algorithms can generate biased outcomes for certain population groups, particularly those identified by sensitive features. To tackle this problem, some research has introduced the concept of demographic parity and developed algorithms that aim to balance the learning goals with fairness constraints. However, these approaches can sometimes lead to outcomes that are still biased. This paper demonstrates a significant issue: the existence of a gap between the intended fairness definitions and the surrogate functions used to approximate fairness. As a result, current algorithms might not effectively minimize disparities among different groups. Moreover, this paper highlights the negative effects of using unbounded surrogate functions on the stability of these solutions. To overcome these challenges, the paper introduces two innovative approaches: a theoretically grounded surrogate function for fairness, termed the general sigmoid, which comes with upper bounds for demographic parity and variance, and a new algorithm known as the balanced surrogate. This algorithm is designed to iteratively narrow the fairness gap throughout the training process. Comprehensive experiments validate the efficiency of the proposed methods.

**Audience:**

Yes

**Claims And Evidence:**

Yes

**Requested Changes:**

1. I believe the figure could benefit from being slightly enlarged to enhance its clarity.

2. The conclusions of this paper can be adapted to explore fairness concerns within large language models, offering insights on identifying and addressing biases in these systems. Authors are encouraged to discuss it.

3.  In section 4.1, we need some intuition and clarifications to understand how to build the surrogate-fairness gap between DP and CP.

4. Could we use other non-linear functions to replace with sigmoid in section 5.1?

**Strengths And Weaknesses:**

Strengths

1. The writing in this paper is very clear, making it easy to understand.

2. The motivation behind this study is well-defined. It addresses how current algorithms tackle fairness issues by employing the conventional notion of demographic parity in a constrained optimization framework. The paper provides a theoretical critique of these methods, pointing out their tendency to produce unfair solutions. As a response, it introduces a theoretically solid surrogate function for fairness and develops an algorithm that leverages this function. The theoretical foundation for the motivation is robust, and the efficiency of the proposed algorithm is convincingly demonstrated.

3. The experimental results validate the method's efficacy and establish a solid groundwork for future research.

Weaknesses

1. Although this paper aims to address the surrogate-fairness gap and variance issues with demographic parity, I think more discussion is needed for other fairness criteria to enlarge the impact of this paper.

2. The conditions required for the theoretical analysis should be combined with practical cases. I think this paper needs more discussion of the real-world scenarios.

---

> ### Author Response · Authors · 2024-03-08
> **Response to Reviewer AFbU (Part 1)**
>
> Thank you for your constructive comments! We address all your questions and concerns in the following responses. Note that some notations have been changed, which are listed in Table 1 at the beginning of the appendix.
>
> **Q1**: Although this paper aims to address the surrogate-fairness gap and variance issues with demographic parity, I think more discussion is needed for other fairness criteria to enlarge the impact of this paper.
>
> **A1**: Thank you for your valuable suggestion. We only discuss DP in the main paper, and the experiments are all for DP. The theoretical analysis in this paper is extended to **Disparate Mistreatment** (including three fairness definitions) and **Balance for Positive/Negative Class** (including two fairness definitions) in Appendix D.2 and D.4.
> - In Appendix D.2, we extend Theorem 1 in the main paper to disparate mistreatment. Furthermore, no matter what the definition of disparate mistreatment is, $S$ defaults to the whole training set in [1]. But we contend that it does not match their expected fairness definitions. The authors correct it in [2], so our proof can also serve as an explanation of the correction. Our proof is in Appendix D.3.
> - In Appendix D.4, we deal with balance for positive/negative class. These two fairness definitions are prepared for probabilistic classifiers. Many probabilistic classifiers do not model the decision boundary directly, so it is hard to relate the proxy to these two definitions. To solve this problem, we creatively link $d_\theta(\mathbf{x})$ and the prediction probability $\hat{P}(d_\theta(\mathbf{x})>0 | z=+1)$ together using an assumption. To the best of our knowledge, we are the first to connect the proxy to these two definitions. Please refer to Appendix D.5 for proof details. We aim to provide the basis for future work about how to choose a suitable $\phi$ to approximate $f$, thereby inspiring the research community. We believe that it can be an interesting direction of future work for these fairness definitions based on predicted probability.
>
> Overall, as the reviewer suggests, there are various fairness definitions. And we try to cover as many definitions as possible. As far as we know, our method mainly applies to prediction-based fairness definitions. In particular, our focus is DP, with theory and experiments. And we cover the other five fairness definitions with theoretical analysis in the appendix. How to extend our insights and algorithms from prediction-based definitions to other definitions is worth exploring, and we may leave this intriguing question for future work.
>
> [1] Muhammad Bilal Zafar, Isabel Valera, Manuel Gomez Rodriguez, and Krishna P. Gummadi. Fairness beyond disparate treatment & disparate impact. Proceedings of the 26th International Conference on World Wide Web, 2017a.
>
> [2] Muhammad Bilal Zafar, Isabel Valera, Manuel Gomez-Rodriguez, and Krishna P. Gummadi. Fairness constraints: A flexible approach for fair classification. Journal of Machine Learning Research, 2019.
>
> **Q2**: The conditions required for the theoretical analysis should be combined with practical cases. I think this paper needs more discussion of the real-world scenarios.
>
> **A2**: Thank you for your suggestion. We focus on the machine learning scenario in this paper, and it is worth exploring in other practical cases like computer vision and natural language processing. Specifically, our Theorems set a foundation for understanding how different data characteristics can impact the fairness of outcomes. The conditions outlined in these theorems, particularly around large margin points, highlight how the proximity of data points to the decision boundary can influence the model's fairness. Here are some examples:
> - In computer vision, if a facial recognition system consistently has large margin points for faces from certain demographics, it may indicate that the system is overly confident in its ability to recognize these faces, potentially at the expense of recognizing faces from other demographics with the same level of accuracy.
> - In natural language processing, these theorems could also be applied to scenarios such as sentiment analysis. For example, in sentiment analysis, the insights from the theorems suggest that a balanced dataset can lead to more equitable outcomes. Specifically, when dealing with high-dimensional data like text, ensuring that the classifier does not overly penalize or benefit certain groups (represented by the large margin points condition) becomes crucial.
>
> In summary, the theoretical analysis provided by Theorems 1 and 2 offers a crucial lens through which to examine and address fairness concerns in practical applications. By understanding the role of large margin points and their impact on algorithmic fairness, practitioners can take targeted actions to improve the fairness of AI systems. And it requires further understanding for future researchers in this field.

---

> ### Author Response · Authors · 2024-03-08
> **Response to Reviewer AFbU (Part 2)**
>
> **Q3**: I believe the figure could benefit from being slightly enlarged to enhance its clarity.
>
> **A3**: Thank you for pointing out this issue. In the revision, we have enlarged all the figures for better clarification.
>
> **Q4**: The conclusions of this paper can be adapted to explore fairness concerns within large language models, offering insights on identifying and addressing biases in these systems. Authors are encouraged to discuss it.
>
> **A4**: Thank you for your constructive comments. Generally, adapting the conclusions from this paper to LLMs involves recognizing the complexity of biases these models can inherit from datasets.
> - For LLMs, identifying biases requires sophisticated analysis tools that can understand language use and cultural contexts. To our knowledge, this issue has not yet been thoroughly explored.
> - Addressing these biases involves not just algorithmic adjustments, but also curating training data, refining model architectures, and implementing continuous feedback loops to identify and mitigate bias.
> - This approach may even emphasize the need for multidisciplinary efforts, combining technical, ethical, and social perspectives to enhance fairness in LLMs.
> - Also, the task considered in this paper is discriminative, while LLM is widely used in generative tasks. Machine learning focuses on algorithms learning from data to make predictions or decisions, whereas natural language processing (NLP) involves understanding, interpreting, and generating human language.
>
> Although addressing unfairness in LLMs is crucial but goes a long way, we believe that some insights in this machine learning paper are also beneficial and enlightening for the NLP community.
> - For example, our theoretical analysis highlights the risks of unbounded functions causing instability. It reminds us that when we pre-train LLMs or conduct AI alignment, **we may consider some theoretically bounded loss function to mitigate drastic fluctuation during training**. Techniques like gradient clipping are also recommended for stability.
> - Furthermore, our theoretical analysis as well as the experiments also shed light on the advantage of a balanced dataset. Additionally, our findings emphasize the benefits of balanced datasets, suggesting that **pre-training language models with data balanced across sensitive attributes may enhance fairness.** Also, we may need careful consideration of data proportions in reinforcement learning from human feedback when we align LLM with human values, particularly fairness.
>
> In Appendix E.2, we have incorporated such a discussion about how to extend the ideas from this work to large language models. We believe that many insights in this work can benefit the NLP community.
>
> **Q5**: In section 4.1, we need some intuition and clarifications to understand how to build the surrogate-fairness gap between DP and CP.
>
> **A5**: We thank the reviewer for pointing out this issue. To describe it more clearly, we change the way we write the formula about the surrogate-fairness gap as follows:
>
> $$
> \underbrace{\widehat{DDP} - \widetilde{DDP}(\phi)}\_{\text{surrogate-fairness gap}} = \frac{\sum\_{(\mathbf{x} ,y) \in \mathcal{N}\_{1a} \cup \mathcal{N}\_{0a}}\left[\mathbb{1}\_{d_\theta(\mathbf{x})>0}-\phi(d_\theta(\mathbf{x}))\right]}{N\_{1a}+N\_{0a}}-\frac{\sum\_{(\mathbf{x} ,y) \in \mathcal{N}\_{1b} \cup \mathcal{N}\_{0b}}\left[\mathbb{1}\_{d\_\theta(\mathbf{x})>0}-\phi(d_\theta(\mathbf{x}))\right]}{N\_{1b}+N\_{0b}}.
> $$
>
> There is a surrogate-fairness gap between $\widehat{DDP}\_{\mathcal{S}}$ and $\widetilde{DDP}\_{\mathcal{S}}(\phi)$.
> - For $\widetilde{DDP}\_{\mathcal{S}}(\phi)$, it can serve as a fairness constraint or regularization in the algorithm. The algorithm will automatically find a solution that penalizes large $\left|\widetilde{DDP}\_{\mathcal{S}}(\phi)\right|$.
> - Unfortunately, it is different for the "gap", which comes from the inherent difference between the indicator function and the fairness surrogate function. For the ideal case $\phi(x)=\mathbb{1}\_{x>0}$, which means that $\phi(d\_\theta(\mathbf{x})) = \mathbb{1}\_{d\_\theta(\mathbf{x})>0}$, the gap is zero and reducing the constraint or regularization term $\left|\widetilde{DDP}\_{\mathcal{S}}(\phi)\right|$ is equivalent to reducing $\left|\widehat{DDP}\_{\mathcal{S}}\right|$.
> - However, for any surrogate function $\phi$, the gap will be inevitably introduced unless $\phi(x) = \mathbb{1}\_{x>0}$. When the surrogate-fairness gap is small enough, there is fairness guarantee for the classifier. In practice, Figure 1(b) shows the surrogate-fairness gap of different fairness surrogate functions, including CP (which is equivalent to linear surrogate function), hinge, log-sigmoid as well as sigmoid, and our general sigmoid surrogate. It suggests that unbounded surrogate functions tend to exhibit a larger surrogate-fairness gap. The bounded surrogate functions, such as sigmoid and general sigmoid, both exhibit a bounded "gap".

---

> ### Author Response · Authors · 2024-03-08
> **Response to Reviewer AFbU (Part 3)**
>
> **Q6**: Could we use other non-linear functions to replace with sigmoid in section 5.1?
>
> **A6**: We agree with you. The goal of fairness surrogate functions in this paper is to approximate the indicator function.
> - As can be seen in Figure 1, we propose the general sigmoid surrogate to better approximate it compared to existing surrogate functions.
> - However, in practice, the effect of surrogate function is also related to the dataset. We believe that more complex and even dynamic no-linear functions can be used. Such no-linear mapping can be learned via a neural network to automatically fit the flexible and complex mapping for each dataset.
> - We hope that our work will help researchers better understand the fairness surrogate functions. We also hope that we have inspired interested researchers to invent fairness surrogate functions that can automatically fit the data distribution.
>
> We incorporate this discussion about future surrogate function into Appendix E.1 and leave such interesting exploration for future work.

---

> > ### Comment · Reviewer_AFbU · 2024-03-09
> > **Thanks for the response**
> >
> > My major concerns have been addressed, the updated version is more clear.

---

> > > ### Author Response · Authors · 2024-03-17
> > > **Thanks again for your constructive comments**
> > >
> > > Thanks very much for your time. Any comments and discussions are welcome!

---

### Decision · Action_Editor_fyVy · 2024-03-28

**Recommendation:** Accept as is

**Comment:**

This paper tackled an important issue in algorithmic fairness - how to measure unfairness. Most existing algorithmic fairness papers simply claim that fairness is improved by smaller fairness surrogates value. Does this really improve the actual fairness? This paper considered the gap between the surrogate fairness metrics and actual fairness. Detailed theoretical analysis and experiments are conducted.

**Decision**: All the reviewers appreciated the paper and voted for acceptance. This AE supports this decision.

**Audience:**

Yes

**Claims And Evidence:**

Yes